# Electronic structure modulation of iron sites with fluorine coordination enables ultra-effective $H_2O_2$ activation

Deyou Yu [1,5], Licong Xu[1,5], Kaixing Fu[2], Xia Liu[3], Shanli Wang[1], Minghua Wu[1], Wangyang Lu[4], Chunyu Lv[2] & Jinming Luo [2] ✉

Electronic structure modulation of active sites is critical important in Fenton catalysis as it offers a promising strategy for boosting $H_2O_2$ activation. However, efficient generation of hydroxyl radicals (•OH) is often limited to the unoptimized coordination environment of active sites. Herein, we report the rational design and synthesis of iron oxyfluoride (FeOF), whose iron sites strongly coordinate with the most electronegative fluorine atoms in a characteristic moiety of F-(Fe(III)O₃)-F, for effective $H_2O_2$ activation with potent •OH generation. Results demonstrate that the fluorine coordination plays a pivotal role in lowering the local electron density and optimizing the electronic structures of iron sites, thus facilitating the rate-limiting $H_2O_2$ adsorption and subsequent peroxyl bond cleavage reactions. Consequently, FeOF exhibits a significant and pH-adaptive •OH yield (~450 μM) with high selectivity, which is 1 ~ 3 orders of magnitude higher than the state-of-the-art iron-based catalysts, leading to excellent degradation activities against various organic pollutants at neutral condition. This work provides fundamental insights into the function of fluorine coordination in boosting Fenton catalysis at atomic level, which may inspire the design of efficient active sites for sustainable environmental remediation.

Clean water scarcity originating from rapid urbanization and heavy industrialization becomes one of the most severe challenges to sustainable society at a global scale[1–6]. It is imperative to explore highly efficient and economically affordable water purification technologies, especially relying on reactive oxygen species generation for pollutant degradation[7,8]. The metal-mediated Fenton catalysis based on $H_2O_2$ activation have been deemed a promising strategy for water decontamination due to its high efficiency and easy operation[9–11]. However, it has several drawbacks that limit further applications, such as harsh operation pH, low $H_2O_2$ utilization, metal-rich sludge, and unsatisfactory stability[12–14]. In addition, the insufficient •OH production and marginal $H_2O_2$-to-•OH selectivity during Fenton catalysis processes

leads to substantial chemical inputs and undesirable by-products formation[9,11,15,16]. To tackle these shortcomings, heterogeneous Fenton catalysts focusing on the microenvironment modulation of active sites via facet, heteroatom, defect, and single-atom engineering have been well-designed and developed to promote $H_2O_2$ activation[13,17–19]. Unfortunately, decomposing $H_2O_2$ molecules into •OH radicals with high yields and selectivity during Fenton catalysis remains great challenging, especially under a neutral condition.

Recently, 2D van der Waals transition-metal oxyhalides, an emerging layered nanomaterials with charming physicochemical properties, have attracted widespread interest in catalysis, energy, and environmental remediation fields[20–24]. Among various metal

[1]Engineering Research Center for Eco-Dyeing and Finishing of Textiles (Ministry of Education), Zhejiang Sci-Tech University, Hangzhou 310018, PR China. [2]School of Environmental Science and Engineering, Shanghai Jiao Tong University, Shanghai 200240, PR China. [3]College of Chemistry and Chemical Engineering, Qingdao University, Qingdao 266071, PR China. [4]School of Material Science & Engineering, Zhejiang Sci-Tech University, Hangzhou 310018, PR China. [5]These authors contributed equally: Deyou Yu, Licong Xu. ✉e-mail: jinming.luo@sjtu.edu.cn

oxyhalides, iron oxychloride (FeOCl) having octahedral $FeO_4Cl_2$ layers held by Cl−Cl van der Waals interaction is recognized as a landmark candidate for $H_2O_2$ activation at pH 7.0 because it exhibits high efficiency in the homolytic cleavage of $H_2O_2$ molecules with the highest •OH generation capacity so far[9,25]. From a mechanical perspective, the electro-withdrawn effect of Cl and O atoms coordination to active iron sites enables the reduction of local electron density via polarization effect[9,26]. This specific coordination regime contributes most in promoting Fe(III) species reduction as well as $H_2O_2$ decomposition toward •OH radicals. Afterward, significant endeavors have been made to further improve the activation performance of FeOCl by doping, intercalation, and nanoconfinement engineering while most fails to optimize •OH radical yields with satisfactory selectivity[27–29]. Interestingly, the $K^+$ intercalation into FeOCl alters the $H_2O_2$ activation pathway from •OH radicals to Fe(IV) = O species with improved tolerance to water matrix while the oxidation efficiency may be partially compromised[30].

In principle, $H_2O_2$ activation mainly relies on the peroxyl bond cleavage induced by electron transfer between active sites (e.g., iron sites) in catalysts. The electron transfer efficiency could be regulated by tuning the coordination environment of active sites, which allows for the optimization of electron states (e.g., charge density) that govern Fenton catalysis performance[8,14,31,32]. Recently, substantial efforts have been made to optimize the microenvironmental of metal active sites with rapid electron transfer and decode the mechanisms of free radical conversion by doping heteroatoms with strong electron-withdrawing functions. For instance, boron, phosphorus, oxygen, and sulfur-doped single-metal atoms with lowered local electron density demonstrate a dramatically enhancement of typical oxidants (e.g., $H_2O_2$, peroxymonosulfate, peracetic acid, etc.) activation over analogy counterparts for boosting organic pollutant degradation[13,18,33–35]. Surprisingly, the manipulation of active iron sites in the FeOCl matrix is often overlooked. Scarce efforts have devoted to introducing such electronegative moieties while great attempts have made to delivery FeOCl composites for application development despite of its relatively low •OH yields (e.g., 11 μM) and $H_2O_2$ utilization (e.g., ~0.73‰ of •OH selectivity)[9]. Inspired by above trails, we speculate that it is feasible to regulate the local electron density of iron sites by coordinating highly electronegative atoms to large improve the Fenton catalysis performance. F (3.98) is the most electronegative element in earth, and it shows the highest electron-withdrawing activity than B (2.04), P (2.19), S (2.58), N (3.04), and Cl (3.16)[35]. To this end, switching Cl atoms of O-Fe-Cl bonds with F atoms to form a O-Fe-F unique coordination would be expected to efficiently optimize the electron density of iron sites and thus enhance the $H_2O_2$ activation performance.

Herein, we rationally designed and synthesized FeOF as a powerful catalyst, in which fluorine atom strongly coordinates to iron sites in a characteristic moiety of F-(Fe(III)O₃)-F, for enabling effective Fenton catalysis with potent •OH generation. By systematically comparing the local electronic structure and Fenton catalysis behaviors of FeOF and FeOCl via both theoretical calculation and experimental analysis, we observed that the strong electron-withdrawing ability of coordinated fluoride atom could largely reduce the local electron density of iron sites and thus the energy barrier of rate-limiting step in $H_2O_2$ activation was greatly lowered. Intriguingly, the FeOF featuring such specified unique coordination environment allows for efficient •OH generation (up to ~450 μM yield and 33.2% selectivity), significantly outperforming FeOCl analogy and the state-of-the-art Fenton catalysts ever recorded. Moreover, the excellent practicality regarding degradation capability, cyclic stability, and wastewater treatment potential of the FeOF/$H_2O_2$ Fenton process was also systematically demonstrated. This work not only affords the rational design principle for Fenton catalysts at the atomic scale but also

provides fundamental insights into the enhancement mechanism of $H_2O_2$ activation.

## Results

### Synthesis and characterizations of FeOF catalyst

The FeOF catalyst was prepared by obtaining $FeSiF_6 \cdot 6H_2O$ from dissolving iron powders in fluorosilicic acid ($H_2SiF_6$) aqueous solution, followed by a simple solvothermal reaction (Fig. 1A). Representative X-ray diffraction (XPD) patterns that clearly well matched the standard cards (PDF #26-0799 and #70−1522) without additional peaks in Supplementary Fig. 1 and Fig. 1B confirmed the successful fabrication of $FeSiF_6 \cdot 6H_2O$ and FeOF catalyst with moderate crystallinity[36,37]. The downshift of (111) and (211) facets in the recorded pattern might be assigned to the presence of oxygen vacancy that expands the crystal lattice[38–40]. As illustrated by high-resolution transmission electron microscope (HRTEM) image in Fig. 1C, a lattice finger distance of 2.24 Å and 1.71 Å (Supplementary Fig. 2B and Fig. 1C) representing the (111) and (211) facets of FeOF can be observed[37], further supporting the XRD result. The FeOF catalyst displayed a collective sphere-like framework with a diameter size of ca. 300 nm, constructed from smaller nanoparticles (ca. 20-50 nm), as demonstrated by the field-emission scanning electron microscopy (FE-SEM) images (Supplementary Fig. 3) and TEM images (Supplementary Fig. 4). To our delight, the element mapping in the high angle annular dark-field scanning TEM (HAADF-STEM) image and energy dispersive X-ray spectroscopy (EDS) analysis of the prepared FeOF (Fig. 1D) verify the uniform distribution of Fe, O, and F elements. Moreover, the distinctive chemical bonding signals in the spectra (Supplementary Fig. 5) was recorded and clarified using Fourier Transform Infrared Spectroscopy (FT-IR) and X-ray photoelectron spectroscopy (XPS). Specifically, the high-resolution XPS spectrum of Fe 2$p$ (Fig. 1E) exhibited two deconvoluted peaks at 728.9 and 715.7 eV, attributing to the Fe $2p_{1/2}$ and Fe $2p_{3/2}$ in Fe(III). The distance of 13.2 eV over these two peaks and the observed shakeup satellite at 722.2 eV indicated the main existence of Fe(III) in FeOF catalyst[11,41]. The peaks referring to iron species in FeOF suffered an obvious shift to a higher binding energy against those in FeOCl, suggesting the electron density around iron sites was largely reduced by fluorine coordination[42–44], which would improve the interaction between $H_2O_2$ molecules and iron sites. A distinguishing type-IV hysteresis loop can be found in the obtained $N_2$ adsorption-desorption isotherms (Supplementary Fig. 6) of FeOF catalyst, indicative of inherent mesoporous structure (Supplementary Table 1) that can facilitate mass transfer and catalytic efficiency[45]. By comparison to FeOCl (Supplementary Fig. 7), the smaller particle size and higher surface area of FeOF are conducive to Fenton reaction by increasing the density and accessibility of active iron sites.

As previous reports demonstrated, the Fenton catalysis activity depends highly on the surface Lewis acid site originating from the inherent coordination environment of iron centers[46]. Therefore, we first performed pyridine-chemisorbed in-situ FT-IR (Py-IR) under three typical degassing temperatures (i.e., 50, 100, and 200 °C) to unravel the Lewis acidity of FeOF catalyst and systematically compare with its analogy FeOCl. The characteristic diffraction signals in XRD pattern (Supplementary Fig. 7A), nanosheet morphology in FE-SEM images (Supplementary Fig. 7B–D), and porous structures of FeOCl (Supplementary Fig. 6) suggested its well synthesis for comparison according to the landmark study[9]. These findings confirmed the structural congruence between FeOF and FeOCl for both molecular and crystal aspects. It can be found that Py-IR spectra for FeOF (Supplementary Fig. 8A) and FeOCl (Supplementary Fig. 8B) exhibit two typical IR bands at 1445 and 1480 cm$^{-1}$, which were all attributed to a pyridine-Lewis acid adduct[46–48]. Whereas, no distinct band detected at ~1540 cm$^{-1}$ excludes the appearance of Brønsted acid sites[46]. In addition, as degassing temperature increased, the band intensity in the Py-IR spectra of FeOF suffered a moderate decrease while a dramatic

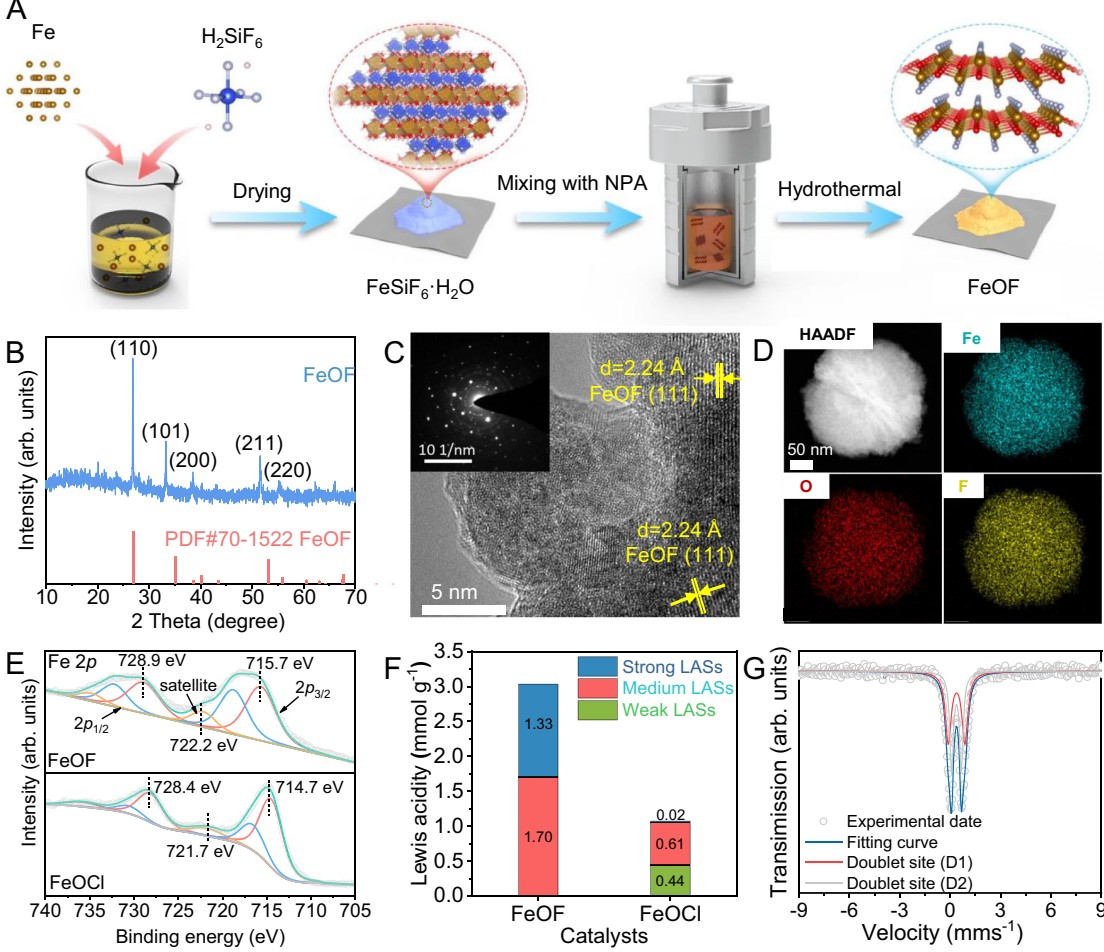

**Fig. 1 | Preparation and structural characterization of FeOF catalyst.**
**A** Schematic illustration for the preparation procedure of nanostructured FeOF catalyst. **B** XRD pattern, **C** HRTEM (inset: SAED image), and **D** HAADF-STEM images of FeOF catalyst. **E** High-resolution Fe 2*p* XPS spectra of FeOF and FeOCl catalysts. **F** Comparison of the Lewis acidity of FeOF and catalysts with different strength. **G** $^{57}$Fe Mössbauer spectroscopy of FeOF. Source data are provided as a Source Data file.

decline occurred for FeOCl. These results demonstrated that the surface of FeOF was covered by a large amount of stronger Lewis acid sites than that of FeOCl, which primarily identified the effect of higher electronegativity of fluoride on significantly reducing the local density electron of iron sites. For a more direct comparison, the Lewis acidity of different acid types including weak, medium, and strong, revealed by the Py-IR spectra recorded at different degassing temperatures, was quantified according to previous reports[46,49,50], and summarized in Fig. 1F. The total Lewis acidity for FeOF was calculated to be 3.037 mmol g$^{-1}$, in which the weak (0.004 mmol g$^{-1}$), medium (1.702 mmol g$^{-1}$), and strong (1.332 mmol g$^{-1}$) site make up 0.1%, 56.0%, and 43.9%, respectively. In contrast, the FeOCl showed a one in three Lewis acidity (1.068 mmol g$^{-1}$) toward FeOF, consisting of 41.2% weak and 56.9% medium sites. Therefore, it is supposed that FeOF is more favorable for Fenton catalysis as it contains high proportion of medium and strong Lewis acid sites that can facilitate the interaction with H$_2$O$_2$ for sustainable activation through the enhanced the Fe(III)/Fe(II) redox cycle[12,32,51,52].

**Coordination environment and electronic structure**
To gain a molecular structure insight into the Lewis acid site, the $^{57}$Fe Mössbauer spectroscopy was employed to discriminate different iron species in FeOF. As demonstrated in Fig. 1G and Supplementary Table 2, the $^{57}$Fe Mössbauer spectrum of FeOF shows two fitted doublets, i.e., D1 and D2, both of which have an isomer shift (IS) lower

than 2.0, suggesting that iron exists in the form of trivalent state as XPS result revealed[53,54]. Importantly, D1 with relatively larger values of IS and quadrupole splitting (QS) values can be assigned to the F-(Fe(III)O$_4$)-F high-spin structure with saturated hexa-coordination numbers, in which F atoms were shown in *cis*-conformation[54]. The structure is so robust that it is hardly employed as active sites for H$_2$O$_2$ activation. Whereas, D2 having smaller IS and QS values can be attributed to the F-(Fe(III)O$_4$) or F-(Fe(III)O$_3$)-F medium-spin structure[53]. Such an unsaturated coordination structure of D2 with fluoride as a mediator for the local electron density of iron sites enables it to be catalytically active for effective Fenton catalysis. The quantitative analysis suggested that the contents of D1 and D2 site were 54.0% and 46.0%, respectively, proving the predominance of unsaturated iron coordination structure (i.e., F-(Fe(III)O$_4$) or F-(Fe(III)O$_3$)-F moiety) in FeOF catalyst.

To further elucidate the local structure information at the atomic level, we examined the electronic structure and coordination environment of iron sites by X-ray absorption spectroscopy (XAS). As shown in Fig. 2A, Fe K-edge X-ray absorption near-edge structure (XANES) spectra indicate that the absorption energy of FeOF was evidently dissociate to Fe foil but intimate to Fe$_2$O$_3$ and FeF$_3$, implying that Fe species in FeOF carries positive charge toward +3[41]. Noteworthily, the valence of Fe species was calculated to be +2.95 based on the linear fitting curve (Fig. 2B) of Fe valence versus Fe K-edge energy that determined from maximum in the first-order derivative. This indicated the popularity of Fe(III) and the presence of limited amount

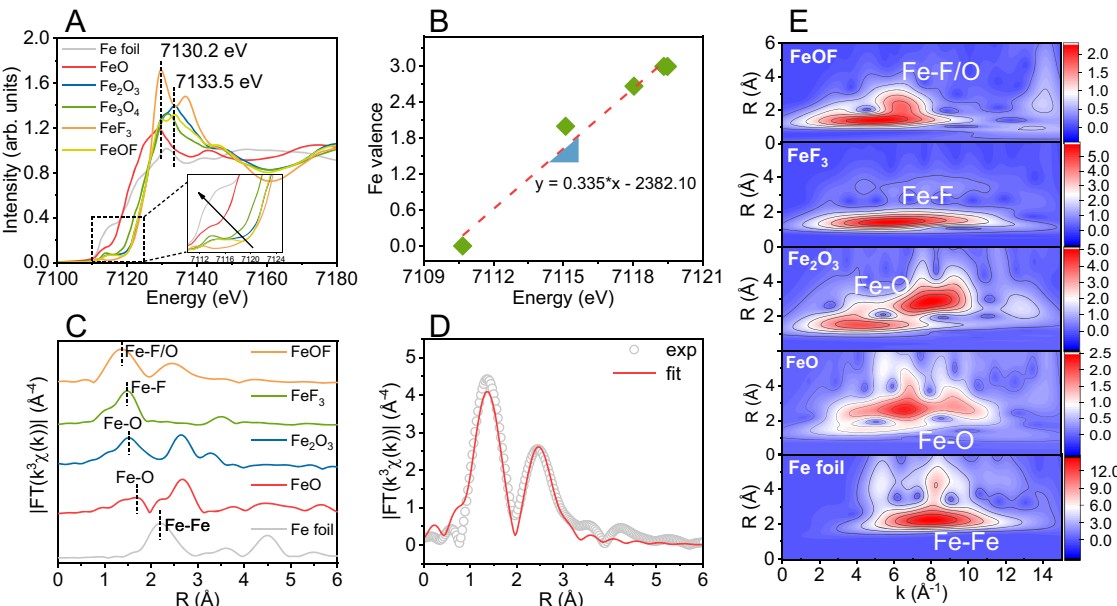

**Fig. 2 | Coordination environments of the active iron sites in FeOF catalyst demonstrated by X-ray absorption spectroscopy and DFT calculations.**
**A** Normalized Fe K-edge XANES spectra of FeOF and reference samples (i.e., Fe foil, FeO, $Fe_2O_3$, $Fe_3O_4$, and $FeF_3$). **B** Chemical valence of iron sites in FeOF and reference samples. **C** Fe K-edge FT-EXAFS of FeOF and reference samples. **D** Fe K-edge EXAFS fitting analysis for FeOF catalyst. R in the x-axis label refers to radial distance. **E** Wavelet transform for the $k^3$-weighted Fe K-edge EXAFS spectra of FeOF and reference samples. Source data are provided as a Source Data file.

of oxygen vacancies in FeOF. Moreover, the lower normalized intensity of white peak locating at near-edge (7130.2 eV) in the XANES spectrum of FeOF implied the five-fold coordination by F and O atoms for unsaturated coordination iron sites[53,55]. To clarify the atomic configuration of iron species in FeOF, the extended X-ray absorption fine structure (EXAFS) spectrum was further analyzed by the Fourier transformed (FT) $k^3$-weighted $\chi$ (k) function (FT-EXAFS). As depicted in Fig. 2C, the FT-EXAFS curve of FeOF presents two main peaks at 1.5 and 2.5 Å, which were assigned to the Fe-F/Fe-O and Fe-Fe shells, respectively[53]. Wavelet transform (WT) was also applied to evaluate the Fe K-edge EXAFS oscillations of FeOF and the references (i.e., Fe foil, FeO, $Fe_2O_3$, $Fe_3O_4$, and $FeF_3$). As displayed in Fig. 2E, we can clearly observe a WT maximum at ~8.1 Å$^{-1}$ that corresponds to the Fe-Fe configuration in Fe foil rather than in FeOF. Comparative intensity maximum at ~5.1 Å$^{-1}$ for FeOF suggested the co-existence of Fe-F and Fe-O coordination bonds as the presence of fluoride atom enables a slight shift toward higher Å$^{-1}$ than individual Fe-O coordination bond found in pure $Fe_2O_3$ reference (~4.2 Å$^{-1}$). According to WT-EXAFS features on $FeF_3$, the location of WT maximum for Fe-F bond is similar with that on FeOF, highlighting the presence Fe-F bond in FeOF. Moreover, a least-square fitting of EXAFS and FT-EXAFS was conducted to obtain quantitative structural parameters of iron site in FeOF and the references. The fitting curves are illustrated in Fig. 2D, and the obtained structural parameters are summarized in Supplementary Table 3. The double-shell fitting based on the Fe-F and Fe-O coordination models that extract from $FeF_3$ and $Fe_2O_3$ conformed well to the experiment result, manifesting the F, O dual coordination for the iron site in FeOF. The coordination numbers of F and O atoms for the iron site were determined to be 1.9 and 2.7 at distances of 1.99 and 1.90 Å, respectively, which is similar to the coordination environment of FeOCl (~1.8 Cl atoms and ~3.5 O atoms) demonstrated by a previous report[56]. This observation justified the characteristic moiety of F-(Fe(III)O₃)-F in the fine local structure elucidation of FeOF, which was consistent with Mössbauer spectroscopy and XANES analysis.

Inspired by above analysis, we compared the electronic structures of FeOF and FeOCl catalysts using density functional theory (DFT) calculations. The applied molecular structure geometries of FeOF and FeOCl after optimization for electron density calculations are

illustrated in Supplementary Fig. 9A. Given its much higher electro-negativity than chlorine atom, the fluoride atom in the as-prepared FeOF enables to alter and optimize the electronic structure of active iron sites. This can be proved by the differences between the two-dimensional valence-electron density color-filled maps of FeOF and FeOCl. Therefore, we first depict the corresponding two-dimensional valence-electron density color-filled maps in Supplementary Fig. 9B. Evidently, the largest electronic distribution area emerged around fluoride atom for FeOF and chloride atom for FeOCl. The preferential electron flow from Fe toward themselves due to the higher electro-negativity than another coordination oxygen atom. The chloride and fluoride atom possess relative higher negative-charge while the neighboring coordinated iron center presents considerably lower electron density. In particular, the maximum valence-electron density of iron center in FeOF was ca. 0.04 $e\,Å^{-3}$ (Supplementary Fig. 10), which was 5 times smaller than that in FeOCl (ca. 0.20 $e\,Å^{-3}$), theoretically confirming the extensive modulation effect of the fluoride atom through strong polarization. Moreover, the valence state analysis indicated that the iron center loses ca. 1.71 $e$ mainly transferring to the neighboring fluoride atom in FeOF while the chloride atom in FeOCl only triggered ca. 0.43 $e$ withdrawn from the neighboring iron center with a total loss of ca. 1.47 $e$ (Supplementary Fig. 9B). These results strongly supported our speculation that the highest electronegative fluoride atom allows for the large reduction of local electron density of iron sites, thereby improving the Fenton catalysis activity, which was further discussed in next section.

## Fenton catalysis performance evaluation

The generation of •OH is one of the most significant descriptors to evaluate the performance of $H_2O_2$-based Fenton catalysis. We thereby qualitatively and quantitatively investigated the •OH generation of FeOF and FeOCl catalysts using electron paramagnetic resonance (EPR) technique and fluorescence method, respectively. While using 5,5-dimethyl-1-pyrroline N-oxide (DMPO) as a trapping agent, we observed four distinctive EPR peaks (Fig. 3A) with an intensity ratio of 1:2:2:1 for the DMPO-•OH adduct, confirming the production of •OH in both FeOF and FeOCl samples[11,57]. Notably, the relative intensity of EPR signals in the neutral FeOF/$H_2O_2$ system largely outcompeted that of

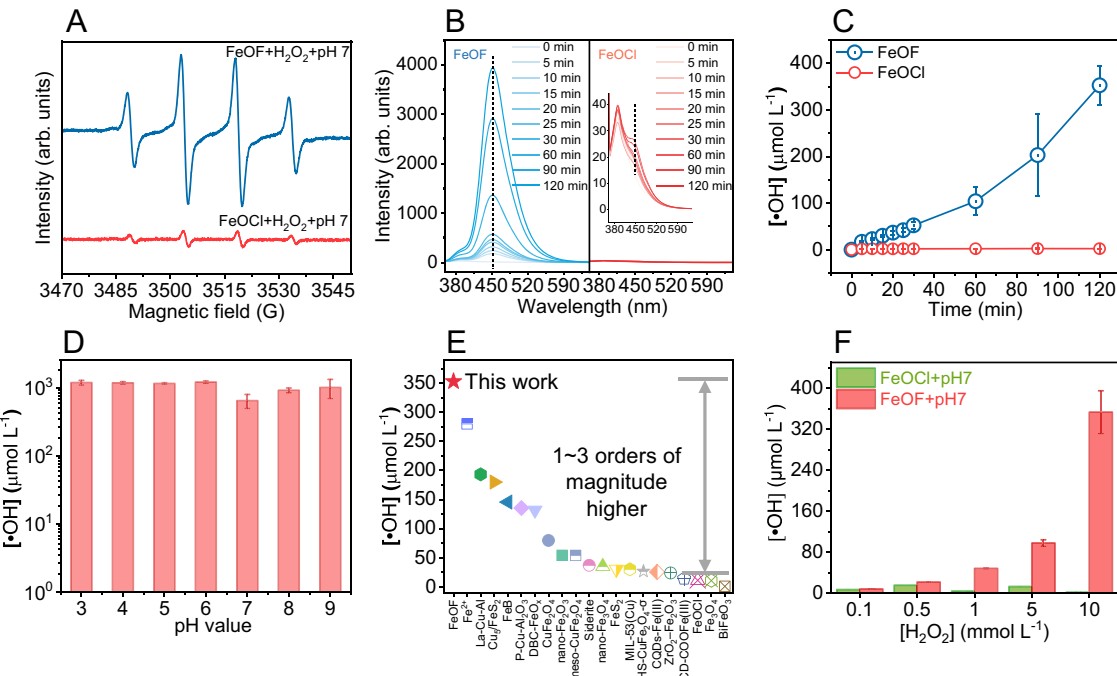

**Fig. 3 | Fenton catalysis performance of FeOF and FeOCl catalysts. A** DMPO-•OH spectra, **B** florescence intensity versus reaction time, and **C** accumulated •OH generation for FeOF and FeOCl systems. The florescence signal of 7-hydroxycoumarin is highlighted by the dash line. **D** Accumulated •OH generation at different pH values for FeOF system. Error bars represent the standard deviation of the experiment in triplicate. **E** Comparison of the accumulated •OH generation over the state-of-the-art Fenton catalysts demonstrated by previous studies. The

relevant references are listed in Supplementary Table 5, Supplementary Information. **F** Accumulated •OH generation at various $H_2O_2$ inputs for FeOF and FeOCl systems. Reaction conditions: $[H_2O_2] = 10$ mM (unless otherwise specified), [catalyst] = 0.1 g $L^{-1}$, [Coumarin] = [MeOH] = [TBA] = 10 mM, pH = 7.0 (unless otherwise specified), Temperature = 20 °C. Source data are provided as a Source Data file.

FeOCl/$H_2O_2$ counterpart. This result suggested that the rationally designed FeOF exhibit much better Fenton catalysis performance than FeOCl. Moreover, the intensity of characteristic signals in FeOF (Supplementary Fig. 11) could be largely declined with the addition of tert-butanol (TBA), a typical •OH scavenger[58]. This further indicated the ascendency of •OH generation in $H_2O_2$ activation using FeOF. However, the addition of 10 mM of TBA fails to completely quench the generated •OH, implying the formation of small amount of surface-bounded •OH (•$OH_{surf}$). Impressively, similar EPR signals of DMPO-•OH adduct were recorded in the Fenton catalysis process under various pH conditions (Supplementary Fig. 12), confirming the production of •OH at a wide pH range. This observation demonstrated that the as-prepared FeOF catalyst is expected to exhibit a broader application in different water matrixes conditions due to its high pH-adaptability, which is a long-term pursued goal in reforming Fenton chemistry because the negative effect of harsh operation pH is difficult to overcome. Enlightened by these results, we deduced that the fluoride atom coordinated to active iron sites enables the as-prepared FeOF to selectively activate $H_2O_2$ with a greater ability than that of FeOCl, showing a more promising potential for Fenton catalysis with high pH-adaptability.

To quantify •OH generation, a fluorescence method[59] using coumarin as the probe was subsequently conducted to examine the specific activity of FeOF and its analogy FeOCl. As depicted in Fig. 3B, C, we observed a margin •OH generation with a normalized rate of 0.12 μM $min^{-1}$ in FeOCl/$H_2O_2$ system at the first 15 min while the accumulated concentration reached to 2.1 μM at 120 min, which is consistent with previous reports[9,25]. In contrast, FeOF/$H_2O_2$ system exhibited a rapid •OH generation with a normalized rate of 2.94 μM $min^{-1}$ and an accumulated concentration of 353.0 μM, which were 17.5 and 168.0 times higher than that of FeOCl/$H_2O_2$ system. We also excluded the homogeneous contribution of leached iron ions

because scarce •OH was generated when using $Fe^{2+}$, $Fe^{3+}$, and $Fe^{2+}/Fe^{3+}$ mixture as the catalyst at equivalent concentrations to the leaching level (Supplementary Fig. 13). The surface area normalized kinetic rate constant for •OH generation by FeOF was determined to be 31.7 μmol g $L^{-1}$ $min^{-1}$ $m^{-2}$, 85.4 times higher than by FeOCl (Supplementary Table 4). Such increment closing to the •OH generation enhancement reveals that the smaller particle size and higher surface area are not the primary factors. Consequently, we deduced that the disparity of $H_2O_2$ activation activity could be ascribed to the presence of F-Fe-O sites, where the iron atom is expected to show much lower electron density over Cl-Fe-O. This capacity could be maintained at a wide pH range (3.0-9.0) since a relative steady •OH generation was monitored and confirmed in Fig. 3D, demonstrating the surprising pH-adaptability of FeOF. In particular, the FeOF catalyst outperformed all the ever-reported heterogeneous Fenton catalysts, showing a great •OH generation capacity (Fig. 3E and Supplementary Table 4). The neutral FeOF/$H_2O_2$ system also outperforms the traditional homogeneous Fenton process initiated by Fe(II) ions at pH 3.0, regarding the produced •OH amount (353.0 μM $vs$ 280.1 μM) (Supplementary Fig. 14) and $H_2O_2$ utilization efficiency (Supplementary Table 5). Notably, the $H_2O_2$ utilization efficiency toward selective •OH generation is found to be 33.2% at neutral condition, which is 1-3 orders of magnitude higher than other Fenton systems facilitated by the state-of-the-art catalysts (Supplementary Table 5). It is also worthy mentioned that even at comparatively low $H_2O_2$ concentrations, such as 0.10, 0.50, 1.00, and 5.00 mM, the FeOF was still capable to generate 8.2, 21.7, 48.4, and 98.0 μM of •OH respectively (Fig. 3F). Whereas, the counterpart FeOCl is hard to produce •OH from $H_2O_2$ selective decomposition at the equal condition. Overall, it is clear that the FeOF demonstrates an excellent Fenton catalysis activity for efficient •OH generation with wide operation pH and high $H_2O_2$ utilization efficiency (i.e., selectivity). This superior activity of FeOF for neutral Fenton process may be

derived from the unique coordination configuration and electronic structure of iron sites mediated by the fluoride atom with high electronegativity.

## Fluorine coordination dominates H₂O₂ activation enhancement

The Lewis acid-base interaction between catalysts and $H_2O_2$ is recognized as the initial step and is of significant importance for $H_2O_2$ activation[46]. To clarify the role of Lewis acid sites (i.e., iron sites) within FeOF catalyst on $H_2O_2$ activation, we assessed the effect of phosphate, a strong Lewis base for blocking interaction between iron sites and $H_2O_2$ molecules[60], on •OH generation. As shown in Supplementary Fig. 15A, the phosphate addition completely suppresses the •OH production confirmed by the no detection of 7-hydroxycoumarin (Supplementary Fig. 15B), demonstrating the unique catalysis role of iron sites for activating $H_2O_2$ into •OH. Although our experimental findings suggested that FeOF had a somewhat higher Lewis acidity than FeOCl (~3× higher, Fig. 1F), this alone failed to explain the much higher observed disparity in the accumulated generation of •OH (~168× higher). Even if we normalized the apparent kinetic rate constant for •OH generation by the total Lewis acidity (Supplementary Table 4), the value for FeOF was 60 times greater than FeOCl. We observed that the ratio of Lewis acid sites with strong strength greatly increased from 1.9% to 44.9% as the fluorine coordination was integrated. Meanwhile, the ratio of Lewis acid sites with medium strength also increased by more than 2 times, accompanied by the disappearance of weak-strength acid sites. According to previous reports[61–63], the type of Lewis acid site typically depends on the electron density of metal sites, which governs the interaction with Lewis bases. The observed increased ratio of stronger acid sites verifies the electron-withdrawing function of fluorine coordination. Therefore, we attribute this significant enhancement to the function of fluorine coordination, which optimizes the electronic structure of iron sites through the high electronegativity. We then further applied DFT calculations to

comprehensively elucidate the interaction between iron sites and $H_2O_2$ molecule at an atomic level.

The interaction configuration is pivotal for digging the origin of Fenton catalysis performance improved by fluorine coordination. By revealing the adsorption energy ($E_b$) and charge transfer, the ascendancy of F-Fe coordination on $H_2O_2$ activation can be clearly demonstrated. In this respect, we systematically calculated these two depicters at four main facets, i.e., (101), (110), (211), and (010), in FeOF and FeOCl, which were extracted from the XRD patterns with high peak intensity. As illustrated in Fig. 4A, B and Supplementary Fig. 16, the corresponding $E_b$ of $H_2O_2$ on the Fe at all calculated facets of FeOF were determined to be −0.93, −0.84, −0.91, and −0.21 eV, respectively, which were lower than those of FeOCl (i.e., −0.43, −0.06, −0.73, and −0.15 eV, respectively). This can be attributed to the stronger electron polarization effect of F over Cl. Notably, the length of O-O bond (denoted as $l_{O-O}$) within $H_2O_2$ molecule prolongs after adsorbing onto (110) and (211) facets of FeOF (Supplementary Table 6). The $l_{O-O}$ is also longer than that adsorbed onto the facets in FeOCl, indicating the superior capacity of FeOF for $H_2O_2$ activation[11,58,64,65].

To unravel the electronic mechanism dominating the Fenton activity, we calculated the projected density of state (PDOS) for both $H_2O_2$ before and after adsorption onto differing facets of FeOF and FeOCl. In addition, the charge density of $H_2O_2$-FeOF and $H_2O_2$-FeOCl complexes were also investigated to elucidate the electron transfer over $H_2O_2$ and catalysts. It is apparent that the overall energy levels of $H_2O_2$ downshift to more negative positions after adsorbed to (101), (110), and (211) facets of FeOF and all the screened facets of FeOCl (Supplementary Fig. 17). Specifically, the highest occupied molecular orbital (HOMO) of $H_2O_2$ is subjected to an obvious splitting after being adsorbed to (101) facets of FeOF and FeOCl. However, new bands caused by splitting contribute slightly above the Fermi level despite of the occurrence of charge depletion and accumulation. This indicated that $H_2O_2$ was inclined to be physically adsorbed onto (101) facets of

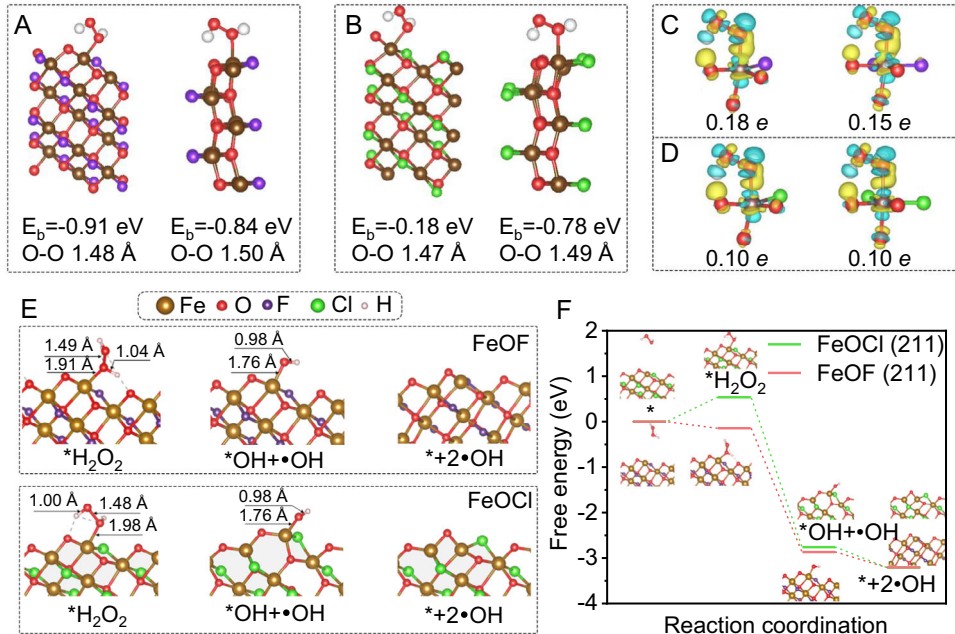

**Fig. 4 | Theoretical investigation of H₂O₂ activation mechanism with DFT calculations.** Optimized adsorption conformations of $H_2O_2$ at Fe site on the (211) (left) and (110) (right) facets of FeOF (**A**) and FeOCl (**B**). Charge density difference for $H_2O_2$ adsorption at Fe site on the (211) (left) and (110) (right) facets of FeOF (**C**) and FeOCl (**D**). Yellow and blue contours represent electron accumulation and deletion, respectively. The isosurface contour is 0.003 e/Å³. **E** The structures of reaction

intermediates on (211) facet for the generation of •OH. At initial step (i), the iron site on the surface of (211) facet shows tetrahedral coordination structure with 3 oxygen atoms and 1 fluorine or chlorine atom. The coordination number of iron site increases by one upon $H_2O_2$ adsorption and returns to the original ones after $H_2O_2$ activation and •OH dissociation. **F** Calculated free energy profiles for $H_2O_2$ activation and •OH formation on (211) facet of FeOF and FeOCl.

both FeOF and FeOCl rather than to be chemically activated[11,58]. On the contrary, the split HOMO of $H_2O_2$ adsorbed onto (110) and (211) facets (Supplementary Fig. 17) were partially unoccupied accompanying with the formation of new states beyond the Fermi level, indicating the occurrence of chemisorption[11,13,35]. In particular, the molecular energy level of $H_2O_2$ adsorbed onto FeOF was much lower than that onto FeOCl. The calculated PDOS results showed that the formed $H_2O_2$-FeOF complex on (110) and (211) facets after adsorption have a higher density state near the Fermi level as compared to the $H_2O_2$-FeOCl complex. Meanwhile, Bader charge analysis revealed that O atoms in $H_2O_2$ contributed 0.10 $e$ to iron centers at both (110) and (211) facets of FeOCl while more charges (i.e., 0.15 and 0.18 $e$) transferred from O to Fe atoms at corresponding facets of FeOF were evidently observed (Fig. 4C, D, and Supplementary Fig. 18). We noted that the charge depletion and accumulation detected in optimized $H_2O_2$-catalyst geometry was highly consistent with the adsorption energies of $H_2O_2$ (Supplementary Table 6) on all facets of catalysts excluding the (010) facet (Supplementary Fig. 18). These results manifested that FeOF exhibits a stronger activation capacity toward $H_2O_2$ than FeOCl via improved electron mobility, which can be ascribed to the robust electron polarization effect of fluoride atoms. However, the facet-dependent catalytic performance of FeOF for $H_2O_2$ activation still needs to be further explored.

The generation of •OH and intermediate species during Fenton process were subsequently compared for FeOF and FeOCl. The •OH generation occurring on the various crystal surfaces of the catalyst includes $H_2O_2$ adsorption and consequent reactions. The free energy diagram for the crystal surfaces of catalysts was optimized as following: * (i) → *$H_2O_2$ (ii) → *OH + •OH (iii) → *+2•OH (iv) (Fig. 4E, F), where * indicates the adsorption state. The changes in the free energies for •OH formation in (211), (110), (101), and (010) facets of FeOF and FeOCl are displayed in Fig. 4E, F, and Supplementary Figs. 19–21. The coordination environment evolutions of corresponding iron sites are presented in the caption. The $H_2O_2$ adsorption (ii) was observed as the rate-limiting step for all facets due to the highest energy barrier. Notably, $H_2O_2$ adsorption on FeOF catalyst showed much lower free energy than FeOCl for (211), (110), and (101) facets, indicating the easier access of iron centers coordinated with fluoride atoms. Therefore, $H_2O_2$ adsorption to FeOF was thermodynamically more favorable than to FeOCl. Subsequently, the $H_2O_2$ on catalyst surface initiated to exo-thermically split into *OH accompanying with one •OH desorption (iii) followed by the dissociation of another •OH (iv) (Fig. 4E). The energy barriers requiring for $H_2O_2$ split and •OH desorption on the (211) (−2.87 eV), (110) (−2.54 eV), and (101) (−2.45 eV) facets of FeOF were found to be much lower than those of FeOCl (Supplementary Table 7), demonstrating the superior activity of FeOF for $H_2O_2$ activation. Of note, although $H_2O_2$ adsorption to (010) facets of FeOF and FeOCl show a similar free energy value (ca. 0.9 eV), the following split reaction demonstrated a relative higher activation energy barrier over (211), (110), and (101) facets (Supplementary Fig. 21), suggesting the inferior catalytic performance of (010) facets. Interestingly, the rate-determining step (i.e., $H_2O_2$ adsorption) in calculated facets (>0 eV) was endothermic except the (211) (−0.14 eV) facet of FeOF (Fig. 4F and Supplementary Table 7). Theoretically, the catalyst with the lowest free energy in rate-limiting step would show optimum activity for Fenton reactions[58,65]. It is therefore expected that the (211) facet of FeOF out-performs the additional facets in terms of $H_2O_2$ activation due to the exothermic nature of $H_2O_2$ adsorption following by the spontaneous decomposition[14,58]. We conducted ab initio molecular dynamics (MD) simulations to further unveil the evolution and fate of $H_2O_2$, detailed in Supplementary Fig. 22. $H_2O_2$ adsorbed on FeOF could be readily cleaved into two •OH radicals without energy barriers to overcome. Conversely, the adsorbed $H_2O_2$ on FeOCl cannot split into •OH radicals. The divergence in $H_2O_2$ split performance between FeOF and FeOCl further highlights the unique function of fluorine coordination.

Overall, by changing the coordination environmental of iron centers with more electronegative fluoride atoms, the Fenton activity could be largely improved through optimizing the energy barrier for $H_2O_2$ adsorption and subsequent splitting reactions.

As electron transfer is responsible for $H_2O_2$ activation, we measured the Tafel polarization curves, linear sweep voltammetry (LSV) plots, and electrochemical impedance spectroscopy (EIS), to detect real electron mitigation property of FeOF in Fenton process using electrochemical methods[60]. According to the Tafel polarization curves (Supplementary Fig. 23), the corrosion current values were determined to be $8.49 \times 10^{-7}$ and $1.640 \times 10^{-8}$ A while the corrosion current densities reached to 4.33 and 0.08 $\mu A \, cm^{-2}$ for FeOF and FeOCl, respectively. The large difference of corrosion current values and densities imply that FeOF exhibited a superior electron transfer performance and catalytic reactivity than FeOCl toward $H_2O_2$ activation. The LSV plot of FeOF (Supplementary Fig. 24) presented distinctly larger current density over the analogy, further confirming the faster electron migration between FeOF and $H_2O_2$ for a rapid •OH generation and enhanced Fenton reaction. In addition, the relative lower arc radius observed in the EIS spectrum (Supplementary Fig. 25) of FeOF catalyst matches well with the findings of Tafel polarization curves and LSV plots. These results experimentally elucidate that the electronic structure engineering of active iron sites via coordinatively bonding fluoride atoms with the highest electronegativity significantly can promote the electron migration for the enhanced •OH generation, which shows promising Fenton catalysis activity for potential applications.

## Performance of FeOF/$H_2O_2$ system for practical water purification

The requirement of acid environment (pH 3.0-4.0) for $H_2O_2$ activation often causes undesirable complexity and difficulty, largely retarding the practicality of Fenton process[9,30]. Attributed to the unique fluorine coordination, the FeOF catalyst can easily trigger $H_2O_2$ decomposition into abundant •OH at wide pH range, which is expected to tremendously facilitate the water purification applications for different treating purposes. As a concept of proof, we examined the degradation efficiency of FeOF/$H_2O_2$ system using p-nitrophenol (4-NP), a refractory organic pollutant to ROS (e.g., $^1O_2$ and $\cdot O_2^-$) with moderate oxidation capacity, as the model substrate. The degradation of 4-NP as a function of time during Fenton reaction is illustrated in Fig. 5A. The FeOCl/$H_2O_2$ system could only remove about 6.2% of 4-NP under current conditions. The 4-NP decomposition rate reaches ~100% at pH 7.0 accompanying a total organic carbon (TOC) removal of 63% (Supplementary Fig. 26) within a short period (~30 min) at a pseudo-first-order rate constant ($k_{obs}$) of 0.362 min$^{-1}$ (Fig. 5B), which was 19.0 and 12.5 times higher compared to FeOCl (0.019 min$^{-1}$) and the conventional ferrihydrite (0.029 min$^{-1}$) catalyst, being one of the most active Fe-based catalysts for $H_2O_2$ activation so far (Supplementary Table 4). However, 4-NP can be hardly removed by individual $H_2O_2$ (Supplementary Fig. 27), FeOF alone (Supplementary Fig. 28) or leaching iron ions (Supplementary Fig. 29) at neutral condition. In particular, the FeOF/$H_2O_2$ system shows similar performance to the typical homogeneous Fenton process (pH 3.0) initiated by $Fe^{2+}$ ions with the equivalent dosage (Supplementary Fig. 30). We also observed that less than 0.40 mg L$^{-1}$ of fluoride leached during Fenton reaction (Supplementary Fig. 31), which conformed to the drinking water standards issued by the United States of America (2.00 mg L$^{-1}$), World Health Organization (1.00 mg L$^{-1}$), and China (1.00 mg L$^{-1}$).

Considering the possibility that empty orbitals of unsaturated iron centers in FeOF play a decisive role in $H_2O_2$ activation, we also attempted to block the orbital via a Lewis acid-base interaction using phosphate. Similar to •OH generation inhibition, we observed a strong suppression of 4-NP decay with almost zero percent degradation rate (Fig. 5C), suggesting that unsaturated iron centers (e.g., F-(Fe(III)$O_3$)-F

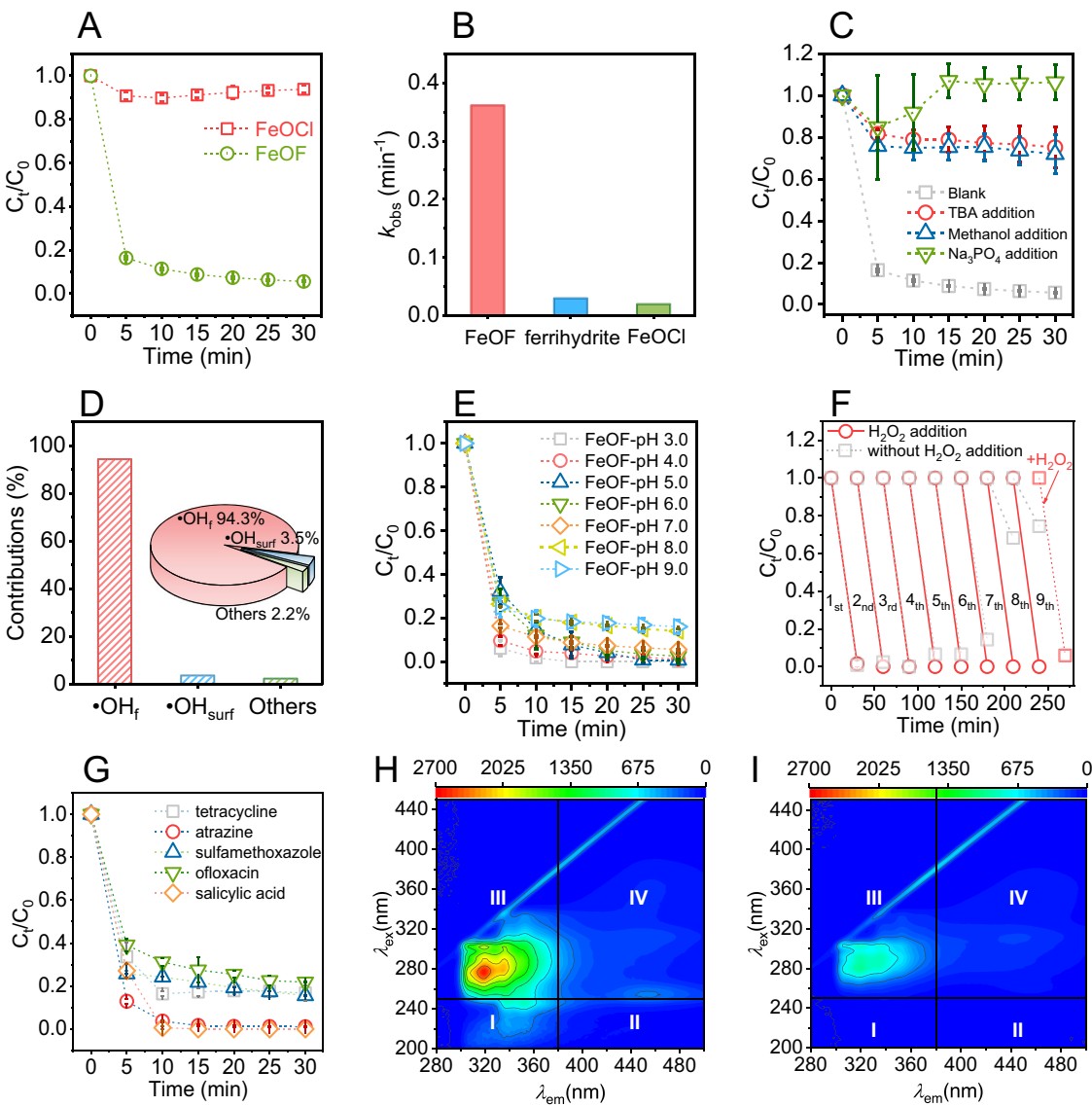

**Fig. 5 | Application potential of FeOF/H$_2$O$_2$ system for water purification.**
**A** 4-NP degradation kinetics in the neutral FeOF/H$_2$O$_2$ and FeOCl/H$_2$O$_2$ systems.
**B** Comparison of the observed rate constant of 4-NP degradation by FeOF, FeOCl, and ferrihydrite catalysts. **C** Quenching or inhibition effect of TBA, methanol, and Na$_3$PO$_4$ on the 4-NP degradation efficiency by FeOF. **D** Contribution of •OH (i.e., •OH$_f$ and •OH$_{surf}$) and others on the 4-NP degradation. **E** Influence of pH values on 4-NP degradation in the FeOF/H$_2$O$_2$ system. **F** 4-NP degradation in eight successive cycles in the neutral FeOF/H$_2$O$_2$ system with and without adding H$_2$O$_2$ after each

running. Note that the 4-NP degradation efficiency could be recovered by adding H$_2$O$_2$ in the 9th running. **G** Degradation of various organic pollutants (i.e., tetracycline, atrazine, sulfamethoxazole, ofloxacin, and salicylic acid) by neutral FeOF/H$_2$O$_2$ system. EEM spectra of collected dyeing & printing wastewater before (**H**) and after treatment (**I**) by neutral FeOF/H$_2$O$_2$ system. Reaction conditions: [H$_2$O$_2$] = 10 mM, [catalyst] = 0.1 g L$^{-1}$, [Pollutant] = 20 mg L$^{-1}$, [MeOH] = [TBA] = 10 mM (if used), [Na$_3$PO$_4$] = 2.0 g L$^{-1}$ (if used), pH = 7.0, Temperature = 20 °C. Source data are provided as a Source Data file.

moiety) is responsible for 4-NP destruction. Moreover, we also observed that both methanol and tert-butanol, two representative scavengers for •OH, showcased a severe inhibition effect in 4-NP removal at an excessive dosage (Supplementary Fig. 32). The residual 4-NP decomposition rate (-1.0%) is probably attributed to the marginal adsorption capability of FeOF. In particular, the 4-NP decomposition rate showed an evident decline from 95.4% to 28.1% (Fig. 5C) in the presence of TBA at a relatively lower concentration (i.e., 10 mM). The equal addition of MeOH retarded 70.6% of 4-NP being degraded by FeOF/H$_2$O$_2$ system, which was about 2.3% higher than TBA inhibition, highlighting the slight contribution of surface-bound •OH radicals for 4-NP degradation. The contribution of free and surface-bound •OH radicals were quantified and calculated according to previous study using the $k_{obs}$ value before and after quenching[58]. Specifically, the free and surface-bound •OH radicals account for 94.3% and 3.5% of the contributions (Fig. 5D) to the neutral Fenton process initiated by FeOF

catalyst. The inhibition results manifested that both the free and surface-bound •OH are generally produced via Fenton reaction, which matched well with our observations in EPR and •OH probing results.

The relationship between the Fenton degradation activity and reaction parameters, (e.g., pH value, temperature, and H$_2$O$_2$ dosage) was also investigated. Increasing pH values from 3.0 to 9.0 moderately impeded the 4-NP decay (Fig. 5E). This phenomenon was attributed to the relatively low redox potential of formed •OH under alkaline conditions. Nevertheless, the $k_{obs}$ underwent a slight decrement with over 90% decomposition rate of 4-NP as pH increased to 9.0. Given the pH-adaptive Fenton performance of FeOF/H$_2$O$_2$ system, the long-lasting challenge to produce •OH at a wide pH for applications, especially at neutral condition that closes to most of realistic scenarios, could be easily addressed via the polarization of electron distribution over iron sites with fluorine coordination. Unlike pH impacts, the 4-NP degradation efficiency

was largely enhanced when increasing either the temperature (Supplementary Fig. 34) or $H_2O_2$ dosage (Supplementary Fig. 35). Specifically, a significant reduction of time to destruct 100% 4-NP (20 mg $L^{-1}$) was observed as 20 mmol $L^{-1}$ of $H_2O_2$ or 40 °C of temperature was applied. The 4-NP could be also quickly removed at a lower $H_2O_2$ concentration (e.g., 1.00 mM), suggesting the strong ability of FeOF to activate trace $H_2O_2$ for effective water decontaminations (Supplementary Fig. 35). We also observed that the FeOF/$H_2O_2$ system maintained satisfactory degradation capacity for successive seven times running (Fig. 5F) without adding $H_2O_2$ each time. The degradation efficiency could be recovered once $H_2O_2$ is supplemented, suggesting the excellent stability and reusability of FeOF catalyst. Similarly, successive recycle tests (Supplementary Fig. 36) with the addition of both 4-NP and $H_2O_2$ strongly indicated a self-enhanced activity feature of FeOF, which is an indicative of the formation of Fe(II). We further performed XPS, Mössbauer, and electrochemical measurements to support the speculation of Fe(II) formation (Supplementary Figs. 37–39)[53,54,66]. The results suggested that ~26.5% of Fe(II) was generated after Fenton process, which substantially facilitated the cycling of Fe(III)/Fe(II) redox as well as the degradation efficiency for 4-NP. The observation of higher open-circuit potential of Fe(III) in FeOF and smaller peak-to-peak separation ($\Delta E_p$) (from 0.40 V to 0.28 V) depicted in Supplementary Fig. 39 indicated the more facile reversibility of the Fe(III)/Fe(II) cycle of FeOF with fluorine coordination[67–69]. Besides, we also found that the proportion of leaching $Fe^{2+}$ ions increased from 3% to 42% (Supplementary Fig. 39B), further evidencing the enhancement of Fe(III)/Fe(II) redox cycle. Note that sodium chloride (NaCl) is commonly presented at various industrial wastewaters and proposed as a major interference for Fenton process with undesirable side effects. We did not observe any marginal deterioration in 4-NP decay (Supplementary Fig. 40) even at a concentration as high as 5.0 g $L^{-1}$, demonstrating the FeOF/$H_2O_2$ system shows great potential in purifying high salinity wastewaters.

We further screened a broad spectrum of organic pollutants with high recalcitrance, such as dyes and antibiotics that have wide industrial and pharmaceutical applications, to investigate the versatility of FeOF/$H_2O_2$ system. As shown in Fig. 5G, FeOF is efficient to degrade all the model organics applied with a $k_{obs}$ (Supplementary Fig. 41) in a range of 0.19-0.41 $min^{-1}$ (Supplementary Fig. 42). Moreover, we observed that FeOF could inactivate 99.8% of a model microorganism *E.coli* (Supplementary Fig. 43), whereas the control system (6.8% for $H_2O_2$ only) exhibited a marginal antibacterial activity, demonstrating its superiority in bacterial deactivation. Together with the high reactivity, the as-prepared FeOF catalyst can be operated at more adaptive pH range than the state-of-the-art Fenton catalysts. It is suggested that there is no need to adjust the pH value of target wastewaters, further simplifying the overall operation process and lowering the net cost as well. To this end, we collected several secondary effluents from pharmaceutical and textile industries for testing the practicality of FeOF/$H_2O_2$ system without pH adjustment. Indicated by the observation in excitation-emission matrix (EEM) spectra (Fig. 5H, I), soluble microbial byproducts and humic acid-like organics presented in these effluents were easily removed with evidencing by the significant decrease of fluorescence intensity at detected peaks[70,71]. A moderate TOC removal (up to 50.5%) (Supplementary Fig. 44) was found after treating the collected effluents by FeOF/$H_2O_2$ system while the pH suffered a slightly increase to ~8.0 (Supplementary Fig. 45). Noting that the TOC removal is influenced by the specific content of organic pollutants in wastewater streams and could be facilely adjusted by increasing the input of $H_2O_2$. The results of tentative attempts highlighted that the rationally designed FeOF with unique fluorine coordination could reform the chemical-intensive water treatment process for broad applications with high feasibility.

## Discussion

In summary, we demonstrated a perspective for understanding the Fenton catalysis by the electronic structure modulation of iron sites with unique fluorine coordination. Sufficient evidence suggested that the coordinated fluorine atom featuring the most electronegativity could largely reduce the local electron density of iron sites, thereby facilitating $H_2O_2$ activation through lowering the energy barriers of the rate-limiting $H_2O_2$ adsorption step and subsequent O-O bond splitting reactions. As a result, the rationally designed FeOF exhibited an extraordinary Fenton activity for efficient •OH generation in terms of quantity (up to 450 μM) and selectivity (33.2%), significantly outperforming FeOCl analogy and the state-of-the-art Fenton catalysts ever recorded. Moreover, the enhanced interaction between iron sites and $H_2O_2$ molecules endowed FeOF with high pH-adaptability, thus breaking the limitations of harsh operation pH condition. The established FeOF/$H_2O_2$ system shows great practical potential for water purification as it is highly efficient to degrade various recalcitrant pollutants as well as to decontaminate different wastewaters. To balance the •OH production and utilization, further studies should be conducted to integrate the FeOF catalyst with the nanoconfined space for maximizing the practice potential. This work provides a significant insight into the electronic structure modulation of active sites with fluorine coordination at the atomic scale, which affords a rational design principle for developing advanced Fenton catalysts that effectively and selectively activate $H_2O_2$ into •OH radicals.

## Methods

### Materials

All chemicals and materials are analytically pure or higher and directly used without further purification (Supplementary Note 1). Ultrapure water from a Milli-Q system ($R > 18$ MΩ) was used for all experiments.

### Catalyst preparation

A two-step reaction was conducted to synthesize FeOF catalyst according to previous work[36] with modification. Typically, the precursor $FeSiF_6 \cdot 6H_2O$ was first prepared by adding iron powder (10.0 g) into fluorosilicic acid solution (100 mL) with vigorous stirring at 50 °C for 12 h. Note that the reaction should be carefully carried out at a Teflon breaker. The resultant solution was then suffered vacuum filtration to remove residual iron powder. After drying at 60 °C for 24 h, a blue crystalline solid of $FeSiF_6 \cdot 6H_2O$ was collected and restored for usage. In the second step, 1.0 g of the as-prepared $FeSiF_6 \cdot 6H_2O$ precursor was mixed with 150 mL of n-propanol stirred for 30 min. The mixture was then transferred into a 200 mL Teflon-lined autoclave and heated at 200 °C for 10 h to complete the solvothermal reaction. After being naturally cooled to room temperature, the product was centrifuged, washed with ethanol, and vacuum-dried at 80 °C for 24 h. The synthesis of FeOCl catalyst was referring to Sun et al.'s work[9] and detailed in Supplementary Note 1.

### Characterization methods

The crystal structures of the catalysts were assessed by powder XRD with a Thermo Scientific ARL-XTRA X-ray diffractometer. The microscopic morphologies were observed using a field-emission scanning electron microscope (SEM, Zeiss Ultra 55) and transmission electron microscope (TEM, TalosF200X). The HAADF-STEM images were recorded with an FEI Themis Z scanning transmission electron microscope with double aberration correctors in STEM mode (Super-X EDS). XPS (Thermo Scientific K-Alpha) was performed to analyze the surface elemental compositions and chemical states of the catalysts. The Lewis acid sites were analyzed by in situ pyridine-chemosorbed Fourier Transform-Infrared Spectroscopy (Py-IR, Thermo Scientific Nicolet iS10), and the measurement details were provided in Supplementary Note 2. The $^{57}Fe$ Mössbauer spectroscopy (Topologic 500 A) technique was performed to unravel the coordination environment of

iron sites in the catalysts before and after application under room temperature using $^{57}Co$ (Rh) λ-ray as irradiation source. The XAFS spectra of the Fe K-edge were recorded at the Singapore Synchrotron Radiation Light Source. The electrochemical performances were measured on a CHI 660E workstation (CH Instruments) with a three-electrode cell. The leaching concentrations of iron ions in Fenton reaction solutions were measured by ICP-MS (Agilent 720ES).

## Theoretical calculations

All theoretical calculations based on the DFT were performed using Vienna ab initio simulation package (VASP) within the generalized gradient approximation using the Perdew–Burke–Ernzerhof functional[72,73]. Detailed information is shown in Supplementary Note 3.

## Experiment procedures

All the experiments were carried out in a 100-ml beaker at $20 \pm 2\,°C$ using a shaker incubator. The initial pH values of the solutions were adjusted with 0.1 M HCl or 0.1 M NaOH. For Fenton activity evaluation, the accumulated •OH generation concentration of established Fenton systems with various catalysts $(0.1\,g\,L^{-1})$ was evaluated using coumarin (10 mM) as the selective probe. For pollutant degradation assessment, a certain amount of the catalyst was added to 50 mL 4-NP $(20\,mg\,L^{-1})$ and was dispersed uniformly using ultrasonication for 1 min. Then, a given volume of $H_2O_2$ was added to the suspension to initiate the Fenton reaction. At the specified time intervals, a 2.0-mL sample was withdrawn, filtered through a polyether sulfone membrane $(0.22\,\mu m)$, and immediately quenched using TBA before analysis. For the •OH scavenging experiment, MeOH (10 mM) and TBA (10 mM) were used to quench •OH (i.e., $•OH_f$ and $•OH_{surf}$) and $•OH_f$ radicals, respectively[74]. The cyclic experiments (with or without adding $H_2O_2$), Fe leaching experiments, coexisting ion (i.e., $Cl^-$) experiments, pH (i.e., 3.0, 4.0, 5.0, 6.0, 7.0, 8.0, and 9.0) experiments, $H_2O_2$ dosage (1.0, 5.0, 10.0, 15.0, and 20.0 mM) experiments, temperature experiments (i.e., 20, 30, and $40\,°C$), contaminants (i.e., tetracycline, atrazine, sulfamethoxazole, ofloxacin, and salicylic acid) experiments, and industrial effluents experiments were conducted to explore the practical considerations of pH-adaptive $FeOF/H_2O_2$ Fenton system. All the experiments were performed three times to ensure reproducibility.

## Analytic methods

The •OH radicals were quantified using a fluorescence method with coumarin as the probe, which could be converted into 7-hydroxycoumarin (7-HC) with 29% selectivity[59]. Further details for •OH measurement and quantification are provided in Supplementary Note 4. The concentrations of the contaminants were determined by high-performance liquid chromatography (Agilent 1260) with DAD detector and ZORBAX Eclipse XDB-C18 column $(4.6 \times 150\,mm, 3.5\,\mu m)$ at the corresponding detection wavelengths. The $H_2O_2$ concentration was determined by a titanium (IV) spectrophotometric method[75]. The presence of •OH was identified by an EPR spectrometer (Bruker A300) using DMPO as the trapping agent[65]. TOC was measured on an Elementar liquiTOC II Total Organic Carbon analyzer. The graphs with error bars are presented with standard deviation of the experiment in triplicate.

## Reporting summary

Further information on research design is available in the Nature Portfolio Reporting Summary linked to this article.

## Data availability

The data generated in this study are provided with in the article and the Supplementary Information file. Source data are provided with this paper.

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

## Acknowledgements

This work was supported by the National Natural Science Foundation of China (22106141, D.Y. and 22176124, J.L.), China Postdoctoral Science Foundation (2022M712827, D.Y.), Research Start-up Funding from Shanghai Jiao Tong University (WH220416002, J.L.), and Fundamental Research Funds of Zhejiang Sci-Tech University (23202131-Y, D.Y.).

## Author contributions

D.Y. and J.L. designed research; D.Y. and L.X. performed the experiments; D.Y., L.X., X.L., K.F., S.W., M.W., W.L., C.L., and J.L. analyzed the data; and D.Y., L.X., and J.L. wrote the paper. All authors discussed the results and commented on the manuscript. 1D.Y. and L.X. contributed equally to this work.

## Competing interests

The authors declare no competing interests.
