## [Peer Review File · Nature Communications]

REVIEWER COMMENTS

Reviewer #1 (Remarks to the Author):

In this manuscript, Luo et al. presented an investigation of synthesis of FeOF as a catalyst for Fenton-like reaction in degradation of water pollutants. In the work, they have comprehensively investigated the structure of FeOF as compared with FeOCl and the reaction with H₂O₂ for OH radical generation. The experimental tests and DFT calculations have suggested that the fluorine coordination in FeOF plays a pivotal role in lowering the local electron density and optimizing the electronic structures of iron sites, thus facilitating the rate-limiting H₂O₂ adsorption and subsequent peroxy bond cleavage reactions for OH radical generation. The work is useful and can be accepted for publication. Some other comments are listed for consideration for a revision.

1. Page 5 on Fig.S8, the IR peaks for FeOF changed significantly at different temperatures, not as claimed that they are nearly unchanged.
2. In Fenton-like reaction, Fe leaching and Fe ion's contributions to OH radical generation are important. The authors should check the Fe species and concentrations and report the homogeneous contribution.
3. It was suggested that Lewis acid sites are responsible for OH radical and the authors have checked the weak, medium and strong acids. They should discuss the roles of the acid sites for the reaction.
4. For the prepared catalysts, FeOF and FeOCl, they show significant difference in particle size and surface area, which should be considered for comparison of kinetic rate.
5. In Fig.3E, different materials for OH generations were presented. It was not clearly indicated if the data for other materials are from references or obtained in this work.
6. For those Figure showing test results, the reaction conditions should be presented in the captions.

Reviewer #2 (Remarks to the Author):

As numbers of reported studies focus on Fe-based heterogenous catalysts on Fenton catalysis, this manuscript reported an interesting and crucial work on developing the Fe-based Fenton catalysts by artfully utilizing the fluorine coordination to modulate the electronic structure of Fe sites for achieving ultra-efficient H₂O₂ activation. Specifically, this work takes a fresh perspective on original scientific questions of how the electronic structure of active sites governs the Fenton catalysis activity towards •OH generation. The authors conducted sufficient experimental tests and DFT calculations to clarify the pivotal role of fluorine coordination in H₂O₂ activation, which may inspire the design of efficient active sites. Intriguingly, the rational designed FeOF exhibited an unprecedented •OH yield with considerable selectivity, leading to excellent degradation activities against various organic pollutants at neutral condition, greatly solving the practical application issues for water treatment. Overall, the design strategy in this work is novel, the experiments were performed in high quality, the mechanism was

clearly uncovered, and the manuscript was well-written. Therefore, I would recommend its publication in Nature Communications after a minor revision according to the following comments:

1. From the point view of Fenton catalysis, the sustainable generation of •OH is the core field. The authors therefore firstly examined the Fenton catalysis activity of FeOF and FeOCl by qualitatively and quantitatively investigating the •OH generation with various advanced techniques. However, the specific experimental setup and details of measurements as well as calculation are not sufficient. It is suggested to add more descriptions to clarify the necessary details in Methods/Supplementary Information.
2. The iron species in both FeOF and FeOCl is important for H₂O₂ activation. Although it has been characterized by XPS, ⁵⁷Fe Mössbauer spectroscopy, and XAFS, the chemical environment difference of iron sites between FeOF and FeOCl needs more discussion because in Figure 1E, the binding energy of iron species seems to have an obvious shift.
3. For DFT calculations, the authors investigated the adsorption energy, charge transfer, band structure, and the free energies change for H₂O₂ activation. The evidences gained there were sufficient but indirect since the fate of H₂O₂ remained elusive. I wonder that if it's possible to study the evolution and fate of H₂O₂ on FeOF and FeOCl via ab initio molecular dynamics (MD) simulations. This can be another evidence to support the function of fluorine coordination.
4. The formation of •OH is generally accompanied by the generation of other ROS in Fenton systems. In this work, what's the proportion of •OH as an active species in pollutant degradation? In addition, the authors claimed that both surface-bounded •OH and free •OH contributed to pollutant degradation, but how to distinguish these two species should be highlighted.
5. Some formatting and grammar comments: Line 547 and 549, "i.e." needs to be italic; "Degradation rate" in Fig. S35, "Detailed information" in caption of Table S5, and "N.Z." in author contributions should be revised.

Reviewer #3 (Remarks to the Author):

This study presents an enhancement in Fenton catalysis achieved through the incorporation of fluoride, the most electronegative halogen, into the iron oxyhalide structure. The results clearly indicate that the distinct electron density polarization in FeOF leads to higher Fenton efficiency compared to FeOCl. However, certain aspects of the mechanisms, such as the regeneration of Fe(III), and the quantitative relationship between Fenton efficiency and local electron density, lack comprehensive explanations.

It is challenging to discern the individual contributions of facets and electronegativity to variations in Fenton efficiency. The primary facets of FeOCl and FeOF exhibit differences, with FeOCl predominantly exposing facets (010) and (110), while facets (211) and (101) are dominant for FeOF. Although the paper includes thermodynamic calculations for H₂O₂ activation on each facet, it is difficult to evaluate efficiency comprehensively when the relative abundance of each facet significantly influences the results.

As suggested in Line 364, if the dominant (010) facet of FeOCl indeed has a lower capacity for H₂O₂ activation, it becomes especially challenging to quantitatively relate electron density polarization to Fenton efficiency merely through a comparison of the as-synthesized FeOCl and FeOF.

Furthermore, as evidenced by Figure S3 and Figure S7, the physical characteristics of synthesized FeOF and FeOCl appear distinct. FeOF seems more particulate, while FeOCl has a planar structure. Therefore, although electronegativity of the halide component largely determines Fenton efficiency, directly attributing experimental results to electronegativity differences seems oversimplified.

Building upon the previous point, as the synthesis pathway for FeOF and FeOCl differ, FeOF through preconcentration of FeSiF₆·H₂O and FeOCl through simple partial pyrolysis of FeCl₆·6H₂O, how can the authors ensure their structural congruence?

The manuscript does provide evidence of reduced electron density at the iron center in FeOF, yet the link to the Fenton reaction remains somewhat missing. Especially, the redox cycle of the iron center during the Fenton reaction remains unclear. The authors describe the H₂O₂ activation pathway in Line 347-348 but do not elucidate the valence state or coordination environment of the iron active site during this reaction pathway. While the authors mention the formation of Fe(II) through XPS and Mössbauer measurements, this does not fully explain the reasons for the facile reversibility of the Fe(II)/Fe(III) cycle, which is necessary for high Fenton efficiency.

If ROS other than •OH were responsible for the degradation of 4-NP in the presence of •OH quenchers (e.g., MeOH and TBA), the formation pathway of these ROS needs to be elucidated.

Is the removal of 4-NP genuinely attributed to other ROS species, as suggested in Line 396-397, which states that 4-NP is recalcitrant towards other ROS species? It's worth noting that the concentration of •OH quenchers, 10 mM, appears low compared to the H₂O₂ concentration (10 mM).

Finally, it's important to investigate whether there is any leaching of iron and fluoride during the process.

Line 186 – 216: Correct “EXANS” to “XANES”

Line 412: “Similar to c generation inhibition”

Line 415: Correct “methane” to “methanol”

Reviewer #4 (Remarks to the Author):

In this study, the authors reported a case study on the fabrication of FeOF for efficient activation of H₂O₂ to degrade organic pollutants. The enhancement of F doping (over FeOCl) was systematically explored based on extensive characterization of the resultant catalyst, theoretical calculation of the involved chemical processes and states, as well as the preliminary evaluation in water treatment using 4-NP as the model target. Generally, I think the highlight of this study does not rely on the idea of how to fabricate the efficient catalyst, but the significant enhancement of Fenton-like catalysis over the well-known FOCl. Not only surprising to the authors but also to me, why no research has been carried out on the F replacement of Cl since such enhancement seems clearly expected to produce high-efficiency catalyst. Generally, it is a systematic work to advance Fenton-like system to a new level, however, I have still several issues on this work, which should be addressed before further consideration. (1) For heterogeneous Fenton system, except for the chemical structure of the solid catalyst, its size and morphology are also very important since they both are directly related to the effective contact of H₂O₂ to the active sites. Unluckily, the authors did not mention such important factors; (2) as for the utilization of Fenton system, fast production of hydroxyl radical is really crucial, however, the effective usage of the produced radicals in decontamination, particularly for organic degradation, is another side that should be particularly concerned. The authors should discuss how to balance the production process of radicals and its effective attack to the target compounds since the half life time of the radical is very short, and, if not used, they will be captured by the matrix composition. (3) As for the 4-NP degradation, the authors stated that part degradation is attributed to the other radicals (singlet oxygen and others, Line 417-418). It seems contradictory to the early description that NP is refractory to such radicals (L 397). Please check. (4) why 4-NP cannot be removed by FeOCl/H₂O₂? As limited by the experimental conditions? Need further clarification; (5) L 448. Self-enhanced? How to realize such effect? (6) L 415 methane or methanol? (7) I think the below comment is somewhat unfair since every world has multiple aspects, and comparison should lie on the fair basis. please check “For instance, the K⁺ intercalation into FeOCl alters the H₂O₂ activation pathway to generate Fe(IV)=O species rather than •OH radicals (30), which compromises the overall water treatment efficiency because of its high 78 selectivity toward organic substrate oxidation.”

Response to Referees

Manuscript ID: NCOMMS-23-45318

Title: Electronic Structure Modulation of Iron Sites with Fluorine Coordination Enables Ultra-effective H₂O₂ Activation

Authors: Deyou Yu, Licong Xu, Xia Liu, Kaixing Fu, Shanli Wang, Minghua Wu, Wangyang Lu, Chunyu Lv, Jinming Luo

General note: We sincerely thank all reviewers and the editor for their valuable comments and suggestions, which are certainly helpful in improving the quality of this manuscript. The review comments are presented in black *italic* font style. Responses to each comment are formatted with indentation and displayed in regular style. The “revised text as it appears in the manuscript” is presented in **blue** and indented. The reference numbers provided in this document are specific to the references cited herein and may not correspond with those used in the manuscript or Supplementary Information. This document contains the responses to the comments from the four reviewers.

Reviewer #1

General comment: *In this manuscript, Luo et al. presented an investigation of synthesis of FeOF as a catalyst for Fenton-like reaction in degradation of water pollutants. In the work, they have comprehensively investigated the structure of FeOF as compared with FeOCl and the reaction with H₂O₂ for OH radical generation. The experimental tests and DFT calculations have suggested that the fluorine coordination in FeOF plays a pivotal role in lowering the local electron density and optimizing the electronic structures of iron sites, thus facilitating the rate-limiting H₂O₂ adsorption and subsequent peroxy bond cleavage reactions for OH radical generation. The work is useful and can be accepted for publication. Some other comments are listed for consideration for a revision.*

Response: We express our gratitude to the reviewer for acknowledging the research value and providing constructive comments to enhance the quality of this work. We have answered the reviewer's comments point by point and rectified any relevant issues. We would greatly appreciate it if the reviewer could reassess our revised manuscript for potential publication in the journal.

Comment 1: *Page 5 on Fig.S8, the IR peaks for FeOF changed significantly at different temperatures, not as claimed that they are nearly unchanged.*

Response: We appreciate your valuable comment. We rechecked and reanalyzed the result of Py-IR spectra of FeOF in Fig. S8. Our findings indicated that the intensity of characteristic peaks suffered a moderate decrease while their shape remained almost unchanged. On the contrary, the corresponding peak representing strong Lewis acid sites for FeOCl disappeared at a degassing temperature of 200 °C. We attributed this observation to the difference amounts of the strong Lewis acid sites between FeOF and FeOCl, which might be derived from the fluoride coordination.

To clarify this result, we have revised the manuscript description as follows.

Line 158-160, “In addition, as degassing temperature increased, the band intensity in the Py-IR spectra of FeOF suffered a moderate decrease while a dramatic decline occurred for FeOCl.”

Comment 2: *In Fenton-like reaction, Fe leaching and Fe ion's contributions to OH radical generation are important. The authors should check the Fe species and concentrations and report the homogeneous contribution.*

Response: We thank the reviewer for such insightful suggestion. We conducted additional experiments to evaluate the iron leaching and the contribution of iron ions. Initially, by applying ICP-OES measurements, the total iron leaching of FeOF/H₂O₂ Fenton system determined as 0.52 mg L⁻¹ at 30 min. Subsequently, when testing the Fenton activity of individual Fe³⁺ with considerable higher dosage (10.00 mg L⁻¹) than the leaching level, we observed that almost no

4-NP degradation occurred in the reaction (**Figure R1**). To comprehensively understand the homogeneous contribution, we further identified the species of leached iron ions via a spectrometric method using 1,10-phenanthroline as the probe according to a previous report (1). It was found that the Fe leaching concentration remained relative constant at a range of 0.20~0.40 mg L⁻¹ during reaction (**Figure R2**). As the reaction prolonged, the proportion of Fe²⁺ ions increased from 3% to 42%, indicating an enhancement of Fe(II)/Fe(III) redox cycle. In this context, we examined the •OH generation ability and 4-NP degradation performance of individual Fe³⁺ (0.15 mg L⁻¹) and Fe²⁺ (0.15 and 0.30 mg L⁻¹) ions, as well as their mixture at similar concentrations. As illustrated in **Figure R3**, marginal •OH (~0.1 μM) was generated, and little 4-NP was destructed when employing iron ions as the catalyst. Therefore, we excluded the homogeneous contribution of iron ions to the Fenton reaction.

To clarify this result, we have revised the manuscript description as follows.

Line 277-280, “We also excluded the homogeneous contribution of leached iron ions because scarce •OH was generated when using Fe²⁺, Fe³⁺, and Fe²⁺/Fe³⁺ mixture as the catalyst at equivalent concentrations to the leaching level (Fig. S13).”

Line 430-431, “However, 4-NP can be hardly removed by individual H₂O₂ (Fig. S27), FeOF alone (Fig. S28), or leaching iron ions (Fig. S29) at neutral condition.”

Figure R1. H₂O₂ activation performance of free Fe³⁺ ions with much higher concentration than the leachate of FeOF/H₂O₂ system. Reaction conditions: [H₂O₂] = 10 mM, [Fe³⁺] = 10 mg L⁻¹, [4-NP] = 20 mg L⁻¹, pH = 7.0, Temperature = 20 °C.

Figure R2. Concentration and proportion of leached Fe³⁺ and Fe²⁺ ions in the FeOF/H₂O₂ system.

Figure R3. Accumulated •OH generation and 4-NP degradation by H₂O₂ activation using Fe²⁺ or/and Fe³⁺ ions as the catalyst. Reaction conditions: [H₂O₂] = 10 mM, [Coumarin] = 10 mM (if used), [4-NP] = 20 mg L⁻¹ (if used), pH = 7.0, Temperature = 20 °C.

References:

1. Harvey, A. E.; Smart, J.; Amis, E. S., Simultaneous Spectrophotometric Determination of Iron(II) and Total Iron with 1,10-Phenanthroline. Anal. Chem. 1955, 27, 26-29.

Comment 3: It was suggested that Lewis acid sites are responsible for OH radical and the authors have checked the weak, medium and strong acids. They should discuss the roles of the acid sites for the reaction.

Response: We appreciate your valuable comment. In our study, the $\bullet\text{OH}$ is generated through the interaction between H_2O_2 molecules and iron sites coordinated with fluorine atoms, which perform as typical Lewis acid sites. It has been reported that the such Lewis acid-base interaction is recognized as the initial step and is of significant importance for H_2O_2 activation (1). Sites possessing stronger Lewis acidity demonstrate greater ability to activate H_2O_2 molecules by enhancing the H_2O_2 adsorption and electron transfer. Accordingly, we quantified the amount of active iron sites by calculating the Lewis acidity based on the Py-IR spectra. We observed that the proportion of medium and strong LASs in FeOF was much higher than that in FeOCl (**Figure 1F**). This means that the fluorine coordination enhances the Lewis acidity of iron sites, which in turns promotes the H_2O_2 decomposition for producing more $\bullet\text{OH}$. To elucidate the role of acid sites for the reaction, we investigated the effect of phosphate, a strong Lewis base for blocking interaction between iron sites and H_2O_2 molecules (2), on $\bullet\text{OH}$ generation. **Figure R4** demonstrates that the phosphate addition completely suppresses the $\bullet\text{OH}$ production, highlighting the unique catalysis role of Lewis acid sites in H_2O_2 activation. To give a more direct comparison, we evaluated the specific rate constant for $\bullet\text{OH}$ generation normalized by the amount of Lewis acid sites. The Fenton activity of FeOF outperformed FeOCl with 60.7-fold, 61.8-fold, and 5.2-fold increments in the specific rate constant normalized by the amounts of total, medium, and strong Lewis acid sites, respectively (**Table R1**). Meanwhile, we also assessed the effect of phosphate on 4-NP degradation by FeOF/ H_2O_2 system, where almost no removal of 4-NP was observed (**Figure R5**). Therefore, the role of Lewis acid sites is to interact with the H_2O_2 molecules to generate $\bullet\text{OH}$, in which the sites with different acid strengths exhibit distinct performances.

To clarify this result, we have revised the manuscript description as follows.

Line 305-312, “The Lewis acid-base interaction between catalysts and H_2O_2 is recognized as the initial step and is of significant importance for H_2O_2 activation (44). To clarify the role of Lewis acid sites (*i.e.*, iron sites) within FeOF catalyst on H_2O_2 activation, we assessed the effect of phosphate, a strong Lewis base for blocking interaction between iron sites and H_2O_2 molecules (52), on $\bullet\text{OH}$ generation. As shown in Fig. S14A, the phosphate addition completely suppresses the $\bullet\text{OH}$ production confirmed by the no detection of 7-hydroxycoumarin (Fig. S14B), demonstrating the unique catalysis role of iron sites for activating H_2O_2 into $\bullet\text{OH}$.”

Line 437-442, “Considering the possibility that empty orbitals of unsaturated iron centers in FeOF play a decisive role in H₂O₂ activation, we also attempted to block the orbital via a Lewis acid-base interaction using phosphate. Similar to •OH generation inhibition, we observed a strong suppression of 4-NP decay with almost zero percent degradation rate (Fig. 5C), suggesting that unsaturated iron centers (*e.g.*, F-(Fe(III)O₃)-F moiety) is responsible for 4-NP destruction.”

Figure R4. Effect of Na₃PO₄ on •OH generation during H₂O₂ activation by FeOF. (A) Accumulated •OH generation versus reaction time and (B) fluorescence spectra of 7-HC in the FeOF/H₂O₂ system with and without Na₃PO₄ addition. Reaction conditions: [H₂O₂] = 10 mM, [catalyst] = 0.1 g L⁻¹, [Coumarin] = 10 mM, [Na₃PO₄] = 2.0 g L⁻¹ (if used), pH = 7.0, Temperature = 20 °C.

Figure R5. Quenching or inhibition effect of TBA, methanol, and Na_3PO_4 on the 4-NP degradation efficiency by FeOF/ H_2O_2 system.

Table R1. Summary of apparent rate constants for $\bullet\text{OH}$ generation by FeOF and FeOCl with and without normalized by specific surface area and the amount of Lewis acid sites.

Catalyst	Apparent rate constant for $\bullet\text{OH}$ generation, $K_{(\bullet\text{OH})}$ ($\mu\text{mol L}^{-1} \text{min}^{-1}$)	Apparent rate constant for $\bullet\text{OH}$ generation normalized by specific surface area, K_s ($\mu\text{mol L}^{-1} \text{min}^{-1}$)	Apparent rate constant for $\bullet\text{OH}$ generation normalized by the amount of Lewis acid sites			
			Total	Weak	Medium	Strong
FeOF	2.94	31.7	194.2	-	346.1	442.3
FeOCl	0.017	0.371	3.2	7.8	5.6	85.9

References:

1. Mo, F.; Song, C.; Zhou, Q.; Xue, W.; Ouyang, S.; Wang, Q.; Hou, Z.; Wang, S.; Wang, J., The optimized Fenton-like activity of Fe single-atom sites by Fe atomic clusters-mediated electronic configuration modulation. *PNAS* **2023**, *120*, (15), e2300281120
2. Gao, C.; Chen, S.; Quan, X.; Yu, H.; Zhang, Y., Enhanced Fenton-like catalysis by iron-based metal organic frameworks for degradation of organic pollutants. *J. Catal.* **2017**, *356*, 125-132.

Comment 4: *For the prepared catalysts, FeOF and FeOCl, they show significant difference in particle size and surface area, which should be considered for comparison of kinetic rate.*

Response: We express our gratitude to the reviewer for this constructive suggestion. In the view of heterogeneous catalysis, both the particle size and surface area significantly affect the activities of catalysts through a combined protocol involving mass-transfer and site-exposure. Recent studies have demonstrated that the decrease of particle size generally increased the exposed surface area of catalysts' active sites, thereby strengthening the contact between reactants and catalyst substantially (1, 2). It has been also reported that downsizing the catalysts' particle presents an efficient strategy to maximize active site density, improve the sites accessibility, and enhance catalytic performance (3-5).

In our study, the spherical FeOF particles (~50 nm) exhibit a much smaller size than the rectangular FeOCl (~500 × 1000 nm) while the specific surface area of FeOF shows 2-fold increment compared to FeOCl. Inspired by the reviewer's comment, these observations suggest that the effect of particle size and surface area should be considered. Although we do not have a unified theory that can explain and predict the behavior of catalysts with different particle sizes for reactions such as Fenton catalysis (6), we alternatively calculated the surface area normalized kinetic rate constant for •OH generation to gain a comprehensive comparison as the effect of particle size could be explained in a manner of surface area. We observed that the FeOF exhibited a value of 31.7 μmol g L⁻¹ min⁻¹ m⁻², which was 85.4 times higher than the FeOCl (**Table R2**). Such increment is close to the observed 168.0-fold enhancement in •OH generation, revealing that the smaller particle size and higher surface area are conducive to Fenton reaction but not the primary factors.

To clarify this result, we have revised the manuscript description as follows.

Line 143-145, “By comparison to FeOCl (Fig. S7), the smaller particle size and higher surface area of FeOF are conducive to Fenton reaction by increasing the density and accessibility of active iron sites.”

Line 280-286, “The surface area normalized kinetic rate constant for •OH generation by FeOF was determined to be 31.7 μmol g L⁻¹ min⁻¹ m⁻², 85.4 times higher than by FeOCl (Table S4). Such increment closing to the •OH generation enhancement reveals that the smaller particle size

and higher surface area are not the primary factors. Consequently, we deduced that the disparity of H₂O₂ activation activity could be ascribed to the presence of F-Fe-O sites, where the iron atom is expected to show much lower electron density over Cl-Fe-O.”

Table R2. Summary of apparent rate constants for •OH generation by FeOF and FeOCl with and without normalized by specific surface area and the amount of Lewis acid sites.

Catalyst	Apparent rate constant for •OH generation, K _(•OH) (μmol L ⁻¹ min ⁻¹)	Apparent rate constant for •OH generation normalized by specific surface area, K _s (μmol L ⁻¹ min ⁻¹)	Apparent rate constant for •OH generation normalized by the amount of Lewis acid sites			
			Total	Weak	Medium	Strong
FeOF	2.94	31.7	194.2	-	346.1	442.3
FeOCl	0.017	0.371	3.2	7.8	5.6	85.9

References:

1. Polshettiwar, V.; Varma, R. S., Green chemistry by nano-catalysis. *Green Chem.* **2010**, *12*, (5), 743-754.
2. Tang, Z.; Zhao, P.; Wang, H.; Liu, Y.; Bu, W., Biomedicine Meets Fenton Chemistry. *Chem. Rev.* **2021**, *121*, (4), 1981-2019.
3. Zhao, J.-W.; Wang, H.-Y.; Feng, L.; Zhu, J.-Z.; Liu, J.-X.; Li, W.-X., Crystal-Phase Engineering in Heterogeneous Catalysis. *Chem. Rev.* **2023**, doi: <https://doi.org/10.1021/acs.chemrev.3c00402>.
4. Kim, J. H.; Yoon, S.; Baek, D. S.; Kim, J.; Kim, J.; An, K.; Joo, S. H., Boosting Thermal Stability of Volatile Os Catalysts by Downsizing to Atomically Dispersed Species. *Jacs Au* **2022**, *2*, (8), 1811-1817.
5. Li, Z.; Ji, S.; Liu, Y.; Cao, X.; Tian, S.; Chen, Y.; Niu, Z.; Li, Y., Well-Defined Materials for Heterogeneous Catalysis: From Nanoparticles to Isolated Single-Atom Sites. *Chem. Rev.* **2020**, *120*, (2), 623-682.
6. Liu, L.; Corma, A., Metal Catalysts for Heterogeneous Catalysis: From Single Atoms to Nanoclusters and Nanoparticles. *Chem. Rev.* **2018**, *118*, (10), 4981-5079.

Comment 5: *In Fig.3E, different materials for OH generations were presented. It was not clearly indicated if the data for other materials are from references or obtained in this work.*

Response: Thanks for the careful suggestion. The data for other materials in Fig. 3E was collected from references. To clarify this, we have revised the manuscript description as follows.

Line 801-803, “(E) Comparison of the accumulated •OH generation by the state-of-the-art Fenton catalysts demonstrated by previous studies. The relevant references are listed in Table S5, Supplementary Information.”

Comment 6: *For those Figure showing test results, the reaction conditions should be presented in the captions.*

Response: We appreciate your valuable comment. We have added the reaction conditions in the captions of Fig. 3, Fig.5, Fig. S11, Fig. S12, Fig. S14, Fig. S15, Fig. S26, Fig. S27, Fig. S29, Fig. S30, Fig. S31, Fig. S32, Fig. S34, Fig. S35, Fig. S36, Fig. S40, Fig. S43, and Fig. S44.

To clarify these details, we have revised the manuscript description as follows.

Line 798-806, “**Fig. 3.** Fenton catalysis performance of FeOF and FeOCl catalysts. (A) DMPO-•OH spectra, (B) fluorescence intensity versus reaction time, and (C) accumulated •OH generation for FeOF and FeOCl systems. (D) Accumulated •OH generation at different pH values for FeOF system. (E) Comparison of the accumulated •OH generation over the state-of-the-art Fenton catalysts demonstrated by previous studies. The relevant references are listed in Table S5, Supplementary Information. (F) Accumulated •OH generation at various H₂O₂ inputs for FeOF and FeOCl systems. Reaction conditions: [H₂O₂] = 10 mM (unless otherwise specified), [catalyst] = 0.1 g L⁻¹, [Coumarin] = [MeOH] = [TBA] = 10 mM, pH = 7.0 (unless otherwise specified), Temperature = 20 °C.”

Line 820-834, “**Fig. 5.** Application potential of FeOF/H₂O₂ system for water purification. (A) 4-NP degradation kinetics in the neutral FeOF/H₂O₂ and FeOCl/H₂O₂ systems. (B) Comparison of the observed rate constant of 4-NP degradation by FeOF, FeOCl and ferrihydrite catalysts. (C) Quenching or inhibition effect of TBA, methanol, and Na₃PO₄ on the 4-NP degradation

efficiency by FeOF. (D) Contribution of $\bullet\text{OH}$ (*i.e.*, $\bullet\text{OH}_f$ and $\bullet\text{OH}_{\text{surf}}$) and others on the 4-NP degradation. (E) Influence of pH values on 4-NP degradation in the FeOF/H₂O₂ system. (F) 4-NP degradation in eight successive cycles in the neutral FeOF/H₂O₂ system with and without adding H₂O₂ after each running. Note that the 4-NP degradation efficiency could be recovered by adding H₂O₂ in the 9th running. (G) Degradation of various organic pollutants (*i.e.*, tetracycline, atrazine, sulfamethoxazole, ofloxacin, and salicylic acid) by neutral FeOF/H₂O₂ system. EEM spectra of collected dyeing & printing wastewater before (H) and after treatment by neutral FeOF/H₂O₂ system. Reaction conditions: [H₂O₂] = 10 mM, [catalyst] = 0.1 g L⁻¹, [Pollutant] = 20 mg L⁻¹, [MeOH] = [TBA] = 10 mM (if used), [Na₃PO₄] = 2.0 g L⁻¹ (if used), pH = 7.0, Temperature = 20 °C.”

Page 15 in Supplementary Information, “Fig. S11. DMPO- $\bullet\text{OH}$ signals in EPR spectra of FeOF/H₂O₂ system with and without TBA addition. Reaction conditions: [H₂O₂] = 10 mM, [catalyst] = 1.0 g L⁻¹, [DMPO] = 10 mM, [TBA] = 10 mM (if used), pH = 7.0, Temperature = 20 °C.”

Page 16 in Supplementary Information, “Fig. S12. DMPO- $\bullet\text{OH}$ signals in EPR spectra of FeOF/H₂O₂ system at various pH values. Reaction conditions: [H₂O₂] = 10 mM, [catalyst] = 1.0 g L⁻¹, [DMPO] = 10 mM, pH = 7.0, Temperature = 20 °C.”

Page 18 in Supplementary Information, “Fig. S14. (A) Accumulated $\bullet\text{OH}$ generation versus reaction time in neutral ferrihydrite and conventional homogeneous Fenton systems. (B) Comparison of $\bullet\text{OH}$ generation in the FeOF, FeOCl, ferrihydrite and Fe²⁺ systems. Reaction conditions: [H₂O₂] = 10 mM, [catalyst] (except Fe²⁺) = 0.1 g L⁻¹, [Fe²⁺] = 1.10 mM, [Coumarin] = 10 mM, pH = 7.0, Temperature = 20 °C.”

Page 19 in Supplementary Information, “Fig. S15. Effect of Na₃PO₄ on $\bullet\text{OH}$ generation during H₂O₂ activation by FeOF. (A) Accumulated $\bullet\text{OH}$ generation versus reaction time and (B) fluorescence spectra of 7-HC in the FeOF/H₂O₂ system with and without Na₃PO₄ addition. Reaction conditions: [H₂O₂] = 10 mM, [catalyst] = 0.1 g L⁻¹, [Coumarin] = 10 mM, [Na₃PO₄] = 2.0 g L⁻¹ (if used), pH = 7.0, Temperature = 20 °C.”

Page 30 in Supplementary Information, “Fig. S26. Effects of pH value on TOC removal by the FeOF/H₂O₂ system. Reaction conditions: [H₂O₂] = 10 mM, [catalyst] = 0.1 g L⁻¹, [4-NP] = 20 mg L⁻¹, pH = 7.0, Temperature = 20 °C.”

Page 31 in Supplementary Information, “Fig. S27. Degradation rate of 4-NP by individual H₂O₂ or FeOF system. Reaction conditions: [H₂O₂] = 10 mM (if used), [catalyst] = 0.1 g L⁻¹ (if used), [4-NP] = 20 mg L⁻¹, pH = 7.0, Temperature = 20 °C.”

Page 32 in Supplementary Information, “Fig. S28. H₂O₂ activation performance of free Fe³⁺ ions with much higher concentration than the leachate of FeOF/H₂O₂ system. Reaction conditions: [H₂O₂] = 10 mM, [Fe³⁺] = 10 mg L⁻¹, [4-NP] = 20 mg L⁻¹, pH = 7.0, Temperature = 20 °C.”

Page 33 in Supplementary Information, “Fig. S29. Accumulated •OH generation and 4-NP degradation by H₂O₂ activation using Fe²⁺ or/and Fe³⁺ ions as the catalyst. Reaction conditions: [H₂O₂] = 10 mM, [Coumarin] = 10 mM (if used), [4-NP] = 20 mg L⁻¹ (if used), pH = 7.0, Temperature = 20 °C.”

Page 34 in Supplementary Information, “Fig. S30. Degradation of 4-NP by ferrihydrite/H₂O₂ and conventional homogeneous Fe(II) systems. Reaction conditions: [H₂O₂] = 10 mM, [catalyst] = 0.1 g L⁻¹, [Fe²⁺] = 1.10 mM, [4-NP] = 20 mg L⁻¹, pH = 7.0, Temperature = 20 °C.”

Page 35 in Supplementary Information, “Fig. S31. Concentration of fluoride leaching during Fenton reaction with FeOF as the catalyst. Reaction conditions: [H₂O₂] = 10 mM, [catalyst] = 0.1 g L⁻¹, [4-NP] = 20 mg L⁻¹, pH = 7.0, Temperature = 20 °C.”

Page 36 in Supplementary Information, “Fig. S32. Quenching effect of high concentration TBA and methanol on the 4-NP degradation efficiency by FeOF. Reaction conditions: [H₂O₂] = 10 mM, [catalyst] = 0.1 g L⁻¹, [4-NP] = 20 mg L⁻¹, [MeOH] = [TBA] = 200 mM, pH = 7.0, Temperature = 20 °C.”

Page 38 in Supplementary Information, “Fig. S34. Effects of reaction temperature on 4-NP degradation. Reaction conditions: [H₂O₂] = 10 mM, [catalyst] = 0.1 g L⁻¹, [4-NP] = 20 mg L⁻¹, pH = 7.0.”

Page 39 in Supplementary Information, “Fig. S35. Effects of H₂O₂ dosage on 4-NP degradation. Reaction conditions: [catalyst] = 0.1 g L⁻¹, [4-NP] = 20 mg L⁻¹, pH = 7.0, Temperature = 20 °C.”

Page 40 in Supplementary Information, “Fig. S36. Successive recycle tests with the addition of both 4-NP and H₂O₂ after each cycle. Reaction conditions: [H₂O₂] = 10 mM, [catalyst] = 0.1 g L⁻¹, [4-NP] = 20 mg L⁻¹, pH = 7.0, Temperature = 20 °C.”

Page 44 in Supplementary Information, “Fig. S40. Effects of NaCl dosage on 4-NP degradation. Reaction conditions: [H₂O₂] = 10 mM, [catalyst] = 0.1 g L⁻¹, [4-NP] = 20 mg L⁻¹, [NaCl] = 0~5.0 g L⁻¹, pH = 7.0, Temperature = 20 °C.”

Page 47 in Supplementary Information, “Fig. S43. Bacterial inactivation performance of the FeOF/H₂O₂ system. Reaction conditions: [H₂O₂] = 10 mM, [catalyst] = 0.1 g L⁻¹, [*E. coli*] = 10⁷ CFU mL⁻¹, pH = 7.0, Temperature = 20 °C.”

Page 48 in Supplementary Information, “Fig. S44. TOC value change of collected wastewaters before and after treatment by FeOF/H₂O₂ system. Reaction conditions: [H₂O₂] = 10 mM, [catalyst] = 0.1 g L⁻¹, pH = 7.0, Temperature = 20 °C.”

Reviewer #2

General comment: *As numbers of reported studies focus on Fe-based heterogenous catalysts on Fenton catalysis, this manuscript reported an interesting and crucial work on developing the Fe-based Fenton catalysts by artfully utilizing the fluorine coordination to modulate the electronic structure of Fe sites for achieving ultra-efficient H₂O₂ activation. Specifically, this work takes a fresh perspective on original scientific questions of how the electronic structure of active sites governs the Fenton catalysis activity towards •OH generation. The authors conducted sufficient experimental tests and DFT calculations to clarify the pivotal role of fluorine coordination in H₂O₂ activation, which may inspire the design of efficient active sites. Intriguingly, the rational designed FeOF exhibited an unprecedented •OH yield with considerable selectivity, leading to excellent degradation activities against various organic pollutants at neutral condition, greatly solving the practical application issues for water treatment. Overall, the design strategy in this work is novel, the experiments were performed in high quality, the mechanism was clearly uncovered, and the manuscript was well-written. Therefore, I would recommend its publication in Nature Communications after a minor revision according to the following comments.*

Response: We express our gratitude to the reviewer for acknowledging the research value and providing constructive comments to enhance the quality of this work. We have answered the reviewer's comments point by point and rectified any relevant issues. We would greatly appreciate it if the reviewer could reassess our revised manuscript for potential publication in the journal.

Comment 1: *From the point view of Fenton catalysis, the sustainable generation of •OH is the core field. The authors therefore firstly examined the Fenton catalysis activity of FeOF and FeOCl by qualitatively and quantitatively investigating the •OH generation with various advanced techniques. However, the specific experimental setup and details of measurements as well as calculation are not sufficient. It is suggested to add more descriptions to clarify the necessary details in Methods/Supplementary Information.*

Response: We appreciate your valuable suggestion. In our study, we assessed the Fenton catalysis activity by the •OH generation that reflects the valuable and sustainable H₂O₂

activation, which is the core for Fenton reaction. To this end, the measurement and quantification of •OH are significantly important for clearly elucidating the performance. The accumulated concentration of •OH was determined by a fluorescence method with coumarin as the probe according to previous reports (1, 2). We first plotted the standard curve of 7-hydroxyl coumarin concentration versus fluorescence intensity. Then, we measured the intensity of withdrawn samples, whose 7-hydroxyl coumarin (7-HOC) concentration could be easily determined with a fluorescence spectrometer. Finally, we calculated the accumulated concentration of •OH using the following equation.

$$[\bullet\text{OH}] = 2 \times [\text{7-HOC}] / \text{Se}$$

where [7-HOC] and Se indicate the concentration of 7-HOC and the selectivity, respectively.

Therefore, the specific experimental setup and details of measurements as well as the calculations are rephrased and added.

To clarify this information, we have revised the manuscript description as follows.

Line 591-594, “The •OH radicals were quantified using a fluorescence method with coumarin as the probe, which could be converted into 7-hydroxycoumarin (7-HC) with 29% selectivity (48). Further details for •OH measurement and quantification are provided in Text S3.”

Pages 2-3 in Supplementary Information, “We first plotted the standard curve of 7-hydroxyl coumarin concentration versus fluorescence intensity. The resultant equation is determined to be “Intensity = 37.7 × [7-HOC] (R² > 0.999)”. Then, we measured the intensity of withdrawn samples, whose 7-hydroxyl coumarin (7-HOC) concentration could be easily determined by the obtained equation. Finally, we calculated the accumulated concentration of •OH by following equation.

$$[\bullet\text{OH}] = 2 \times [\text{7-HOC}] / \text{Se}$$

where [7-HOC] and Se indicate the concentration of 7-HOC and the selectivity, respectively.”

References:

1. Tokumura, M.; Morito, R.; Hatayama, R.; Kawase, Y., Iron redox cycling in hydroxyl radical generation during the photo-Fenton oxidative degradation: Dynamic change of

hydroxyl radical concentration. *Applied Catalysis B-Environmental* **2011**, *106*, (3-4), 565-576.

2. De-Nasri, S. J.; Nagarajan, S.; Robertson, P. K. J.; Ranade, V. V., Quantification of Hydroxyl Radicals in Photocatalysis and Acoustic Cavitation: Utility of Coumarin as a Chemical Probe. *Chem. Eng. J.* **2020**, 127560.

Comment 2: *The iron species in both FeOF and FeOCl is important for H₂O₂ activation. Although it has been characterized by XPS, ⁵⁷Fe Mössbauer spectroscopy, and XAFS, the chemical environment difference of iron sites between FeOF and FeOCl needs more discussion because in Figure 1E, the binding energy of iron species seems to have an obvious shift.*

Response: We thank the reviewer for such insightful suggestion. We reanalyzed and compared the binding energy of iron species of FeOF and FeOCl in the XPS spectra (**Figure R6**). Interestingly, we observed that the peaks referring to iron species in FeOF suffered an obvious shift to a higher binding energy as compared to those in FeOCl. It has been reported that the electron density around iron species become relatively lower as the binding energy shifts to a higher value (1-3). In this context, we thereby attributed this shift to the fluorine coordination that allows for withdrawing more electrons of iron species to fluorine atoms, which improves the interaction between H₂O₂ molecules and iron sites.

To clarify this result, we have revised the manuscript description as follows.

Line 137-140, “The peaks referring to iron species in FeOF suffered an obvious shift to a higher binding energy against those in FeOCl, suggesting the electron density around iron sites was largely reduced by fluorine coordination, which would improve the interaction between H₂O₂ molecules and iron sites (40-42).”

Figure R6. High-resolution Fe 2p XPS spectra of FeOF and FeOCl catalysts

References:

1. Zarate, X.; Schott, E.; Arratia-Perez, R., Effects of the peripheral substituents (-NH₂, -OH, -CH₃, -H, -C₆H₅, -Cl, -CO₂H and -NO₂) on molecular properties of a Ni-Porphyrine dimers family. *Polyhedron* **2013**, *50*, (1), 131-138.
2. Liang, H.; Liu, R.; Hu, C.; An, X.; Zhang, X.; Liu, H.; Qu, J., Synergistic effect of dual sites on bimetal-organic frameworks for highly efficient peroxide activation. *J. Hazard. Mater.* **2021**, *406*.
3. Li, H.; Hu, T.; Du, N.; Zhang, R.; Liu, J.; Hou, W., Wavelength-dependent differences in photocatalytic performance between BiOBr nanosheets with dominant exposed (001) and (010) facets. *Applied Catalysis B-Environmental* **2016**, *187*, 342-349.

Comment 3: For DFT calculations, the authors investigated the adsorption energy, charge transfer, band structure, and the free energies change for H₂O₂ activation. The evidences gained there were sufficient but indirect since the fate of H₂O₂ remained elusive. I wonder that if it's possible to study the evolution and fate of H₂O₂ on FeOF and FeOCl via *ab initio* molecular dynamics (MD) simulations. This can be another evidence to support the function of fluorine coordination.

Response: We appreciate your valuable suggestion. We conducted *ab initio* molecular dynamics (MD) simulations to investigate the behavior and fate of H₂O₂ on both FeOF and FeOCl. An initial structure consisting of one H₂O₂ molecule and an exposed (211) facet of the FeOF or FeOCl was applied for MD simulation. The results of MD simulation including the structural snapshots of the reaction and the fluctuations of electronic energy are displayed in **Figure R7**. The adsorbed H₂O₂ on FeOF could be readily cleaved into two •OH radicals without energy barriers to overcome. In contrast, the adsorbed H₂O₂ on FeOCl cannot split into •OH radicals. The divergence in H₂O₂ split performance between FeOF and FeOCl further highlights the unique function of fluorine coordination.

To clarify this result, we have revised the manuscript description as follows.

Line 390-395, “We conducted *ab initio* molecular dynamics (MD) simulations to further unveil the evolution and fate of H₂O₂, detailed in Fig. S22. H₂O₂ adsorbed on FeOF could be readily cleaved into two •OH radicals without energy barriers to overcome. Conversely, the adsorbed H₂O₂ on FeOCl cannot split into •OH radicals. The divergence in H₂O₂ split performance between FeOF and FeOCl further highlights the unique function of fluorine coordination.”

Figure R7. Evolution of temperature and energy during the *ab initio* molecular dynamics (MD) simulation for H₂O₂ activation onto FeOCl (A) and FeOF (B). The brown, red, green blue, and white balls denote Fe, O, C, Cl, F, and H atoms, respectively.

Comment 4: *The formation of •OH is generally accompanied by the generation of other ROS in Fenton systems. In this work, what’s the proportion of •OH as an active species in pollutant*

degradation? In addition, the authors claimed that both surface-bounded •OH and free •OH contributed to pollutant degradation, but how to distinguish these two species should be highlighted.

Response: We express our gratitude to the reviewer for this comment. To identify whether other ROS species were simultaneously formed or not, we performed additional ERP measurements to monitor the presence of singlet oxygen and superoxide radicals. We found that there were no characteristic signals in the EPR spectra for FeOF/H₂O₂ system (**Figure R8**). This suggests that neither singlet oxygen nor superoxide radicals were generated and responsible for pollutant degradation. Besides, we also found that single H₂O₂ or leaching iron failed to oxidize/remove 4-NP while individual FeOF could eliminate ~1.0% of 4-NP via adsorption (**Figure R9**). Meanwhile, inspired by Reviewer #3's Comment 5, we increased the dosage of •OH quenchers to 200 mM, much greater than the H₂O₂ input, to evaluate their inhibition effects (**Figure R10**). The 4-NP removal also reduced to ~1.0% with the addition of high dosage quencher. Collectively, the proportion of •OH as an active species in pollutant degradation was determined to be about 99.0%.

Previous study proposed that MeOH and TBA can scavenge •OH (*i.e.*, •OH_f and •OH_{surf}) and •OH_f radicals at a relative lower concentration, respectively (1). Therefore, we assessed the contribution of surface-bounded •OH and free •OH by quenching experiments with 10 mM of MeOH and TBA. As shown in **Figure R11**, the 4-NP decomposition rate showed an evident decline from 95.4% to 28.1% in the presence of 10 mM TBA. The equal addition of MeOH retarded 70.6% of 4-NP being degraded by FeOF/H₂O₂ system, which was about 2.3% higher than TBA inhibition, highlighting the slight contribution of surface-bound •OH radicals for 4-NP degradation. The contribution of free and surface-bound •OH radicals were quantified and calculated by following equations according to previous study using the k_{obs} before and after quenching (**Figure R12**) (2).

$$\theta(\bullet \text{OH}) = [(k_{\text{obs}}^0 - k_{\text{obs}}^1)/k_{\text{obs}}^0] \times 100\%$$

$$\theta(\bullet \text{OH}_{\text{surf}}) = [(k_{\text{obs}}^0 - k_{\text{obs}}^2)/k_{\text{obs}}^0] \times 100\%$$

$$\theta(\bullet \text{OH}_f) = \theta(\bullet \text{OH}) - \theta(\bullet \text{OH}_{\text{surf}})$$

where the $\theta(\bullet \text{OH})$, $\theta(\bullet \text{OH}_{\text{surf}})$, and $\theta(\bullet \text{OH}_f)$ indicate the contribution ratio of total •OH, surface-bound •OH, and free •OH, respectively. The k_{obs}^0 , k_{obs}^1 , and k_{obs}^2 represent the

apparent kinetic rate constant of 4-NP degradation by FeOF/H₂O₂ system with no quencher addition, TBA addition, and MeOH addition, respectively. After calculations, the free and surface-bound •OH radicals account for 94.3% and 3.5% of the contributions to the neutral Fenton process initiated by FeOF.

To clarify this result, we have revised the manuscript description as follows.

Line 430-431, “However, 4-NP can be hardly removed by individual H₂O₂ (Fig. S27), FeOF alone (Fig. S28) or leaching iron ions (Fig. S29) at neutral condition.”

Line 442-452, “Moreover, we also observed that both methanol and tert-butanol, two representative scavengers for •OH, showcased a severe inhibition effect in 4-NP removal at an excessive dosage (Fig. S32). The residual 4-NP decomposition rate (~1.0%) is probably attributed to the marginal adsorption capability of FeOF. In particular, the 4-NP decomposition rate showed an evident decline from 95.4% to 28.1% (Fig. 5C) in the presence of TBA at a relatively lower concentration (*i.e.*, 10 mM). The equal addition of MeOH retarded 70.6% of 4-NP being degraded by FeOF/H₂O₂ system, which was about 2.3% higher than TBA inhibition, highlighting the slight contribution of surface-bound •OH radicals for 4-NP degradation. The contribution of free and surface-bound •OH radicals were quantified and calculated according to previous study using the k_{obs} value before and after quenching (Fig. S33) (47).”

Page 3 in Supplementary Information, “The contribution of free and surface-bound •OH radicals were quantified and calculated by following equations according to previous study using the k_{obs} value before and after quenching (6).

$$\begin{aligned}\theta(\bullet\text{OH}) &= [(k_{\text{obs}}^0 - k_{\text{obs}}^1)/k_{\text{obs}}^0] \times 100\% \\ \theta(\bullet\text{OH}_{\text{surf}}) &= [(k_{\text{obs}}^0 - k_{\text{obs}}^2)/k_{\text{obs}}^0] \times 100\% \\ \theta(\bullet\text{OH}_f) &= \theta(\bullet\text{OH}) - \theta(\bullet\text{OH}_{\text{surf}})\end{aligned}$$

where the $\theta(\bullet\text{OH})$, $\theta(\bullet\text{OH}_{\text{surf}})$, and $\theta(\bullet\text{OH}_f)$ indicate the contribution ratio of total •OH, surface-bound •OH, and free •OH, respectively. The k_{obs}^0 , k_{obs}^1 , and k_{obs}^2 represent the apparent kinetic rate constant of 4-NP degradation by FeOF/H₂O₂ system with no quencher addition, TBA addition, and MeOH addition, respectively.”

Figure R8. EPR spectrum of DMPO- $\bullet\text{O}_2^-$ (A) and TEMP- $^1\text{O}_2$ (B) for FeOF and FeOCl systems.

Figure R9. Degradation rate of 4-NP by individual H_2O_2 or FeOF system. Reaction conditions: $[\text{H}_2\text{O}_2] = 10 \text{ mM}$ (if used), $[\text{catalyst}] = 0.1 \text{ g L}^{-1}$ (if used), $[\text{4-NP}] = 20 \text{ mg L}^{-1}$, $\text{pH} = 7.0$, Temperature = $20 \text{ }^\circ\text{C}$.

Figure R10. Quenching effect of high concentration TBA and methanol on the 4-NP degradation efficiency by FeOF. Reaction conditions: [H₂O₂] = 10 mM, [catalyst] = 0.1 g L⁻¹, [4-NP] = 20 mg L⁻¹, [MeOH] = [TBA] = 200 mM, pH = 7.0, Temperature = 20 °C.

Figure R11. Quenching or inhibition effect of relatively low concentration TBA and methanol on the 4-NP degradation efficiency by FeOF.

Figure R12. Contribution of $\bullet\text{OH}$ (*i.e.*, $\bullet\text{OH}_f$ and $\bullet\text{OH}_{\text{surf}}$) and others on the 4-NP degradation.

References:

1. Li, M.; Wang, P.; Zhang, K.; Zhang, H.; Bao, Y.; Li, Y.; Zhan, S.; Crittenden, J. C., Single cobalt atoms anchored on Ti₃C₂T_x with dual reaction sites for efficient adsorption-degradation of antibiotic resistance genes. *PNAS* **2023**, *120*, (29), e2305705120.
2. Xu, X.; Zhang, Y.; Chen, Y.; Liu, C.; Wang, W.; Wang, J.; Huang, H.; Feng, J.; Li, Z.; Zou, Z., Revealing *OOH key intermediates and regulating H₂O₂ photoactivation by surface relaxation of Fenton-like catalysts. *PNAS* **2022**, *119*, (36), e2205562119.

Comment 5: *Some formatting and grammar comments: Line 547 and 549, “i.e.” needs to be italic; “Degradation rate” in Fig. S35, “Detailed information” in caption of Table S5, and “N.Z.” in author contributions should be revised.*

Response: We express our gratitude to the reviewer for this comment. Throughout the manuscript, we have adjusted all instances of “*i.e.*” and “*e.g.*” to the italic format for consistency. The table caption has been revised into “Information (*i.e.*, E_b, I_{O-O}, and charge transfer) for the H₂O₂ adsorption onto various facets of FeOF and FeOCl.”. The notion “N.Z.” has been removed from author contributions section. In addition, we also carefully check the formatting and grammar of our manuscript to avoid misleading description.

To clarify these changes, we listed the revised description of manuscript as follows.

Line 191-194, “As shown in Fig. 2A, Fe K-edge X-ray absorption near-edge structure (XANES) spectra indicate that the absorption energy of FeOF was evidently dissociate to Fe foil but intimate to Fe₂O₃ and FeF₃, implying that Fe species in FeOF carries positive charge toward +3 (39).”

Line 197-200, “Moreover, the lower normalized intensity of white peak locating at near-edge (7130.2 eV) in the XANES spectrum of FeOF implied the five-fold coordination by F and O atoms for unsaturated coordination iron sites (46, 48).”

Line 220-222, “This observation justified the characteristic moiety of F-(Fe(III)O₃)-F in the fine local structure elucidation of FeOF, which was consistent with Mössbauer spectroscopy and XANES analysis.”

Line 439-442, “Similar to •OH generation inhibition, we observed a strong suppression of 4-NP decay with almost zero percent degradation rate (Fig. 5C), suggesting that unsaturated iron centers (*e.g.*, F-(Fe(III)O₃)-F moiety) is responsible for 4-NP destruction.”

Line 442-444, “Moreover, we also observed that both methanol and tert-butanol, two representative scavengers for •OH, showcased a severe inhibition effect in 4-NP removal at an excessive dosage (Fig. S32).”

Line 498-499, “Together with the high reactivity, the as-prepared FeOF catalyst can be operated at more adaptive pH range than the state-of-the-art Fenton catalysts.”

Line 581-583, “For the •OH scavenging experiment, MeOH (10 mM) and TBA (10 mM) were used to quench •OH (*i.e.*, •OH_f and •OH_{surf}) and •OH_f radicals, respectively (63).”

Line 583-589, “The cyclic experiments (with or without adding H₂O₂), Fe leaching experiments, coexisting ion (*i.e.*, Cl⁻) experiments, pH (*i.e.*, 3.0, 4.0, 5.0, 6.0, 7.0, 8.0, and 9.0) experiments, H₂O₂ dosage (1.0, 5.0, 10.0, 15.0, and 20.0 mM) experiments, temperature experiments (*i.e.*, 20, 30, and 40 °C), contaminants (*i.e.*, tetracycline, atrazine, sulfamethoxazole, ofloxacin, and salicylic acid) experiments, and industrial effluents experiments were conducted to explore the practical considerations of pH-adaptive FeOF/H₂O₂ Fenton system.”

Reviewer #3

General comment: *This study presents an enhancement in Fenton catalysis achieved through the incorporation of fluoride, the most electronegative halogen, into the iron oxyhalide structure. The results clearly indicate that the distinct electron density polarization in FeOF leads to higher Fenton efficiency compared to FeOCl. However, certain aspects of the mechanisms, such as the regeneration of Fe(III), and the quantitative relationship between Fenton efficiency and local electron density, lack comprehensive explanations.*

Response: We express our gratitude to the reviewer for acknowledging the research value and providing constructive comments to enhance the quality of this work. We have answered the reviewer's comments point by point and rectified any relevant issues. We would greatly appreciate it if the reviewer could reassess our revised manuscript for potential publication in the journal.

Comment 1: *It is challenging to discern the individual contributions of facets and electronegativity to variations in Fenton efficiency. The primary facets of FeOCl and FeOF exhibit differences, with FeOCl predominantly exposing facets (010) and (110), while facets (211) and (101) are dominant for FeOF. Although the paper includes thermodynamic calculations for H₂O₂ activation on each facet, it is difficult to evaluate efficiency comprehensively when the relative abundance of each facet significantly influences the results. As suggested in Line 364, if the dominant (010) facet of FeOCl indeed has a lower capacity for H₂O₂ activation, it becomes especially challenging to quantitatively relate electron density polarization to Fenton efficiency merely through a comparison of the as-synthesized FeOCl and FeOF.*

Response: We sincerely thank the reviewer for such insightful and meaningful comment. Indeed, significant challenges remain in discerning the contribution of facets to variation in Fenton efficiency. To identify the main contribution of electronegativity against facets to Fenton efficiency, we first investigated the electronic structures of FeOF and FeOCl with DFT calculations. Our investigations revealed distinct characteristics regarding electronic structures of iron sites between FeOF and FeOCl. Specifically, iron sites in FeOF catalyst exhibited a maximum valence-electron density of ca. $0.04 e \text{ \AA}^{-3}$ (Fig. S10), which was 5 times smaller than that in FeOCl (ca. $0.20 e \text{ \AA}^{-3}$). With the analysis of valence state, we observed that the iron center lost ca. 1.71 e mainly transferring to the neighboring fluoride atom in FeOF while the

chloride atom in FeOCl only triggered *ca.* 0.43 *e* withdrawn from the neighboring iron center with a total loss of *ca.* 1.47 *e* (Fig. S9B). Note that the calculated data was for bulk iron centers and presented using the (010) facet. This difference in electronic structure of FeOF and FeOCl could be attributed to the electron density polarization facilitated by fluoride coordination.

To gain more insights into H₂O₂ activation, we also applied DFT calculations to study the interaction between iron centers on each exposing facet and H₂O₂ molecules for both FeOF and FeOCl. Due to the stronger electron polarization effect of the fluorine atom than chlorine atom, the corresponding adsorption energies of H₂O₂ on the iron centers at four facets of FeOF were lower than those of FeOCl (Fig. 4A-B and Fig. S14). We further evaluated the thermodynamics for H₂O₂ activation on each facet and concluded that H₂O₂ activation was more effective on iron centers of all facets within FeOF than those within FeOCl except (010) facet (Fig. 4 E-F and Fig. S19-S21). In addition, we found that the ratio of Lewis acid sites with strong strength was greatly increased from 1.9% to 44.9% as the fluorine coordination was integrated. This highlights the function of fluorine toward the modulation of iron sites since the Lewis acidity is normally dependent on the electron density of metal sites (1, 2). These findings supported our deduction that the electron polarization governs Fenton catalysis enhancement through the optimization of surface iron site configuration.

It has been reported that the facets of heterogeneous Fenton catalysts are closely related to their surface atomic configuration and active site abundance, which have a great influence on their physical and chemical properties (3). We conducted DFT calculations to verify the impact of fluorine coordination on the surface iron atom configuration within four main facets (*i.e.*, (101), (110), (211), and (010)). Except the similar H₂O₂ activation performance of (010) facet, other facets of FeOF catalyst have a more favorable thermodynamic on H₂O₂ adsorption and splitting reactions than the corresponding facets of FeOCl. Therefore, the issue of facet contribution concerned most by the reviewer is the relative abundance of each facet, which shows diversity in the amounts of active iron sites. While quantifying the amount of iron sites within each facet poses a considerable challenge, we opted to evaluate the Lewis acidity as an alternative approach because previous reports demonstrated that the Lewis acid sites derived from unsaturated coordination metal sites were responsible for H₂O₂ activation (4-7). Our experimental findings suggested that FeOF had a somewhat higher Lewis acidity than FeOCl (*e.g.*, ~3× higher, Fig. 1F), which means more active sites for H₂O₂ activation. However, this

alone cannot explain the much higher observed disparity in the accumulated generation of •OH (~168× higher). Even if we normalized the apparent kinetic rate constant for •OH generation by total Lewis acidity (**Table R3**), the value for FeOF was about 60 times greater than FeOCl. Therefore, we attribute this notable enhancement to the function of fluorine coordination, which optimizes the electronic structure of iron sites through the high electronegativity.

To clarify this result, we have revised the manuscript description as follows.

Line 312-319, “Although our experimental findings suggested that FeOF had a somewhat higher Lewis acidity than FeOCl (~3× higher, Fig. 1F), this alone failed to explain the much higher observed disparity in the accumulated generation of •OH (~168× higher). Even if we normalized the apparent kinetic rate constant for •OH generation by the total Lewis acidity (Table S4), the value for FeOF was 60 times greater than FeOCl. Therefore, we attribute this significant enhancement to the function of fluorine coordination, which optimizes the electronic structure of iron sites through the high electronegativity.”

Table R3. Summary of apparent rate constants for •OH generation by FeOF and FeOCl with and without normalized by specific surface area and the amount of Lewis acid sites.

Catalyst	Apparent rate constant for •OH generation, $K_{(\bullet\text{OH})}$ ($\mu\text{mol L}^{-1} \text{min}^{-1}$)	Apparent rate constant for •OH generation normalized by specific surface area, K_s ($\mu\text{mol L}^{-1} \text{min}^{-1}$)	Apparent rate constant for •OH generation normalized by the amount of Lewis acid sites			
			Total	Weak	Medium	Strong
FeOF	2.94	31.7	194.2	-	346.1	442.3
FeOCl	0.017	0.371	3.2	7.8	5.6	85.9

References:

1. Yang, Q.; Xu, W.; Gong, S.; Zheng, G.; Tian, Z.; Wen, Y.; Peng, L.; Zhang, L.; Lu, Z.; Chen, L., Atomically dispersed Lewis acid sites boost 2-electron oxygen reduction activity of carbon-based catalysts. *Nature Communications* **2020**, *11*, (1), 5478.
2. Brown, I. D.; Skowron, A., Electronegativity and Lewis acid strength. *J. Am. Chem. Soc.* **1990**, *112*, 3401-3403.

3. Zhu, Y.; Zhu, R.; Xi, Y.; Zhu, J.; Zhu, G.; He, H., Strategies for enhancing the heterogeneous Fenton catalytic reactivity: A review. *Applied Catalysis B-Environmental* **2019**, *255*, 117739.
4. Tang, J.; Wang, J., Metal Organic Framework with Coordinatively Unsaturated Sites as Efficient Fenton-like Catalyst for Enhanced Degradation of Sulfamethazine. *Environ. Sci. Technol.* **2018**, *52*, (9), 5367-5377.
5. Ahmad, M.; Quan, X.; Chen, S.; Yu, H., Tuning Lewis acidity of MIL-88B-Fe with mix-valence coordinatively unsaturated iron centers on ultrathin Ti3C2 nanosheets for efficient photo-Fenton reaction. *Applied Catalysis B-environmental* **2020**, *264*, 118534.
6. Liang, H.; Liu, R.; An, X.; Hu, C.; Zhang, X.; Liu, H., Bimetal-organic frameworks with coordinatively unsaturated metal sites for highly efficient Fenton-like catalysis. *Chem. Eng. J.* **2021**, *414*, 128669.
7. Wu, M.; Wu, Q.; Yang, Y.; He, Z.; Yang, H., Regulating Lewis acidity and local electron density of iron-based metal organic frameworks via cerium doping for efficient photo-Fenton process. *J. Colloid Interface Sci.* **2023**, *630*, 866-877.

Comment 2: Furthermore, as evidenced by Figure S3 and Figure S7, the physical characteristics of synthesized FeOF and FeOCl appear distinct. FeOF seems more particulate, while FeOCl has a planar structure. Therefore, although electronegativity of the halide component largely determines Fenton efficiency, directly attributing experimental results to electronegativity differences seems oversimplified.

Response: We express our gratitude to the reviewer for this constructive suggestion. In the view of heterogeneous catalysis, the physical characteristics such as particle size and surface area significantly affect the activities of catalysts through a combined protocol involving mass-transfer and site-exposure. Recent studies have demonstrated that the decrease of particle size generally increased the exposed surface area of catalysts' active sites, thereby strengthening the contact between reactants and catalyst substantially (1, 2). It has been also reported that downsizing the catalysts' particle presents an efficient strategy to maximize active site density, improve the sites accessibility, and enhance catalytic performance (3-5)

In our study, the spherical FeOF particles (~50 nm) exhibit a much smaller size than the rectangular FeOCl (~500 × 1000 nm) while the specific surface area of FeOF shows 2-fold

increment compared to FeOCl. Inspired by the reviewer’s comment, these observations suggest that the effect of particle size and surface area should be considered. Although we do not have a unified theory that can explain and predict the behavior of catalysts with different particle sizes for reactions such as Fenton catalysis (6), we alternatively calculated the surface area normalized kinetic rate constant for •OH generation to gain a comprehensive comparison as the effect of particle size could be explained in a manner of surface area. We observed that the FeOF exhibited a value of 31.7 $\mu\text{mol g L}^{-1} \text{min}^{-1} \text{m}^{-2}$, which was 85.4 times higher than the FeOCl (**Table R4**). Such increment is close to the observed 168.0-fold enhancement in •OH generation, revealing that the smaller particle size and higher surface area are conducive to Fenton reaction but not the primary factors.

To clarify this result, we have revised the manuscript description as follows.

Line 143-145, “By comparison to FeOCl (Fig. S7), the smaller particle size and higher surface area of FeOF are conducive to Fenton reaction by increasing the density and accessibility of active iron sites.”

Line 280-286, “The surface area normalized kinetic rate constant for •OH generation by FeOF was determined to be 31.7 $\mu\text{mol g L}^{-1} \text{min}^{-1} \text{m}^{-2}$, 85.4 times higher than by FeOCl (Table S4). Such increment closing to the •OH generation enhancement reveals that the smaller particle size and higher surface area are not the primary factors. Consequently, we deduced that the disparity of H₂O₂ activation activity could be ascribed to the presence of F-Fe-O sites, where the iron atom is expected to show much lower electron density over Cl-Fe-O.”

Table R4. Summary of apparent rate constants for •OH generation by FeOF and FeOCl with and without normalized by specific surface area and the amount of Lewis acid sites.

Catalyst	Apparent rate constant for •OH generation, $K_{(\text{OH})}$ ($\mu\text{mol L}^{-1} \text{min}^{-1}$)	Apparent rate constant for •OH generation normalized by specific surface area, K_s ($\mu\text{mol L}^{-1} \text{min}^{-1}$)	Apparent rate constant for •OH generation normalized by the amount of Lewis acid sites			
			Total	Weak	Medium	Strong
FeOF	2.94	31.7	194.2	-	346.1	442.3
FeOCl	0.017	0.371	3.2	7.8	5.6	85.9

References:

1. Polshettiwar, V.; Varma, R. S., Green chemistry by nano-catalysis. *Green Chem.* **2010**, *12*, (5), 743-754.
2. Tang, Z.; Zhao, P.; Wang, H.; Liu, Y.; Bu, W., Biomedicine Meets Fenton Chemistry. *Chem. Rev.* **2021**, *121*, (4), 1981-2019.
3. Zhao, J.-W.; Wang, H.-Y.; Feng, L.; Zhu, J.-Z.; Liu, J.-X.; Li, W.-X., Crystal-Phase Engineering in Heterogeneous Catalysis. *Chem. Rev.* **2023**, doi: <https://doi.org/10.1021/acs.chemrev.3c00402>.
4. Kim, J. H.; Yoon, S.; Baek, D. S.; Kim, J.; Kim, J.; An, K.; Joo, S. H., Boosting Thermal Stability of Volatile Os Catalysts by Downsizing to Atomically Dispersed Species. *Jacs Au* **2022**, *2*, (8), 1811-1817.
5. Li, Z.; Ji, S.; Liu, Y.; Cao, X.; Tian, S.; Chen, Y.; Niu, Z.; Li, Y., Well-Defined Materials for Heterogeneous Catalysis: From Nanoparticles to Isolated Single-Atom Sites. *Chem. Rev.* **2020**, *120*, (2), 623-682.
6. Liu, L.; Corma, A., Metal Catalysts for Heterogeneous Catalysis: From Single Atoms to Nanoclusters and Nanoparticles. *Chem. Rev.* **2018**, *118*, (10), 4981-5079.

Comment 3: *Building upon the previous point, as the synthesis pathway for FeOF and FeOCl differ, FeOF through preconcentration of FeSiF₆·H₂O and FeOCl through simple partial pyrolysis of FeCl₆·6H₂O, how can the authors ensure their structural congruence?*

Response: We sincerely appreciate the valuable comment. From a molecule structure perspective, both FeOF and FeOCl feature isostructural octahedral FeO₄X₂ (X=F or Cl) layers, belonging to 2D van der Waals iron oxyhalides. The primary difference between FeOF and FeOCl is the type of halogen atoms coordinated to iron centers (Fig. S9). This difference also leads to the variation in the distance between each layer. Wang *et al.* investigated the stable and intrinsic antiferromagnetic properties of FeOF and FeOCl monolayers with similar molecule structures by employing first-principles calculations and Monte Carlo simulations (1). Inspired by their work, we also applied a similar molecule structure for DFT calculations to initially elucidate the structure diversity of iron centers. By comparing the electronic structures of FeOF and FeOCl, we found that FeOF potentially showed greater ability to activate H₂O₂ than FeOCl.

Then, we further conducted experiments to investigate and compare the Fenton efficiency (*i.e.*, •OH generation ability) of synthesized FeOF and FeOCl catalysts. Since reports on FeOF synthesis is relatively scarce, we developed a two-step reaction to fabricate FeOF according to previous report with modification using cost-effective iron powder as the precursor (2). It involved the synthesis and solvothermal processing of $\text{FeSiF}_6 \cdot 6\text{H}_2\text{O}$ to obtain FeOF powder. Although the synthesis method differs from FeOCl, the comparison between FeOF and FeOCl would be reasonable if the crystal structure matches well with the standard cards because they theoretically have isostructural octahedral FeO_4X_2 (X=F or Cl) structures. Therefore, we recorded the XRD pattern of synthesized FeOF and FeOCl catalysts, and observed identical peaks with the standard cards (PDF #70-1522 and PDF #74-1369) without additional peaks, which were consistent with previous reports (2-4). Together with the auxiliary evidence with TEM, EDS, XPS, and FT-IR measurements, their structural congruence could be ensured for both molecule and crystal aspects. However, the fine structure of iron centers in FeOF and FeOCl regarding the coordination environment and electronic structure showed significant diversity as DFT calculations revealed (Fig. S9 and S10), dictating the Fenton efficiency.

To clarify this result, we have revised the manuscript description as follows.

Line 153-154, “These findings confirmed the structural congruence between FeOF and FeOCl for both molecular and crystal aspects.”

References:

1. Wang, S.; Wang, J.; Khazaei, M., Discovery of stable and intrinsic antiferromagnetic iron oxyhalide monolayers. *PCCP* **2020**, *22*, (20), 11731-11739.
2. Pereira, N.; Badway, F.; Wartelsky, M.; Gunn, S.; Amatucci, G. G., Iron Oxyfluorides as High Capacity Cathode Materials for Lithium Batteries. *J. Electrochem. Soc.* **2009**, *156*, (6), A407-A416.
3. Kitajou, A.; Komatsu, H.; Nagano, R.; Okada, S., Synthesis of FeOF using roll-quenching method and the cathode properties for lithium-ion battery. *J. Power Sources* **2013**, *243*, 494-498.
4. Sun, M.; Chu, C.; Geng, F.; Lu, X.; Qu, J.; Crittenden, J.; Elimelech, M.; Kim, J.-H., Reinventing Fenton Chemistry: Iron Oxychloride Nanosheet for pH-Insensitive H_2O_2 Activation. *Environmental Science & Technology Letters* **2018**, *5*, (3), 186-191.

Comment 4: *The manuscript does provide evidence of reduced electron density at the iron center in FeOF, yet the link to the Fenton reaction remains somewhat missing. Especially, the redox cycle of the iron center during the Fenton reaction remains unclear. The authors describe the H₂O₂ activation pathway in Line 347-348 but do not elucidate the valence state or coordination environment of the iron active site during this reaction pathway. While the authors mention the formation of Fe(II) through XPS and Mössbauer measurements, this does not fully explain the reasons for the facile reversibility of the Fe(II)/Fe(III) cycle, which is necessary for high Fenton efficiency.*

Response: We appreciate your thoughtful and constructive comments. As comparison to FeOCl, the electron density of iron centers in FeOF is significantly decreased by 0.24 *e* with the fluorine coordination. By linking the reduced electron density to •OH generation, we found that the Fenton efficiency could be largely improved by 700 times as reducing one electron within iron centers if other margin contributions, such as morphology, are disregarded. However, it's important to acknowledge that this tentative correlation remains further validation and enrichment.

In the meantime, we acknowledge the need for a more comprehensive discussion regarding the redox cycle of the iron center during the Fenton reaction. In response to your comment, we have revised the manuscript to provide a more detailed analysis of the coordination environment of the iron site during H₂O₂ activation. The iron site on the surface of (211) facet shows tetrahedral coordination structure with 3 oxygen atoms and 1 fluorine or chlorine atom (Fig. 4 C-D) at initial stage while the surfaces of (110) and (101) facets contain pentahedral coordination iron centers with 3 oxygen atoms and 2 fluorine or chlorine atoms (Fig. S18). Upon H₂O₂ adsorption, the coordination number of iron site increases by one and returns to original ones after H₂O₂ activation and •OH dissociation (Fig. 4E). For (010) facets (Fig. S18), our observations demonstrate the presence of saturated coordination iron sites with 4 oxygen atoms and 2 fluorine or chlorine atoms. In this context, H₂O₂ is hard to be adsorbed and activated by iron sites, which explains the inferior catalytic performance of (010) facets.

It has been reported that the reversibility Fe(II)/Fe(III) cycle in the Fenton catalyst during reaction could be estimated using electrochemical techniques (1). To address the concern about

the facile reversibility of the Fe(II)/Fe(III) cycle, we conducted electrochemical studies, including the open-circuit potential analysis and cyclic voltammetry measurements, to gain more insights for comparison of FeOF and FeOCl. The open-circuit potential displayed in **Figure R13A** highlights a higher potential of Fe(III) in FeOF (0.142 V) than that in FeOCl (0.108 V), affirming the stronger activity to produce Fe(II) that results in the stronger oxidation capacity of the FeOF-catalyzed Fenton system (2). As shown in **Figure R13B**, obvious oxidation and reduction peaks appeared at 0.04 and 0.32 V with FeOF, implying the excellent reversibility of the Fe(II)/Fe(III) redox process. The fluorine coordination also leads to a decrease in peak-to-peak separation (ΔE_p) from 0.40 V to 0.28 V, indicating the improved Fe(II)/Fe(III) redox capability of FeOF (3). Furthermore, the in-situ electrode potential analysis was performed to investigate the redox cycle of the iron center during the Fenton reaction. The addition of H₂O₂ triggers the increase in open-circuit potentials for both FeOF and FeOCl while the former exhibits a more significant increment (**Figure R13C**), which further elucidates the more facile reversibility of the Fe(II)/Fe(III) cycle with fluorine coordination (4). In addition, we found that the proportion of leaching Fe²⁺ ions increased from 3% to 42% (**Figure R14**). The increase of leaching Fe²⁺ ions proportion indicates an enhancement in the Fe(II)/Fe(III) redox cycle. Collectively, these supplementary results well elucidate the presence of the more facile reversibility of Fe(II)/Fe(III) cycle in FeOF than FeOCl, which is crucial for the improved efficiency of the Fenton reaction.

To clarify this result, we have revised the manuscript description as follows.

Line 368-371, “The changes in the free energies for •OH formation in (211), (110), (101), and (010) facets of FeOF and FeOCl are displayed in Fig. 4 E-F and Fig. S19-S21. The coordination environment evolutions of corresponding iron sites are presented in the caption.”

Line 814-817, “At initial step (i), the iron site on the surface of (211) facet shows tetrahedral coordination structure with 3 oxygen atoms and 1 fluorine or chlorine atom. The coordination number of iron site increases by one upon H₂O₂ adsorption and returns to the original ones after H₂O₂ activation and •OH dissociation.”

Line 477-486, “We further performed XPS, Mössbauer, and electrochemical measurements to support the speculation of Fe(II) formation (Fig. S37, S38, and S39) (43, 44, 55). The results suggested that ~26.5% of Fe(II) was generated after Fenton process, which substantially

facilitated the cycling of Fe(III)/Fe(II) redox as well as the degradation efficiency for 4-NP. The observation of higher open-circuit potential of Fe(III) in FeOF and smaller peak-to-peak separation (ΔE_p) (from 0.40 V to 0.28 V) depicted in Fig. S39 indicated the more facile reversibility of the Fe(III)/Fe(II) cycle of FeOF with fluorine coordination (56-58). Besides, we also found that the proportion of leaching Fe²⁺ ions increased from 3% to 42% (Fig. S39B), further evidencing the enhancement of Fe(III)/Fe(II) redox cycle.”

Page 23 in Supplementary Information, “Fig. S19. (A) Calculated free energy profiles for H₂O₂ activation and •OH formation on (110) facet of FeOF and FeOCl. (B) The corresponding structures of reaction intermediates as well as the coordination environment evolution of iron site on (110) facet for the generation of •OH. At initial step (i), the iron site on the surface of (110) facet shows pentahedral coordination structure with 3 oxygen atoms and 2 fluorine or chlorine atoms. The coordination number of iron site increases by one upon H₂O₂ adsorption and returns to original ones after H₂O₂ activation and •OH dissociation.”

Page 24 in Supplementary Information, “Fig. S20. (A) Calculated free energy profiles for H₂O₂ activation and •OH formation on (101) facet of FeOF and FeOCl. (B) The corresponding structures of reaction intermediates as well as the coordination environment evolution of iron site on (101) facet for the generation of •OH. At initial step (i), the iron site on the surface of (101) facet shows pentahedral coordination structure with 3 oxygen atoms and 2 fluorine or chlorine atoms. The coordination number of iron site increases by one upon H₂O₂ adsorption and returns to original ones after H₂O₂ activation and •OH dissociation.”

Page 25 in Supplementary Information, “Fig. S21. (A) Calculated free energy profiles for H₂O₂ activation and •OH formation on (010) facet of FeOF and FeOCl. (B) The corresponding structures of reaction intermediates as well as the coordination environment evolution of iron site on (010) facet for the generation of •OH. At initial step (i), the iron site on the surface of (010) facet shows saturated coordination structure with 4 oxygen atoms and 2 fluorine or chlorine atoms. The coordination number of iron site remains unchanged during Fenton reaction. The highest energy barrier for H₂O₂ adsorption indicates that H₂O₂ is hard to be adsorbed and activated by iron sites, which explains the inferior catalytic performance of (010) facets.”

Page 43 in Supplementary Information, “The open-circuit potential displayed in Fig. S39A highlights a higher potential of Fe(III) in FeOF (0.142 V) than that in FeOCl (0.108 V), affirming the stronger activity to produce Fe(II) that results in the stronger oxidation capacity of the FeOF-catalyzed Fenton system (10). As shown in Fig. S39B, obvious oxidation and reduction peaks appeared at 0.04 and 0.32 V with FeOF, implying the excellent reversibility of the Fe(II)/Fe(III) redox process. The fluorine coordination also leads to a decrease in peak-to-peak separation (ΔE_p) from 0.40 V to 0.28 V, indicating the improved Fe(II)/Fe(III) redox capability of FeOF (11). Furthermore, the in-situ electrode potential analysis was performed to investigate the redox cycle of the iron center during the Fenton reaction. The addition of H_2O_2 triggers the increase in open-circuit potentials for both FeOF and FeOCl while the former exhibits a more significant increment (Fig. S39C), which further elucidates the more facile reversibility of the Fe(II)/Fe(III) cycle with fluorine coordination (12). In addition, we found that the proportion of leaching Fe^{2+} ions increased from 3% to 42% (Fig. S13), indicating the enhancement of Fe(II)/Fe(III) redox cycle.”

Figure R13. (A) Open-circuit potential curves measurements, (B) cyclic voltammery measurements, and (C) open-circuit potential curves with the in-situ addition of H_2O_2 on FeOF and FeOCl coated glassy carbon electrode.

Figure R14. Concentration and proportion of leached Fe³⁺ and Fe²⁺ ions in the FeOF/H₂O₂ system.

References:

1. Wang, C.; Zhang, W.; Wang, J.; Xia, P.; Duan, X.; He, Q.; Sires, I.; Ye, Z., Accelerating Fe(III)/Fe(II) redox cycling in heterogeneous electro-Fenton process via S/Cu-mediated electron donor-shuttle regime. *Applied Catalysis B-Environment and Energy* **2024**, *342*, 123457.
2. Tan, W.; Ren, W.; Wang, C.; Fan, Y.; Deng, B.; Lin, H.; Zhang, H., Peroxymonosulfate activated with waste battery-based Mn-Fe oxides for pollutant removal: Electron transfer mechanism, selective oxidation and LFER analysis. *Chem. Eng. J.* **2020**, *394*, 124864.
3. Liu, M.; Feng, Z.; Luan, X.; Chu, W.; Zhao, H.; Zhao, G., Accelerated Fe²⁺ Regeneration in an Effective Electro-Fenton Process by Boosting Internal Electron Transfer to a Nitrogen-Conjugated Fe(III) Complex. *Environ. Sci. Technol.* **2021**, *55*, (9), 6042-6051.
4. Mei, S.-C.; Li, L.; Huang, G.-X.; Pan, X.-Q.; Yu, H.-Q., Heterogeneous Fenton water purification catalyzed by iron phosphide (FeP). *Water Res.* **2023**, *241*, 120151-120151.

Comment 5: If ROS other than •OH were responsible for the degradation of 4-NP in the presence of •OH quenchers (e.g., MeOH and TBA), the formation pathway of these ROS needs to be elucidated. Is the removal of 4-NP genuinely attributed to other ROS species, as suggested in Line 396-397, which states that 4-NP is recalcitrant towards other ROS species? It's worth noting that

the concentration of $\bullet\text{OH}$ quenchers, 10 mM, appears low compared to the H_2O_2 concentration (10 mM)

Response: We thank the reviewer for the valuable suggestion. To identify whether other ROS species were simultaneously formed or not, we performed additional ERP measurements to monitor the presence of singlet oxygen and superoxide radicals. We found that there were no characteristic signals in the EPR spectra for FeOF/ H_2O_2 system (**Figure R15**). This suggests that neither singlet oxygen nor superoxide radicals were generated and responsible for pollutant degradation. Besides, we also found that single H_2O_2 or leaching iron failed to oxidize/remove 4-NP while individual FeOF could eliminate $\sim 1.0\%$ of 4-NP via adsorption (**Figure R16**). Meanwhile, inspired by your comment, we increased the dosage of $\bullet\text{OH}$ quenchers to 200 mM, much greater than the H_2O_2 input, to evaluate the inhibition effect (**Figure R17**). The 4-NP removal also reduced to $\sim 1.0\%$ with the addition of high dosage quencher (*i.e.*, 200 mM), which matched well to the marginal adsorption capacity. Therefore, we assured that the generated $\bullet\text{OH}$ was responsible for the 4-NP without the participation of other ROS.

To clarify this result, we have revised the manuscript description as follows.

Line 444-445, “The residual 4-NP decomposition rate ($\sim 1.0\%$) is probably attributed to the marginal adsorption capability of FeOF.”

Figure R15. EPR spectrum of DMPO- $\bullet\text{O}_2^-$ (A) and TEMP- $^1\text{O}_2$ (B) for FeOF and FeOCl systems.

Figure R16. Degradation rate of 4-NP by individual H₂O₂ or FeOF system. Reaction conditions: [H₂O₂] = 10 mM (if used), [catalyst] = 0.1 g L⁻¹ (if used), [4-NP] = 20 mg L⁻¹, pH = 7.0, Temperature = 20 °C.

Figure R17. Quenching effect of high concentration TBA and methanol on the 4-NP degradation efficiency by FeOF. Reaction conditions: [H₂O₂] = 10 mM, [catalyst] = 0.1 g L⁻¹, [4-NP] = 20 mg L⁻¹, [MeOH] = [TBA] = 200 mM, pH = 7.0, Temperature = 20 °C.

Comment 6: Finally, it's important to investigate whether there is any leaching of iron and fluoride during the process.

Response: We thank the reviewer for such insightful suggestion. We conducted additional experiments to evaluate the iron leaching and the contribution of iron ions. Initially, by applying ICP-OES measurements, the total iron leaching of FeOF/H₂O₂ Fenton system determined as 0.52 mg L⁻¹ at 30 min. Subsequently, when testing the Fenton activity of individual Fe³⁺ with considerable higher dosage (10.00 mg L⁻¹) than the leaching level, we observed that almost no 4-NP degradation occurred in the reaction (**Figure R18**). To comprehensively understand the homogeneous contribution, we further identified the species of leached iron ions via a spectrometric method using 1,10-phenanthroline as the probe according to a previous report (1). It was found that the Fe leaching concentration remained relative constant at a range of 0.20~0.40 mg L⁻¹ during reaction (**Figure R19**). As the reaction prolonged, the proportion of Fe²⁺ ions increased from 3% to 42%, indicating an enhancement of Fe(II)/Fe(III) redox cycle. In this context, we examined the •OH generation ability and 4-NP degradation performance of individual Fe³⁺ (0.15 mg L⁻¹) and Fe²⁺ (0.15 and 0.30 mg L⁻¹) ions, as well as their mixture at similar concentrations. As illustrated in **Figure R20**, marginal •OH (~0.1 μM) was generated, and little 4-NP was destructed when employing iron ions as the catalyst. Therefore, we excluded the homogeneous contribution of iron ions to the Fenton reaction.

We measured the fluoride leaching by a fluoride-selective electrode (Leici PF-202, China) connected to a pH meter according to previous studies (2, 3). As shown in **Figure R21**, we observed that less than 0.40 mg L⁻¹ of fluoride leached in the solution during Fenton reaction, which conformed to the drinking water standard issued by USA (2.00 mg L⁻¹), WHO (1.00 mg L⁻¹), and China (1.00 mg L⁻¹).

To clarify these results, we have revised the manuscript description as follows.

Line 277-280, “We also excluded the homogeneous contribution of leached iron ions because scarce •OH was generated when using Fe²⁺, Fe³⁺, and Fe²⁺/Fe³⁺ mixture as the catalyst at equivalent concentrations to the leaching level (Fig. S13).”

Line 430-431, “However, 4-NP can be hardly removed by individual H₂O₂ (Fig. S27), FeOF alone (Fig. S28), or leaching iron ions (Fig. S29) at neutral condition.”

Line 433-436, “We also observed that less than 0.40 mg L⁻¹ of fluoride leached during Fenton reaction (Fig. S31), which conformed to the drinking water standards issued by USA (2.00 mg L⁻¹), WHO (1.00 mg L⁻¹), and China (1.00 mg L⁻¹)”

Figure R18. H₂O₂ activation performance of free Fe³⁺ ions with much higher concentration than the leachate of FeOF/H₂O₂ system. Reaction conditions: [H₂O₂] = 10 mM, [Fe³⁺] = 10 mg L⁻¹, [4-NP] = 20 mg L⁻¹, pH = 7.0, Temperature = 20 °C.

Figure R19. Concentration and proportion of leached Fe³⁺ and Fe²⁺ ions in the FeOF/H₂O₂ system.

Figure R20. Accumulated $\bullet\text{OH}$ generation and 4-NP degradation by H_2O_2 activation using Fe^{2+} or/and Fe^{3+} ions as the catalyst. Reaction conditions: $[\text{H}_2\text{O}_2] = 10 \text{ mM}$, $[\text{Coumarin}] = 10 \text{ mM}$ (if used), $[\text{4-NP}] = 20 \text{ mg L}^{-1}$ (if used), $\text{pH} = 7.0$, Temperature = $20 \text{ }^\circ\text{C}$.

Figure R21. Concentration of fluoride leaching during Fenton reaction with FeOF as the catalyst. Reaction conditions: $[\text{H}_2\text{O}_2] = 10 \text{ mM}$, $[\text{catalyst}] = 0.1 \text{ g L}^{-1}$, $[\text{4-NP}] = 20 \text{ mg L}^{-1}$, $\text{pH} = 7.0$, Temperature = $20 \text{ }^\circ\text{C}$.

References:

1. Harvey, A. E.; Smart, J.; Amis, E. S., Simultaneous Spectrophotometric Determination of Iron(II) and Total Iron with 1,10-Phenanthroline. *Anal. Chem.* 1955, 27, 26-29.
2. Pan, B.; Xu, J.; Wu, B.; Li, Z.; Liu, X., Enhanced Removal of Fluoride by Polystyrene Anion Exchanger Supported Hydrous Zirconium Oxide Nanoparticles. *Environ. Sci. Technol.* **2013**, 47, (16), 9347-9354.
3. Zhang, X.; Zhang, L.; Li, Z.; Jiang, Z.; Zheng, Q.; Lin, B.; Pan, B., Rational Design of Antifouling Polymeric Nanocomposite for Sustainable Fluoride Removal from NOM-Rich Water. *Environ. Sci. Technol.* **2017**, 51, (22), 13363-13371.

Comment 7: Line 186 – 216: Correct “EXANS” to “XANES”.

Line 412: “Similar to c generation inhibition”.

Line 415: Correct “methane” to “methanol”.

Response: We express our gratitude to the reviewer for this comment. In order to avoid similar typo and other mistakes, we thoroughly rechecked through the manuscript, and made following revisions.

Line 191-194, “As shown in Fig. 2A, Fe K-edge X-ray absorption near-edge structure (XANES) spectra indicate that the absorption energy of FeOF was evidently dissociate to Fe foil but intimate to Fe₂O₃ and FeF₃, implying that Fe species in FeOF carries positive charge toward +3 (39).”

Line 197-200, “Moreover, the lower normalized intensity of white peak locating at near-edge (7130.2 eV) in the XANES spectrum of FeOF implied the five-fold coordination by F and O atoms for unsaturated coordination iron sites (46, 48).”

Line 220-222, “This observation justified the characteristic moiety of F-(Fe(III)O₃)-F in the fine local structure elucidation of FeOF, which was consistent with Mössbauer spectroscopy and XANES analysis.”

Line 439-442, “Similar to $\bullet\text{OH}$ generation inhibition, we observed a strong suppression of 4-NP decay with almost zero percent degradation rate (Fig. 5C), suggesting that unsaturated iron centers (*e.g.*, F-(Fe(III)O₃)-F moiety) is responsible for 4-NP destruction.”

Line 442-444, “Moreover, we also observed that both methanol and tert-butanol, two representative scavengers for $\bullet\text{OH}$, showcased a severe inhibition effect in 4-NP removal at an excessive dosage (Fig. S32).”

Line 498-499, “Together with the high reactivity, the as-prepared FeOF catalyst can be operated at more adaptive pH range than the state-of-the-art Fenton catalysts.”

Line 581-583, “For the $\bullet\text{OH}$ scavenging experiment, MeOH (10 mM) and TBA (10 mM) were used to quench $\bullet\text{OH}$ (*i.e.*, $\bullet\text{OH}_f$ and $\bullet\text{OH}_{\text{surf}}$) and $\bullet\text{OH}_f$ radicals, respectively (63).”

Line 583-589, “The cyclic experiments (with or without adding H₂O₂), Fe leaching experiments, coexisting ion (*i.e.*, Cl⁻) experiments, pH (*i.e.*, 3.0, 4.0, 5.0, 6.0, 7.0, 8.0, and 9.0) experiments, H₂O₂ dosage (1.0, 5.0, 10.0, 15.0, and 20.0 mM) experiments, temperature experiments (*i.e.*, 20, 30, and 40 °C), contaminants (*i.e.*, tetracycline, atrazine, sulfamethoxazole, ofloxacin, and salicylic acid) experiments, and industrial effluents experiments were conducted to explore the practical considerations of pH-adaptive FeOF/H₂O₂ Fenton system.”

Reviewer #4

General comment: *In this study, the authors reported a case study on the fabrication of FeOF for efficient activation of H₂O₂ to degrade organic pollutants. The enhancement of F doping (over FeOCl) was systematically explored based on extensive characterization of the resultant catalyst, theoretical calculation of the involved chemical processes and states, as well as the preliminary evaluation in water treatment using 4-NP as the model target. Generally, I think the highlight of this study does not rely on the idea of how to fabricate the efficient catalyst, but the significant enhancement of Fenton-like catalysis over the well-known FOCl. Not only surprising to the authors but also to me, why no research has been carried out on the F replacement of Cl since such enhancement seems clearly expected to produce high-efficiency catalyst. Generally, it is a systematic work to advance Fenton-like system to a new level, however, I have still several issues on this work, which should be addressed before further consideration.*

Response: We express our gratitude to the reviewer for acknowledging the research value and providing constructive comments to enhance the quality of this work. We have answered the reviewer's comments point by point and rectified any relevant issues. We would greatly appreciate it if the reviewer could reassess our revised manuscript for potential publication in the journal.

Comment 1: *For heterogeneous Fenton system, except for the chemical structure of the solid catalyst, its size and morphology are also very important since they both are directly related to the effective contact of H₂O₂ to the active sites. Unluckily, the authors did not mention such important factors.*

Response: We express our gratitude to the reviewer for this constructive suggestion. In the view of heterogeneous catalysis, both the particle size and morphology significantly affect the activities of catalysts through a combined protocol involving mass-transfer and site-exposure. Recent studies have demonstrated that the decrease of particle size generally increased the exposed surface area of catalysts' active sites, thereby strengthening the contact between reactants and catalyst substantially (1, 2). It has been also reported that downsizing the catalysts'

particle presents an efficient strategy to maximize active site density, improve the sites accessibility, and enhance catalytic performance (3-5).

In our study, the spherical FeOF particles (~50 nm) exhibit a much smaller size than the rectangular FeOCl (~500 × 1000 nm) while the specific surface area of FeOF shows 2-fold increment compared to FeOCl. Inspired by the reviewer's comment, these observations suggest that the effect of particle size and surface area should be considered. Although we do not have a unified theory that can explain and predict the behavior of catalysts with different particle sizes for reactions such as Fenton catalysis (6), we alternatively calculated the surface area normalized kinetic rate constant for •OH generation to gain a comprehensive comparison as the effect of particle size could be explained in a manner of surface area. We observed that the FeOF exhibited a value of 31.7 $\mu\text{mol g L}^{-1} \text{min}^{-1} \text{m}^{-2}$, which was 85.4 times higher than the FeOCl (**Table R5**). Such increment is close to the observed 168.0-fold enhancement in •OH generation, revealing that the smaller particle size and higher surface area are conducive to Fenton reaction but not the primary factors.

To clarify this result, we have revised the manuscript description as follows.

Line 143-145, “By comparison to FeOCl (Fig. S7), the smaller particle size and higher surface area of FeOF are conducive to Fenton reaction by increasing the density and accessibility of active iron sites.”

Line 280-286, “The surface area normalized kinetic rate constant for •OH generation by FeOF was determined to be 31.7 $\mu\text{mol g L}^{-1} \text{min}^{-1} \text{m}^{-2}$, 85.4 times higher than by FeOCl (Table S4). Such increment closing to the •OH generation enhancement reveals that the smaller particle size and higher surface area are not the primary factors. Consequently, we deduced that the disparity of H₂O₂ activation activity could be ascribed to the presence of F-Fe-O sites, where the iron atom is expected to show much lower electron density over Cl-Fe-O.”

Table R5. Summary of apparent rate constants for •OH generation by FeOF and FeOCl with and without normalized by specific surface area and the amount of Lewis acid sites.

Catalyst	Apparent rate constant for •OH generation, $K_{(\text{OH})}$ ($\mu\text{mol L}^{-1} \text{min}^{-1}$)	Apparent rate constant for •OH generation normalized by specific surface area, K_s ($\mu\text{mol L}^{-1} \text{min}^{-1}$)	Apparent rate constant for •OH generation normalized by the amount of Lewis acid sites			
			Total	Weak	Medium	Strong
FeOF	2.94	31.7	194.2	-	346.1	442.3
FeOCl	0.017	0.371	3.2	7.8	5.6	85.9

References:

1. Polshettiwar, V.; Varma, R. S., Green chemistry by nano-catalysis. *Green Chem.* **2010**, *12*, (5), 743-754.
2. Tang, Z.; Zhao, P.; Wang, H.; Liu, Y.; Bu, W., Biomedicine Meets Fenton Chemistry. *Chem. Rev.* **2021**, *121*, (4), 1981-2019.
3. Zhao, J.-W.; Wang, H.-Y.; Feng, L.; Zhu, J.-Z.; Liu, J.-X.; Li, W.-X., Crystal-Phase Engineering in Heterogeneous Catalysis. *Chem. Rev.* **2023**, doi: <https://doi.org/10.1021/acs.chemrev.3c00402>.
4. Kim, J. H.; Yoon, S.; Baek, D. S.; Kim, J.; Kim, J.; An, K.; Joo, S. H., Boosting Thermal Stability of Volatile Os Catalysts by Downsizing to Atomically Dispersed Species. *Jacs Au* **2022**, *2*, (8), 1811-1817.
5. Li, Z.; Ji, S.; Liu, Y.; Cao, X.; Tian, S.; Chen, Y.; Niu, Z.; Li, Y., Well-Defined Materials for Heterogeneous Catalysis: From Nanoparticles to Isolated Single-Atom Sites. *Chem. Rev.* **2020**, *120*, (2), 623-682.
6. Liu, L.; Corma, A., Metal Catalysts for Heterogeneous Catalysis: From Single Atoms to Nanoclusters and Nanoparticles. *Chem. Rev.* **2018**, *118*, (10), 4981-5079.

Comment 2: As for the utilization of Fenton system, fast production of hydroxyl radical is really crucial, however, the effective usage of the produced radicals in decontamination, particularly for organic degradation, is another side that should be particularly concerned. The authors should discuss how to balance the production process of radicals and its effective attack to the target

compounds since the half life time of the radical is very short, and, if not used, they will be captured by the matrix composition.

Response: We thank the reviewer for such impressive comment. We understand there are two key pathways to enhance the pollutant degradation for Fenton-based water treatment. One is to develop the highly effective catalysts to improve the H₂O₂ activation performance for sustainable •OH generation. This pathway forms the basis for the practice of Fenton reaction, which subsequently needs to concern the utilization. The other is to prevent the invalid consumption of the generated •OH from matrix composition, such as dissolved organic matter and coexisting inorganic ions. Although it seems to extend the scope of the current work, we'd like to discuss the potential strategy to achieve the balance as the reviewer enlightened.

It has been reported that the ultrashort half-life time ($10^{-6}\sim 10^{-9}$ s) of the radicals hinder their mass transfer from the generation sites on the catalyst surface to the target pollutants in bulk solution, severely limiting their utilization in the heterogeneous reactions (1, 2). In addition, the •OH-mediated oxidation reaction generally occurs near the surface of the Fenton catalysts (3, 4). Recently, nanoconfined catalysis has emerged as a viable strategy to improve the radical's utilization efficiency by restricting the desirable reaction occurred within nano-confined space (5). This concept is known as “nanoconfinement”, which was recently well-elucidated by Qian et. al. in a cutting-edge review paper published at ES&T (6). For example, Zhang *et al.* also reported that a diffusion length of < 25 nm is critical to avoid the loss of short-lived species by assessing the Fenton activity of AAO-based nanoreactors with Fe₃O₄ dispersed on the surface of nanoscale channels (7). Therefore, we propose that integrating our FeOF catalyst with the nanoconfinement would present a promising method to achieve the balance of •OH production and utilization. Specifically, the *in situ* growth of nanostructured FeOF catalyst onto well-controlled porous substrates, such as zeolites, metal oxide, and polymeric membranes, would be more feasible for upscaling.

To gain more attention of the balance, we have added additional discussion in the manuscript as follows.

Line 529-531, “To balance the •OH production and utilization, further studies should be conducted to integrate the FeOF catalyst with the nanoconfined space for maximizing the practice potential.”

References:

1. Liu, T.; Xiao, S.; Li, N.; Chen, J.; Zhou, X.; Qian, Y.; Huang, C.-H.; Zhang, Y., Water decontamination via nonradical process by nanoconfined Fenton-like catalysts. *Nature Communications* **2023**, *14*, (1), 2881.
2. Chen, Y.; Zhang, G.; Liu, H.; Qu, J., Confining Free Radicals in Close Vicinity to Contaminants Enables Ultrafast Fenton-like Processes in the Interspacing of MoS₂ Membranes. *Angewandte Chemie-International Edition* **2019**, *58*, (24), 8134-8138.
3. Zhu, Y.; Zhu, R.; Xi, Y.; Zhu, J.; Zhu, G.; He, H., Strategies for enhancing the heterogeneous Fenton catalytic reactivity: A review. *Applied Catalysis B-Environmental* **2019**, *255*, 117739.
4. Enami, S.; Sakamoto, Y.; Colussi, A. J., Fenton chemistry at aqueous interfaces. *PNAS* **2014**, *111*, (2), 623-628.
5. Guo, D.; Wang, Y.; Lu, P.; Liu, J.; Liu, Y., Flow-through electro-Fenton using nanoconfined Fe-Mn bimetallic oxides: Ionization potential-dependent micropollutants degradation mechanism. *Applied Catalysis B-Environmental* **2023**, *328*, 122538.
6. Qian, J.; Gao, X.; Pan, B., Nanoconfinement-Mediated Water Treatment: From Fundamental to Application. *Environ. Sci. Technol.* **2020**, *54*, (14), 8509-8526.
7. Zhang, S.; Sun, M.; Hedtke, T.; Deshmukh, A.; Zhou, X.; Weon, S.; Elimelech, M.; Kim, J.-H., Mechanism of Heterogeneous Fenton Reaction Kinetics Enhancement under Nanoscale Spatial Confinement. *Environ. Sci. Technol.* **2020**, *54*, (17), 10868-10875.

Comment 3: *As for the 4-NP degradation, the authors stated that part degradation is attributed to the other radicals (singlet oxygen and others, Line 417-418). It seems contradictory to the early description that NP is refractory to such radicals (L 397). Please check.*

Response: We thank the reviewer for the valuable suggestion. To identify whether other ROS species were simultaneously formed or not, we performed additional ERP measurements to monitor the presence of singlet oxygen and superoxide radicals. We found that there were no characteristic signals in the EPR spectra for FeOF/H₂O₂ system (**Figure R22**). This suggests that neither singlet oxygen nor superoxide radicals were generated and responsible for pollutant degradation. Besides, we also found that single H₂O₂ or leaching iron failed to oxidize/remove 4-NP while individual FeOF could eliminate ~1.0% of 4-NP via adsorption (**Figure R23**).

Meanwhile, inspired by Reviewer #3's Comment 5, we increased the dosage of $\bullet\text{OH}$ quenchers to 200 mM, much greater than the H_2O_2 input, to evaluate the inhibition effect (**Figure R24**). The 4-NP removal also reduced to $\sim 1.0\%$ with the addition of high dosage quencher, which matched well to the marginal adsorption capacity. Therefore, we assured that the generated $\bullet\text{OH}$ was responsible for the 4-NP without the participation of other ROS.

To clarify this result, we have revised the manuscript description as follows.

Line 444-445, "The residual 4-NP decomposition rate ($\sim 1.0\%$) is probably attributed to the marginal adsorption capability of FeOF."

Figure R22. EPR spectrum of $\text{DMPO}\cdot\text{O}_2^-$ (A) and $\text{TEMP}\cdot^1\text{O}_2$ (B) for FeOF and FeOCl systems.

Figure R23. Degradation rate of 4-NP by individual H₂O₂ or FeOF system. Reaction conditions: [H₂O₂] = 10 mM (if used), [catalyst] = 0.1 g L⁻¹ (if used), [4-NP] = 20 mg L⁻¹, pH = 7.0, Temperature = 20 °C.

Figure R24. Quenching effect of high concentration TBA and methanol on the 4-NP degradation efficiency by FeOF. Reaction conditions: [H₂O₂] = 10 mM, [catalyst] = 0.1 g L⁻¹, [4-NP] = 20 mg L⁻¹, [MeOH] = [TBA] = 200 mM, pH = 7.0, Temperature = 20 °C.

Comment 4: *Why 4-NP cannot be removed by FeOCl/H₂O₂? As limited by the experimental conditions? Need further clarification.*

Response: We thank the reviewer for this suggestion. The removal efficiency of 4-NP utilizing FeOCl/H₂O₂ was approximately 6.2%, a percentage that potentially stems from a significantly lower production of •OH radicals (2.1 μM) compared to FeOF/H₂O₂ (353.0 μM) at equivalent concentrations. This observation also aligns well with previous findings demonstrated by Sun *et al.*'s work (1). The disparity primarily originates from the distinctly inherent H₂O₂ activation efficiencies in these two iron-based catalysts, constituting the main point of our investigation. It has been reported that adjusting experimental conditions, such as reducing pH value of solutions and increasing H₂O₂ or FeOCl concentration, improved the •OH generation (1-3). Therefore, it is plausible that reducing pH value of solutions and increasing H₂O₂ or FeOCl concentration could yield more •OH radicals for 4-NP degradation.

To further clarify this result, we have revised the manuscript description as follows.

Line 424-425, “The FeOCl/H₂O₂ system could only remove about 6.2% of 4-NP under current conditions.”

References:

1. Sun, M.; Chu, C.; Geng, F.; Lu, X.; Qu, J.; Crittenden, J.; Elimelech, M.; Kim, J.-H., Reinventing Fenton Chemistry: Iron Oxychloride Nanosheet for pH-Insensitive H₂O₂ Activation. *Environmental Science & Technology Letters* **2018**, *5*, (3), 186-191.
2. Sun, M.; Zucker, I.; Davenport, D. M.; Zhou, X.; Qu, J.; Elimelech, M., Reactive, Self-Cleaning Ultrafiltration Membrane Functionalized with Iron Oxychloride Nanocatalysts. *Environ. Sci. Technol.* **2018**, *52*, (15), 8674-8683.
3. Zhang, S.; Hedtke, T.; Zhu, Q.; Sun, M.; Weon, S.; Zhao, Y.; Stavitski, E.; Elimelech, M.; Kim, J.-H., Membrane-Confined Iron Oxychloride Nanocatalysts for Highly Efficient Heterogeneous Fenton Water Treatment. *Environ. Sci. Technol.* **2021**, *55*, (13), 9266-9275.

Comment 5: L 448. *Self-enhanced? How to realize such effect?*

Response: We express our gratitude to the reviewer for this comment. The self-enhancement effect observed in our work arises from the accelerated 4-NP degradation under successive cycling with the addition of both H₂O₂ and 4-NP. In this scenario, the self-enhancement might be triggered by the presence of excessive H₂O₂, which plays two key roles in the Fenton reaction with Fe(III) sites according to previous reports (1, 2).

It has been demonstrated that the single electron transfer H₂O₂ molecule from to Fe(III) sites determined the rate-limiting step of Fe(II) formation, which depends on the electron deficiency of Fe(III) sites. After the formation of Fe(II), the H₂O₂ was subjected to a homogeneous cleavage on Fe(II) to generate •OH radicals (1, 2). The turnover of Fe(II)–Fe(III) in FeOCl catalyst upon reaction with H₂O₂ is illustrated in **Figure R25**.

To further identify the change of iron sites after catalysis, we conducted *ab initio* molecular dynamics (MD) simulations to investigate the evolution and fate of H₂O₂ on FeOF and FeOCl. An initial structure consisting of one H₂O₂ molecule and an exposed (211) facet of the FeOF or FeOCl was applied for MD simulation. The results of MD simulation including the structural snapshots of the reaction and the fluctuations of electronic energy are displayed in **Figure R26**. The adsorbed H₂O₂ on FeOF could be readily cleaved into two •OH radicals without energy barriers to overcome. In contrast, the adsorbed H₂O₂ on FeOCl cannot split into •OH radicals. The divergence in H₂O₂ split performance between FeOF and FeOCl further highlights the unique function of fluorine coordination. Interestingly, our observations revealed an anomalous surface reconstruction of the (211) facet after Fenton catalysis, which could be potentially linked to the self-enhancement. Consequently, we hypothesize that the self-enhancement effect could be triggered by the excessive H₂O₂ through a surface reconstruction. However, this effect needs further investigation and validation.

To clarify this result, we have revised the manuscript description as follows.

Line 390-395, “We conducted *ab initio* molecular dynamics (MD) simulations to further unveil the evolution and fate of H₂O₂, detailed in Fig. S22. H₂O₂ adsorbed on FeOF could be readily cleaved into two •OH radicals without energy barriers to overcome. Conversely, the adsorbed

H₂O₂ on FeOCl cannot split into •OH radicals. The divergence in H₂O₂ split performance between FeOF and FeOCl further highlights the unique function of fluorine coordination.”

Figure R25. Turnover of Fe(II)–Fe(III) in FeOCl catalyst upon reaction with H₂O₂ (2).

Figure R26. Evolution of temperature and energy during the ab initio molecular dynamics (MD) simulation for H₂O₂ activation onto FeOCl (A) and FeOF (B). The brown, red, green blue, and white balls denote Fe, O, C, Cl, F, and H atoms, respectively.

References:

1. Sun, M.; Chu, C.; Geng, F.; Lu, X.; Qu, J.; Crittenden, J.; Elimelech, M.; Kim, J.-H., Reinventing Fenton Chemistry: Iron Oxychloride Nanosheet for pH-Insensitive H₂O₂ Activation. *Environmental Science & Technology Letters* **2018**, 5, (3), 186-191.
2. Zhang, S.; Hedtke, T.; Zhu, Q.; Sun, M.; Weon, S.; Zhao, Y.; Stavitski, E.; Elimelech, M.; Kim, J.-H., Membrane-Confined Iron Oxychloride Nanocatalysts for Highly Efficient Heterogeneous Fenton Water Treatment. *Environ. Sci. Technol.* **2021**, 55, (13), 9266-9275.

Comment 6: *L 415 methane or methanol?*

Response: We appreciate such careful review. The word of “methane” has been revised into “methanol”. In order to avoid similar typo and other mistakes, we thoroughly rechecked through the manuscript, and made following revisions.

Line 191-194, “As shown in Fig. 2A, Fe K-edge X-ray absorption near-edge structure (XANES) spectra indicate that the absorption energy of FeOF was evidently dissociate to Fe foil but intimate to Fe₂O₃ and FeF₃, implying that Fe species in FeOF carries positive charge toward +3 (39).”

Line 197-200, “Moreover, the lower normalized intensity of white peak locating at near-edge (7130.2 eV) in the XANES spectrum of FeOF implied the five-fold coordination by F and O atoms for unsaturated coordination iron sites (46, 48).”

Line 220-222, “This observation justified the characteristic moiety of F-(Fe(III)O₃)-F in the fine local structure elucidation of FeOF, which was consistent with Mössbauer spectroscopy and XANES analysis.”

Line 439-442, “Similar to •OH generation inhibition, we observed a strong suppression of 4-NP decay with almost zero percent degradation rate (Fig. 5C), suggesting that unsaturated iron centers (*e.g.*, F-(Fe(III)O₃)-F moiety) is responsible for 4-NP destruction.”

Line 442-444, “Moreover, we also observed that both methanol and tert-butanol, two representative scavengers for •OH, showcased a severe inhibition effect in 4-NP removal at an excessive dosage (Fig. S32).”

Line 498-499, “Together with the high reactivity, the as-prepared FeOF catalyst can be operated at more adaptive pH range than the state-of-the-art Fenton catalysts.”

Line 581-583, “For the •OH scavenging experiment, MeOH (10 mM) and TBA (10 mM) were used to quench •OH (*i.e.*, •OH_f and •OH_{surf}) and •OH_f radicals, respectively (63).”

Line 583-589, “The cyclic experiments (with or without adding H₂O₂), Fe leaching experiments, coexisting ion (*i.e.*, Cl⁻) experiments, pH (*i.e.*, 3.0, 4.0, 5.0, 6.0, 7.0, 8.0, and 9.0) experiments, H₂O₂ dosage (1.0, 5.0, 10.0, 15.0, and 20.0 mM) experiments, temperature experiments (*i.e.*, 20, 30, and 40 °C), contaminants (*i.e.*, tetracycline, atrazine, sulfamethoxazole, ofloxacin, and salicylic acid) experiments, and industrial effluents experiments were conducted to explore the practical considerations of pH-adaptive FeOF/H₂O₂ Fenton system.”

Comment 7: *I think the below comment is somewhat unfair since every world has multiple aspects, and comparison should lie on the fair basis. please check “For instance, the K⁺ intercalation into FeOCl alters the H₂O₂ activation pathway to generate Fe(IV)=O species rather than •OH radicals (30), which compromises the overall water treatment efficiency because of its high 78 selectivity toward organic substrate oxidation.”*

Response: We express our gratitude to the reviewer for this comment. To clarify this interesting work more reasonably, we have revised the manuscript description as follows.

Line 75-78, “Interestingly, the K⁺ intercalation into FeOCl alters the H₂O₂ activation pathway from •OH radicals to Fe(IV)=O species with improved tolerance to water matrix while the oxidation efficiency may be partially compromised (30).”

REVIEWER COMMENTS

Reviewer #1 (Remarks to the Author):

In the revised manuscript, the authors have well addressed the issues and presented new results to support their claims. It can be accepted for publication.

Reviewer #2 (Remarks to the Author):

The authors have made great effort to revise this manuscript and all of my comments have been addressed properly. Additionally, they also provided serious and reasonable responses to the comments of other reviewers. From the novelty of the manuscript and their attitude towards the revision, I recommend this work for publication.

Reviewer #3 (Remarks to the Author):

Below are some additional suggestions recommended for further clarification of the results.

Please label XRD peaks to indicate which crystalline facets they denote. Also, provide the reason why XRD peaks for (111) and (211) crystalline facets were shifted from 40.245 and 53.344 (for PDF#70-1522), respectively.

The used PDF card (PDF#70-1522) seems to be an outdated version assigned to FeOF. The FeOF PDF card (PDF#04-009-3904) retrieved from the ICSD-FIZ database indicates that a 2.24 Å peak corresponds to the (111) crystalline phase. Both did not match with the lattice fringe shown in the TEM image, namely 2.38 Å (Figure 1 and Figure S2).

In the XPS spectrum (Figure 1), the differences between the FeOCl and FeOF do not seem to be limited to a shift caused by electron migration. The shift for Fe 2p_{1/2} looks more drastic than Fe 2p_{3/2}. Can this be related to the coordination environment of Fe?

Since the author used Py-IR as a critical tool to quantify the amount of Lewis acid on the catalysts, more references or details on how it was carried out should be included. In particular, peaks located at 1450 and 1490 cm⁻¹ should be exclusively indicative of the coordination structure formed between the Lewis

acid on catalyst and the pyridine, not including the peak at 1609 cm⁻¹. Furthermore, Py-IR may not be suitable for quantitative analysis, which should cooperate with a NH₃-TPD test.

The authors should consider XAS of FeOCl for better comparison since only one sample (FeOF) was tested by the authors, while the other reference spectra were retrieved from database. Would the authors be able to provide a comparison with FeOCl at least using previously reported XAS?

In the valence-electron density maps shown in Figure S9, F and Cl both should have the strongest negative signal. Why do they display opposite colors in the diagram?

The selected two depicters at four typical facets, i.e., (101), (110), (211), and (010), of FeOF and FeOCl, are not matching with the characterization results. Only (111) and (211) of FeOF were mentioned in material characterization section (Figure 1). It is recommended to provide further information of why the authors chose these four facets.

The conclusion “high proportion of medium and strong Lewis acid sites that can easily trigger H₂O₂ activation” in Line 171 is confusing, since H₂O₂ molecule requires electrons to activate its peroxide bond. H₂O₂ can be regarded as a Lewis base when electron transfer is not involved.

As suggested in Line 477-486, if Fe(II) is accumulated during Fenton reaction increasing up to 42% only in 30 min, would there be a point where the Fenton reaction is hampered by the slow regeneration of Fe(III)? Would the electronegativity of halide component (Cl and F) impact the kinetics of Fe(II)/Fe(III) redox?

The flow logic in Lines 312-319 is difficult to comprehend. If the discrepancy in Lewis acidity between FeOCl and FeOF (3 times higher) accounts for only a minimal part of the hydroxyl radical generation (168 times higher), there appears to be a more important factor than Lewis acidity. The fluorine coordination does not seem to play a great role than Lewis acidity, given that the authors assumed the chemical structure of FeOCl and FeOF is identical, except for the halide component. Further explanation is required.

Reviewer #4 (Remarks to the Author):

The authors have carefully addressed my early comments and I also shared their response to the comments from other reviewers. Now on my side I have no further comments and the revised work is acceptable in possible publication in Nature Communications.

Reviewer #1

General comment: *In the revised manuscript, the authors have well addressed the issues and presented new results to support their claims. It can be accepted for publication.*

Response: Thank you for the thorough review and valuable feedback on our manuscript. We appreciate the time and effort you dedicated to providing constructive comments. Your insights have been immensely helpful in improving the quality and clarity of our work.

Reviewer #2

General comment: *The authors have made great effort to revise this manuscript and all of my comments have been addressed properly. Additionally, they also provided serious and reasonable responses to the comments of other reviewers. From the novelty of the manuscript and their attitude towards the revision, I recommend this work for publication.*

Response: Thank you for the thorough review and valuable feedback on our manuscript. We appreciate the time and effort you dedicated to providing constructive comments. Your insights have been immensely helpful in improving the quality and clarity of our work.

Reviewer #3

General comment: *Below are some additional suggestions recommended for further clarification of the results.*

Response: We express our heartfelt appreciation to the referee for offering additional suggestions to further enhance our work. We kindly request the reviewer to consider the revised manuscript for potential publication in the journal. Your continued guidance is invaluable, and we are grateful for your time and expertise in this process.

Comment 1: *Please label XRD peaks to indicate which crystalline facets they denote. Also, provide the reason why XRD peaks for (111) and (211) crystalline facets were shifted from 40.245 and 53.344 (for PDF#70-1522), respectively.*

Response: We are grateful for your valuable suggestions. In **Figure R1**, the XRD peaks have been labeled with assigned crystalline facets. As supported by previous reports, the slight downshift observed in the XRD peaks on FeOF can be attributed to the expansion of the crystal lattice (1, 2). Furthermore, it has been reported that the presence of oxygen vacancies could induce the expansion of the crystal lattice (3, 4). To this end, we propose that the observed downshifts of (111) and (211) facets might be assigned to the presence of oxygen vacancies.

To clarify this result, we have revised the manuscript description as follows.

Line 122-124, “The downshift of (111) and (211) facets in the recorded pattern might be assigned to the presence of oxygen vacancy that expands the crystal lattice (39-41).”

Line 806 Fig. 1B,

Fig. 1. Preparation and structural characterization of FeOF catalyst. (A) Schematic illustration for the preparation procedure of nanostructured FeOF catalyst. (B) XRD pattern, (C) HRTEM (inset: SAED image) and (D) HAADF-STEM images of FeOF catalyst. (E) High-resolution Fe 2p XPS spectra of FeOF and FeOCl catalysts. (F) Comparison of the Lewis acidity of FeOF and catalysts with different strength. (G) ^{57}Fe Mössbauer spectroscopy of FeOF.”

Figure R1. XRD pattern of FeOF catalyst with marks for assigned facets.

References:

1. Yun, B.; Maulana, A. Y.; Lee, D.; Song, J.; Futralan, C. M.; Moon, D.; Kim, J., The Effect of Ni Doping on FeOF Cathode Material for High-Performance Sodium-Ion Batteries. *Small* **2023**, 2308011.
2. Li, W.; Chen, Y.; Zangiabadi, A.; Li, Z.; Xiao, X.; Huang, W.; Cheng, Q.; Lou, S.; Zhang, H.; Cao, A.; Roy, X.; Yang, Y., FeOF/TiO₂ Hetero-Nanostructures for High-Areal-Capacity Fluoride Cathodes. *ACS Appl. Mat. Interfaces* **2020**, 12, (30), 33803-33809.
3. Yao, L.; Inkinen, S.; van Dijken, S., Direct observation of oxygen vacancy-driven structural and resistive phase transitions in La_{2.3}Sr_{1.3}MnO₃. *Nature Communications* **2017**, 8, 14544.
4. Juan, D.; Pruneda, M.; Ferrari, V., Localized electronic vacancy level and its effect on the properties of doped manganites. *Scientific Reports* **2021**, 11, 6701.

Comment 2: *The used PDF card (PDF#70-1522) seems to be an outdated version assigned to FeOF. The FeOF PDF card (PDF#04-009-3904) retrieved from the ICSD-FIZ database indicates that a 2.24 Å peak corresponds to the (111) crystalline phase. Both did not match with the lattice fringe shown in the TEM image, namely 2.38 Å (Figure 1 and Figure S2).*

Response: We express our gratitude to the reviewer for providing insightful comments. In response to your suggestion, we conducted a comprehensive document retrieval of recent works to determine the prevalent type of PDF card used for FeOF. Our findings indicate that over the past five years, researchers have consistently employed PDF#70-1522 for their analyses, aligning with the conventions observed in contemporary literatures (1-5). Consequently, considering the consistent usage of PDF#70-1522 in recent literatures, it would be reasonable to evaluate the crystal structure of the prepared FeOF using this specific PDF card.

Additionally, upon reevaluation, we identified an error in the assignment of the d-spacing for the (111) facet. The correct values from PDF#70-1522 are 2.24 Å for (111) and 2.40 Å for (211) facets. Consequently, we have revised Fig. 1 (**Figure R2**) and Fig. S2 (**Figure R3**) accordingly to accurately reflect these adjustments.

To clarify this result, we have revised the manuscript description as follows.

Line 124-127, “As illustrated by high-resolution transmission electron microscope (HRTEM) image in Fig. 1C, a lattice finger distance of 2.24 Å and 1.71 Å (Fig. S2B and Fig. 1C) representing the (111) and (211) facets of FeOF can be observed (38), further supporting the XRD result.”

Line 806 Fig. 1C,

Fig. 1. Preparation and structural characterization of FeOF catalyst. (A) Schematic illustration for the preparation procedure of nanostructured FeOF catalyst. (B) XRD pattern, (C) HRTEM (inset: SAED image) and (D) HAADF-STEM images of FeOF catalyst. (E) High-resolution Fe 2p XPS spectra of FeOF and FeOCl catalysts. (F) Comparison of the Lewis acidity of FeOF and catalysts with different strength. (G) ^{57}Fe Mössbauer spectroscopy of FeOF.”

Page 7 in Supplementary Information, “Fig. S2. (A) HRTEM image of FeOF catalyst with highlights of (211) and (210) facets. (B) Intensity profile of (111) facet in Fig. 1C indicates a d spacing of 2.24 Å. (C) Intensity profile of (111) facet in Fig. S2A indicates a d spacing of 2.38 Å for (210) facet.”

Figure R2. HRTEM (inset: SAED image) of FeOF catalyst.

Figure R3. (A) HRTEM image of FeOF catalyst with highlights of (211) and (210) facets. (B) Intensity profile of (111) facet in Fig. 1C indicates a d spacing of 2.24 Å. (C) Intensity profile of (111) facet in Fig. S2A indicates a d spacing of 2.38 Å for (210) facet.

References:

- Yun, B.; Maulana, A. Y.; Lee, D.; Song, J.; Futralan, C. M.; Moon, D.; Kim, J., The Effect of Ni Doping on FeOF Cathode Material for High-Performance Sodium-Ion Batteries. *Small* **2023**, 2308011.
- Zhai, J.; Lei, Z.; Sun, K.; Zhu, S., MXene enabled binder-free FeOF cathode with high volumetric and gravimetric capacities for flexible lithium ion batteries. *Electrochim. Acta* **2022**, 423.
- Li, W.; Chen, Y.; Zangiabadi, A.; Li, Z.; Xiao, X.; Huang, W.; Cheng, Q.; Lou, S.; Zhang, H.; Cao, A.; Roy, X.; Yang, Y., FeOF/TiO₂ Hetero-Nanostructures for High-Areal-Capacity Fluoride Cathodes. *ACS Appl. Mat. Interfaces* **2020**, 12, (30), 33803-33809.
- Zhou, H.; Zhao, Y.; Lu, Q.; Chen, S.; Chen, M.; Wen, N.; Kuang, Q.; Fan, Q.; Dong, Y., Phosphorus-Doped FeOF Nanoparticle-Based Cathodes for Lithium Storage. *ACS Applied Nano Materials* **2022**, 5, (9), 13444-13454..
- Wang, M.; Li, Z.; Wang, C.; Zhao, R.; Li, C.; Guo, D.; Zhang, L.; Yin, L., Novel Core-Shell FeOF/Ni(OH)₂ Hierarchical Nanostructure for All-Solid-State Flexible Supercapacitors with Enhanced Performance. *Adv. Funct. Mater.* **2017**, 27, (31).

Comment 3: *In the XPS spectrum (Figure 1), the differences between the FeOCl and FeOF do not seem to be limited to a shift caused by electron migration. The shift for Fe 2p_{1/2} looks more drastic than Fe 2p_{3/2}. Can this be related to the coordination environment of Fe?*

Response: We appreciate your valuable comment. In the results of XPS analysis, the upshifts for Fe 2p_{3/2} and Fe 2p_{1/2} were found to be 1.0 eV and 0.5 eV, respectively. The XPS technique, widely employed for investigating the electronic structure of catalysts, proves instrumental in discerning the chemical environment of the target element. This method not only facilitates the identification of elements within a sample but also provides valuable insights into the chemical state and electronic properties of those elements in the sample (1). The binding energies obtained through XPS are known to be highly sensitive to the chemical environment of the atom under investigation. Generally, binding energy increases as the electron density of a specific atom decreases. Marshall-Roth *et al.* have previously reported that a strongly electron-withdrawing environment in the material can account for the positive shift observed in the putative Fe(III) component of the Fe-N-C active site (2). Similarly, Conradie *et al.* have demonstrated that the Fe 2p photoelectron lines, as measured by X-ray photoelectron spectra (XPS), exhibited sensitivity to the electron-donating and withdrawing properties of ligands attached to the iron atom in a series of tris(β -diketonato) iron(III) complexes. The maximum binding energy of the Fe 2p_{3/2} envelope revealed a linear relationship with the combined Gordy scale group electronegativity of the R-groups substituted on the respective β -diketonato ligand (RCOCHCOR'). As the number of CF₃ groups increased, the maximum Fe 2p_{3/2} envelope (eV) ranged from 710.69 to 711.11 eV. (3). Based on this analysis, we deduced that the peaks corresponding to iron species in FeOF exhibited a noticeable shift to higher binding energies compared to those in FeOCl. This shift suggests a significant reduction in electron density around the iron sites due to fluorine coordination (4-6).

Nevertheless, there is a lack of discussions regarding the shift diversity over Fe 2p_{3/2} and Fe 2p_{1/2} in the existing literatures. We believe that it would be valuable to undertake a systematic investigation in the future to elucidate the linkages between shift diversity and the coordination environment.

References:

1. Conradie, J.; Erasmus, E., XPS Fe 2p peaks from iron tris(β -diketonates): Electronic effect of the β -diketonato ligand. *Polyhedron* **2016**, 119, 142-150.
2. Marshall-Roth, T.; Libretto, N. J.; Wrobel, A. T.; Anderton, K. J.; Pegis, M. L.; Ricke, N. D.; Van Voorhis, T.; Miller, J. T.; Surendranath, Y., A pyridinic Fe-

- N₄ macrocycle models the active sites in Fe/N-doped carbon electrocatalysts. *Nature Communications* **2020**, 11, 5283.
- Erasmus, E., X-ray photoelectron spectroscopy: Charge transfer in Fe 2p peaks and inner-sphere reorganization of ferrocenyl-containing chalcones. *J. Electron. Spectrosc. Relat. Phenom.* **2018**, 223, 84-88.
 - H. Li et al., Wavelength-dependent differences in photocatalytic performance between BiOBr nanosheets with dominant exposed (001) and (010) facets. *Appl. Catal. B: Environ.* **2016**, 187, 342-349.
 - H. Liang et al., Synergistic effect of dual sites on bimetal-organic frameworks for highly efficient peroxide activation. *J. Hazard. Mater.* **2021**, 406, 124692.
 - X. Zarate, E. Schott, R. Arratia-Perez, Effects of the peripheral substituents (-NH₂, -OH, -CH₃, -H, -C₆H₅, -Cl, -CO₂H and -NO₂) on molecular properties of a Ni-Porphyrzine dimers family. *Polyhedron* **2013**, 50, 131-138.

Comment 4: *Since the author used Py-IR as a critical tool to quantify the amount of Lewis acid on the catalysts, more references or details on how it was carried out should be included. In particular, peaks located at 1450 and 1490 cm⁻¹ should be exclusively indicative of the coordination structure formed between the Lewis acid on catalyst and the pyridine, not including the peak at 1609 cm⁻¹. Furthermore, Py-IR may not be suitable for quantitative analysis, which should cooperate with a NH₃-TPD test?*

Response: We express our gratitude to the reviewer for this constructive suggestion. IR spectroscopy is a powerful tool for identifying the nature (type and strength) of acid/base sites (1-12). Acid and base strength of solid catalysts can be estimated from the profile of absorbance versus desorption temperature (5). Experimental determination of the integrated molar extinction coefficient enables to estimate the number of acid and basic sites. The quantification of Lewis acid amount was based on the C. A. Emeis's report on Journal of Catalysis in 1993, titled "Determination of Integrated Molar Extinction Coefficients for Infrared Absorption Bands of Pyridine Adsorbed on Solid Acid Catalysts" (5). To prevent any potential misunderstanding, we have included comprehensive details on how Lewis acidity was quantified in the Supplementary Information, accompanied by relevant references. Additionally, the assignment of 1609 cm⁻¹ for the Lewis acid site has been removed, and the discussion of characteristic peaks for identifying the Lewis acid site has been revised. Numerous reports in the literature highlight the utilization of Py-IR for the quantitative assessment of Lewis acidity in solid catalysts (6-12). As an illustration, Wang *et al.* utilized Py-IR to confirm and quantify the presence of Lewis acid sites on mortise-tenon zeolite catalyst. This study employed the molar integral extinction coefficients of 1.42 cm μmol⁻¹ for Lewis acid sites, as suggested by Emeis's report (6). Pan *et al.* conducted an analysis

and quantification of zeolite acidity using the Py-IR technique, examining both Lewis and Brønsted acid sites (7). Furthermore, Tamura *et al.* provided a comprehensive discussion on the infrared (IR) technique for evaluating the acid/base properties of metal oxides (9).

To clarify this result, we have revised the manuscript description as follows.

Line 157-159, “It can be found that Py-IR spectra for FeOF (Fig. S8A) and FeOCl (Fig. S8B) exhibit two typical IR bands at 1445 and 1480 cm^{-1} , which were all attributed to a pyridine-Lewis acid adduct (47-49)”

Line 165-168, “For a more direct comparison, the Lewis acidity of different acid types including weak, medium, and strong, revealed by the Py-IR spectra recorded at different degassing temperatures, was quantified according to previous reports (47, 50, 51), and summarized in Fig. 1F.”

Page 2 in Supplementary Information, “Text S2. Measurement procedure for and quantification details for Lewis acid sites. The Py-IR spectra of prepared catalysts were recorded using a Thermo Nicolet iS10 FTIR spectrometer equipped with KBr windows. Prior to recording, the sample was pressed into a self-supporting wafer and activated in vacuum ($<10^{-4}$ Pa) at 200 °C for 1 h. For pyridine adsorption, the samples were exposed to pyridine for 1 h after cooling down, then the IR transmission cell was evacuated to vacuum. The spectra were collected at degassing temperatures of 50, 100, and 200 °C to verify the type of acid sites. For the quantitative comparison, the area of peak at 1445 cm^{-1} and a molar absorption coefficient of 1.42 $\text{cm} \mu\text{mol}^{-1}$ were used to calculate the Lewis acidity, following the methodology established in previous reports (1-3).”

References:

1. Mo, F.; Song, C.; Zhou, Q.; Xue, W.; Ouyang, S.; Wang, Q.; Hou, Z.; Wang, S.; Wang, J., The optimized Fenton-like activity of Fe single-atom sites by Fe atomic clusters-mediated electronic configuration modulation. *PNAS* **2023**, *120*, e2300281120.
2. Busca, G., Bases and Basic Materials in Chemical and Environmental Processes. Liquid versus Solid Basicity. *Chem. Rev.* **2010**, *110*, (4), 2217-2249.
3. Busca, G., Acid catalysts in industrial hydrocarbon chemistry. *Chem. Rev.* **2007**, *107*, (11), 5366-5410.
4. Paukshtis, E. A.; Yurchenko, É. N., Study of the Acid–Base Properties of Heterogeneous Catalysts by Infrared Spectroscopy. *Russ. Chem. Rev.* **1983**, *52*, 242-258.
5. Emeis, C. A., Determination of Integrated Molar Extinction Coefficients for IR Absorption Bands of Pyridine Adsorbed on Solid Acid Catalysts. *ChemInform* **1993**, *24*.

6. Wang, H.; Shen, B.; Chen, X.; Xiong, H.; Wang, H.; Song, W.; Cui, C.; Wei, F.; Qian, W., Modulating inherent lewis acidity at the intergrowth interface of mortise-tenon zeolite catalyst. *Nature Communications* **2022**, *13*, (1).
7. Pan, Z.; Puente-Urbina, A.; Batool, S. R.; Bodi, A.; Wu, X.; Zhang, Z.; van Bokhoven, J. A.; Hemberger, P., Tuning the zeolite acidity enables selectivity control by suppressing ketene formation in lignin catalytic pyrolysis. *Nature Communications* **2023**, *14*, (1).
8. Jia, Z.; Mao, W.; Bai, Y.; Wang, B.; Ma, H.; Li, C.; Lu, J., Hollow nano-MgF₂ supported catalysts: Highly active and stable in gas-phase dehydrofluorination of 1,1,1,3,3-pentafluoropropane. *Applied Catalysis B-Environmental* **2018**, *238*, 599-608.
9. Tamura, M.; Shimizu, K.-i.; Satsuma, A., Comprehensive IR study on acid/base properties of metal oxides. *Applied Catalysis a-General* **2012**, *433*, 135-145.
10. Xia, H.; Hu, H.; Xu, S.; Xiao, K.; Zuo, S., Catalytic conversion of glucose to 5-hydroxymethylfural over Fe/ β zeolites with extra-framework isolated Fe species in a biphasic reaction system. *Biomass & Bioenergy* **2018**, *108*, 426432.
11. Loveless, B. T.; Gyanani, A.; Muggli, D. S., Discrepancy between TPD- and FTIR-based measurements of Bronsted and Lewis acidity for sulfated zirconia. *Applied Catalysis B-Environmental* **2008**, *84*, (3-4), 591-597.
12. Batool, S. R.; Sushkevich, V. L.; van Bokhoven, J. A., Correlating Lewis acid activity to extra-framework aluminum species in zeolite Y introduced by Ion-exchange. *J. Catal.* **2022**, *408*, 24-35.

Comment 5: *The authors should consider XAS of FeOCl for better comparison since only one sample (FeOF) was tested by the authors, while the other reference spectra were retrieved from database. Would the authors be able to provide a comparison with FeOCl at least using previously reported XAS?*

Response: We appreciate your valuable comment. In accordance with the report by Xu *et al.* (1), the chemical state and coordination environment of FeOCl were analyzed using the Fe K-edge X-ray absorption fine structure (XAFS) spectra, providing a comprehensive understanding of these aspects. Based on the XANES spectra (**Figure R4A**) of FeOCl, the highest K-edge absorption energy of Fe appeared at 7133.4 eV, which was located between Fe₃O₄ (7132.32 eV) and Fe₂O₃ (7134.48 eV), proving the coexistence of Fe(II) and Fe(III) in FeOCl and the average valence between Fe₃O₄ and Fe₂O₃. The Fourier transform (FT) EXAFS R-space fitting curve of FeOCl (**Figure R4A**) elucidates the Fe coordination with O and Cl. The peak at ~1.44 Å can be assigned to the first shell of Fe-O and Fe-Cl, as shown in the curve fitting of EXAFS results (**Figure R4B**). On average, each Fe

atom in FeOCl was coordinated by -3.5 O atoms and -1.8 Cl atoms. Similarly, each Fe atom in FeOF was coordinated by -2.7 O atoms and -1.9 F atoms. Considering the well-known stability of a coordination number of 6 for Fe atoms, it is suggested that there are 0.7 unoccupied coordination sites in the structure of FeOCl. In comparison, FeOF has more than 1.0 unoccupied coordination sites, emphasizing the presence of vacant iron sites in FeOF that play a crucial role in governing the efficiency of Fenton reactions.

To clarify this result, we have revised the manuscript description as follows.

Line 221-224, “The coordination numbers of F and O atoms for the iron site were determined to be 1.9 and 2.7 at distances of 1.99 and 1.90 Å, respectively. This coordination environment is akin to that observed in FeOCl, which demonstrated approximately 1.8 Cl atoms and 3.5 O atoms coordinated to the iron site in a previous report (58).”

Figure R4. (A) XANES spectra of FeOCl and other samples. (B) FT-EXAFS R-space fitting curve of fresh FeOCl (inset: crystal structure of FeOCl). Copied from Xu et al.’s report (1).

References:

1. Xu, X.; Zhang, S.; Wang, Y.; Lin, Y.; Guan, Q.; Chen, C., Identifying the Role of Surface Hydroxyl on FeOCl in Bridging Electron Transfer toward Efficient Persulfate Activation. *Environ. Sci. Technol.* **2023**, *57*, 12922-12930.

Comment 6: In the valence-electron density maps shown in Figure S9, F and Cl both should have the strongest negative signal. Why do they display opposite colors in the diagram?

Response: We sincerely appreciate the reviewer for the comment. The phenomenon of opposite colors displayed in Fig. S9 stems from the non-coincidence of atoms on the same plane during the creation of a two-dimensional cross-section. The colors can be further influenced by the central vacuum region.

This discrepancy arises from the process of plotting cross-sections. Moreover, to be noted, when examining valence states from this 2D diagram, we can only assess the charge quantity within the individual plots by comparing colors. It is unreasonable to contrast valence state differences between different plots based on the color shown in two diagrams. To address this concern, we have ensured that the specific data aligns with the valence states we have carefully marked. Additionally, we have updated a revised figure to enhance clarity for the reader, minimizing any potential confusion. We have also incorporated notations in the figure caption to provide essential clarity and context.

To clarify this result, we have revised the manuscript description as follows.

Page 13 in Supplementary Information,

Fig. S9. (A) Side view of the optimized molecular structure geometries for FeOF and FeOCl, in which the layer-by-layer architecture is well-recognized. (B) Corresponding two-dimensional valence-electron density color-filled maps. The simulated valence states of Fe, F, and Cl were highlighted in both plots. However, it is important to note that direct comparisons of valence states between different plots are not reasonable based solely on the colors displayed in the two diagrams.”

Comment 7: *The selected two depictees at four typical facets, i.e., (101), (110), (211), and (010), of FeOF and FeOCl, are not matching with the characterization results. Only (111) and (211) of FeOF were mentioned in material characterization section (Figure 1). It is recommended to provide further information of why the authors chose these four facets?*

Response: We sincerely thank the reviewer for the insightful comment. Four typical facets (*i.e.*, (101), (110), (211), and (010)) were selected because of the observed main facets with high peak intensity in the XRD patterns. Specifically, both of FeOF and FeOCl exhibit (101) and (110) facets while (211) and (010) belongs to FeOF and FeOCl, respectively. In order to provide a comprehensive comparison, in addition to the two common facets, we also included the (211) and (010) facets for simulations. Upon analyzing all tested facets, we observed that the fluorine coordination could reduce the electron density of iron sites and enhance the H₂O₂ adsorption energy, thereby promoting the Fenton reactions over FeOCl.

To clarify this result, we have revised the manuscript description as follows.

Line 336-338, “In this respect, we systematically calculated these two depictees at four main facets, *i.e.*, (101), (110), (211), and (010), in FeOF and FeOCl, which were extracted from the XRD patterns with high peak intensity.”

Comment 8: *The conclusion “high proportion of medium and strong Lewis acid sites that can easily trigger H₂O₂ activation” in Line 171 is confusing, since H₂O₂ molecule requires electrons to activate its peroxide bond. H₂O₂ can be regarded as a Lewis base when electron transfer is not involved.*

Response: We express our gratitude to the reviewer for this impressive comment. Regarding heterogenous Fenton catalysis, the interaction between H₂O₂ and active sites is believed as the initial step for H₂O₂ activation. As the oxidants, such as H₂O₂ and PMS, are Lewis base, the Lewis acid-base interaction plays a crucial role in facilitating the accessibility to active sites with strong affinity (1-5). Upon binding of H₂O₂ to the active site, a subsequent electron transfer is expected to occur between the active site and oxidant molecule (1-3).

In the case of FeOCl, the proposed mechanism for H₂O₂ activation is depicted in the redox cycle shown in **Figure R5** (6-7). The remarkable catalytic property of FeOCl has been attributed to the favorable reduction of Fe(III) to Fe(II) compared to other catalysts. It is important to note that the Fe(III)-to-Fe(II) reduction is significantly slower than the Fe(II)-to-Fe(III) oxidation, rendering it the rate-limiting step of the redox cycle (8). FeOCl expedites the swift conversion of Fe(III) to Fe(II) through the H₂O₂ dehydrogenation step, initiating the energetically favorable reduction of Fe(III) to Fe(II) (9). The resulting Fe(II) subsequently reacts with H₂O₂ to generate •OH, completing the redox cycle involving the Fe(III)/Fe(II) pair.

Similar to the FeOCl, in Fenton reactions involving transition metal catalysts (Mn⁺²/Mⁿ), the reaction depends on the reduction of Mⁿ⁺² to Mⁿ by H₂O₂ and the

oxidation of M^n to M^{n+2} along with $\bullet\text{OH}$ generation (10). Li *et al.* reported that the Lewis acidity of the FeCu catalyst was greatly improved after modification with Al_2O_3 (2). The catalyst's electron-attracting capability was enhanced, facilitating the easy reaction of active metals with H_2O_2 and their subsequent reduction for pollutant degradation. Based on the above discussion, in the FeOF-involved Fenton system, H_2O_2 functions as both a reductant and oxidant, driving Fe(III) reduction and Fe(II) oxidation sustainably within the Fe(III)/Fe(II) cycle to generate $\bullet\text{OH}$ for pollutant degradation.

To clarify this result, we have revised the manuscript description as follows.

Line 172-175, “Therefore, it is supposed that FeOF is more favorable for Fenton catalysis as it contains high proportion of medium and strong Lewis acid sites that can facilitate the interaction with H_2O_2 for sustainable activation through the enhanced the Fe(III)/Fe(II) redox cycle (32, 52-54).”

Figure R5. Proposed mechanism for FeOCl-mediated activation of H_2O_2 with $\bullet\text{OH}$ production. Copied from Zhang *et al.*'s report (7).

References:

1. Duan, L.; Jiang, H.; Wu, W.; Lin, D.; Yang, K., Defective iron based metal-organic frameworks derived from zero-valent iron for highly efficient Fenton-like catalysis. *J. Hazard. Mater.* **2023**, *445*, 130426.
2. Li, X.; Shen, C.; Ma, J.; Wen, Y., The strong promoting effects of thin layer Al_2O_3 on FeCu Fenton-like components: Enhanced electron transfer. *Sci. Total Environ.* **2022**, *821*, 153151.
3. Mo, F.; Song, C.; Zhou, Q.; Xue, W.; Ouyang, S.; Wang, Q.; Hou, Z.; Wang, S.; Wang, J., The optimized Fenton-like activity of Fe single-atom sites by Fe

- atomic clusters-mediated electronic configuration modulation. *PNAS* **2023**, *120*, e2300281120.
- Zhang, S.; Zhuo, Y.; Ezugwu, C. I.; Wang, C.-c.; Li, C.; Liu, S., Synergetic Molecular Oxygen Activation and Catalytic Oxidation of Formaldehyde over Defective MIL-88B(Fe) Nanorods at Room Temperature. *Environ. Sci. Technol.* **2021**, *55*, (12), 8341-8350.
 - Yang, X.-j.; Tian, P.-f.; Zhang, X.-m.; Yu, X.; Wu, T.; Xu, J.; Han, Y.-f., The Generation of Hydroxyl Radicals by Hydrogen Peroxide Decomposition on FeOCl/SBA-15 Catalysts for Phenol Degradation. *AIChE J.* **2015**, *61*, (1), 166-176.
 - Sun, M.; Chu, C.; Geng, F.; Lu, X.; Qu, J.; Crittenden, J.; Elimelech, M.; Kim, J.-H., Reinventing Fenton Chemistry: Iron Oxychloride Nanosheet for pH-Insensitive H₂O₂ Activation. *Environmental Science & Technology Letters* **2018**, *5*, (3), 186-191.
 - Zhang, S.; Hedtke, T.; Zhu, Q.; Sun, M.; Weon, S.; Zhao, Y.; Stavitski, E.; Elimelech, M.; Kim, J.-H., Membrane-Confined Iron Oxychloride Nanocatalysts for Highly Efficient Heterogeneous Fenton Water Treatment. *Environ. Sci. Technol.* **2021**, *55*, (13), 9266-9275.
 - Xing, M.; Xu, W.; Dong, C.; Bai, Y.; Zeng, J.; Zhou, Y.; Zhang, J.; Yin, Y., Metal Sulfides as Excellent Co-catalysts for H₂O₂ Decomposition in Advanced Oxidation Processes. *Chem* **2018**, *4*, (6), 1359-1372.
 - Ji, X.-X.; Wang, H.-F.; Hu, P.-J., First principles study of Fenton reaction catalyzed by FeOCl: reaction mechanism and location of active site. *Rare Met.* **2019**, *38*, (8), 783-792.
 - Piera, J.; Backvall, J.-E., Catalytic oxidation of organic substrates by molecular oxygen and hydrogen peroxide by multistep electron transfer -: A biomimetic approach. *Angewandte Chemie-International Edition* **2008**, *47*, (19), 3506-3523.

Comment 9: As suggested in Line 477-486, if Fe(II) is accumulated during Fenton reaction increasing up to 42% only in 30 min, would there be a point where the Fenton reaction is hampered by the slow regeneration of Fe(III)? Would the electronegativity of halide component (Cl and F) impact the kinetics of Fe(II)/Fe(III) redox?

Response: We sincerely thank the reviewer for such thoughtful comment. If the Fenton reaction is hampered by the slow regeneration of Fe(III), the ratio of leached Fe(II) would continue to increase. Therefore, we have further monitored the Fe(II) concentration during Fenton reaction, extending reaction time to 120 min. As shown in **Figure R6**, the proportion of leached Fe(II) initially increases and then stabilizes at a ratio of about 66%, indicating that the regeneration of Fe(III) is not

the primary limiting step due to the equilibrium of the Fe(III)/Fe(II) cycle in the presence of H₂O₂. This equilibrium underscores the crucial role of Fe(II) generation in the Fenton reaction. Additionally, the sustainable generation of •OH from the Fenton reaction relies on the rapid oxidation of Fe(II), as demonstrated in previous reports (1-4). Interestingly, the coordination of halides (F and Cl) to iron sites contributes to an increase in the reduction potential of Fe centers. This enhancement facilitates more efficient single electron transfer from H₂O₂ during the Fe(III) to Fe(II) reduction step, as well as homolytic cleavage of H₂O₂ before the Fe(II) to Fe(III) oxidation step (**Figure R5**).

More importantly, we have evidenced that fluorine coordination exhibits a superior ability to enhance the reducibility of iron sites compared to chlorine coordination. It has been reported that the reversibility of the Fe(III)/Fe(II) cycle in the Fenton catalyst during the reaction can be assessed using electrochemical techniques (5). To address the concern about the facile reversibility of the Fe(III)/Fe(II) cycle, we conducted electrochemical studies, the open-circuit potential analysis and cyclic voltammetry measurements, to provide further insights for the comparison of FeOF and FeOCl. The open-circuit potential displayed in **Figure R7A** highlights a higher potential of Fe(III) in FeOF (0.142 V) than that in FeOCl (0.108 V), affirming the stronger activity to produce Fe(II) that results in the stronger oxidation capacity of the FeOF-catalyzed Fenton system (6). As shown in **Figure R7B**, obvious oxidation and reduction peaks appeared at 0.04 and 0.32 V with FeOF, implying the excellent reversibility of the Fe(III)/Fe(II) redox process. The fluorine coordination also leads to a decrease in peak-to-peak separation (ΔE_p) from 0.40 V to 0.28 V, indicating the improved Fe(III)/Fe(II) redox capability of FeOF (7). Furthermore, *in-situ* electrode potential analysis was performed to investigate the redox cycle of the iron center during the Fenton reaction. The addition of H₂O₂ triggers the increase in open-circuit potentials for both FeOF and FeOCl while the former exhibits a more significant increment (**Figure R7C**), which further elucidates the more facile reversibility of the Fe(II)/Fe(III) cycle with fluorine coordination (8). Collectively, these supplementary results well elucidate the presence of the more facile reversibility of Fe(III)/Fe(II) cycle in FeOF than FeOCl, which is crucial for the improved efficiency of the Fenton reaction.

To clarify this result, we have revised the manuscript description as follows.

Line 488-497, “We further performed XPS, Mössbauer, and electrochemical measurements to support the speculation of Fe(II) formation (Fig. S37, S38, and S39) (55, 56, 68). The results suggested that ~26.5% of Fe(II) was generated after Fenton process, which substantially facilitated the cycling of Fe(III)/Fe(II) redox as well as the degradation efficiency for 4-NP. The observation of higher open-circuit potential of Fe(III) in FeOF and smaller peak-to-peak separation (ΔE_p) (from

0.40 V to 0.28 V) depicted in Fig. S39 indicated the more facile reversibility of the Fe(III)/Fe(II) cycle of FeOF with fluorine coordination (69-71). Besides, we also found that the proportion of leaching Fe²⁺ ions increased from 3% to 42% (Fig. S39B), further evidencing the enhancement of Fe(III)/Fe(II) redox cycle.”

Page 43 in Supplementary Information, “The open-circuit potential displayed in Fig. S39A highlights a higher potential of Fe(III) in FeOF (0.142 V) than that in FeOCl (0.108 V), affirming the stronger activity to produce Fe(II) that results in the stronger oxidation capacity of the FeOF-catalyzed Fenton system (13). As shown in Fig. S39B, obvious oxidation and reduction peaks appeared at 0.04 and 0.32 V with FeOF, implying the excellent reversibility of the Fe(II)/Fe(III) redox process. The fluorine coordination also leads to a decrease in peak-to-peak separation (ΔE_p) from 0.40 V to 0.28 V, indicating the improved Fe(II)/Fe(III) redox capability of FeOF (14). Furthermore, the *in-situ* electrode potential analysis was performed to investigate the redox cycle of the iron center during the Fenton reaction. The addition of H₂O₂ triggers the increase in open-circuit potentials for both FeOF and FeOCl while the former exhibits a more substantial increment (Fig. S39C), which further elucidates the more facile reversibility of the Fe(II)/Fe(III) cycle with fluorine coordination (15). In addition, we found that the proportion of leaching Fe²⁺ ions increased from 3% to 42% (Fig. S13), indicating the enhancement of Fe(II)/Fe(III) redox cycle.”

Figure R6. Proportion of leached Fe³⁺ and Fe²⁺ ions in the FeOF/H₂O₂ system within 120 min. Note that the variation in the proportion observed during the first 30 minutes is within the range of experimental error when compared to the previous data.

Figure R7. (A) Open-circuit potential curves measurements, (B) cyclic voltammetry measurements, and (C) open-circuit potential curves with the in-situ addition of H_2O_2 on FeOF and FeOCl coated glassy carbon electrode.

References:

- Zhan, H.; Zhou, R.; Wang, P.; Zhou, Q., Selective hydroxyl generation for efficient pollutant degradation by electronic structure modulation at Fe sites. *Proceedings of the National Academy of Sciences* **2023**, *120*, (26), e2305378120.
- Liu, X.-C.; Zhang, K.-X.; Song, J.-S.; Zhou, G.-N.; Li, W.-Q.; Ding, R.-R.; Wang, J.; Zheng, X.; Wang, G.; Mu, Y., Tuning Fe_3O_4 for sustainable cathodic heterogeneous electro-Fenton catalysis by acetylated chitosan. *Proceedings of the National Academy of Sciences* **2023**, *120*, (13), e2213480120.
- Enami, S.; Sakamoto, Y.; Colussi, A. J., Fenton chemistry at aqueous interfaces. *Proceedings of the National Academy of Sciences* **2014**, *111*, (2), 623-628.
- Xu, X.; Zhang, Y.; Chen, Y.; Liu, C.; Wang, W.; Wang, J.; Huang, H.; Feng, J.; Li, Z.; Zou, Z., Revealing $^*\text{OOH}$ key intermediates and regulating H_2O_2 photoactivation by surface relaxation of Fenton-like catalysts. *Proceedings of the National Academy of Sciences* **2022**, *119*, (36), e2205562119.

Comment 10: *The flow logic in Lines 312-319 is difficult to comprehend. If the discrepancy in Lewis acidity between FeOCl and FeOF (3 times higher) accounts for only a minimal part of the hydroxyl radical generation (168 times higher), there appears to be a more important factor than Lewis acidity. The fluorine coordination does not seem to play a great role than Lewis acidity, given that the authors assumed the chemical structure of FeOCl and FeOF is identical, except for the halide component. Further explanation is required?*

Response: We express our gratitude to the reviewer for the excellent comment. There are three types of Lewis acid sites, including weak, medium, and strong. The Lewis acidity presented in the manuscript for comparison is an indicative of total amount of acid sites. This attempt aims to primarily exclude the effect of acid site amount on the $\bullet\text{OH}$ generation. Although our experimental findings suggested that FeOF had a somewhat higher Lewis acidity than FeOCl ($\sim 3\times$ higher), this alone

failed to explain the much higher observed disparity in the accumulated generation of •OH (~168× higher). Even if we normalized the apparent kinetic rate constant for •OH generation by the total Lewis acidity (Table S4 in the Supplementary Information), the value for FeOF was still 60 times greater than that for FeOCl. This observation highlights that there would be a more important factor for H₂O₂ activation rather than simplified by the amount of total Lewis acid sites.

In seeking a possible explanation, we observed a significant increase in the ratio of Lewis acid sites with strong strength, rising from 1.9% to 44.9% with the integration of fluorine coordination. Simultaneously, the ratio of Lewis acid sites with medium strength also increased by more than two times, as the weak strength acid sites disappeared. According to previous reports (1-3), the type of Lewis acid site is usually dependent on the electron density of metal sites that govern the interaction with Lewis base. The observed increased ratio of stronger acid sites verifies the electron-withdrawing function of fluorine coordination, which significantly reduces the electron density of iron sites. In other words, the stronger acid site is a representative of iron sites with more electron deficiency. It should be noted that the correlation between the enhancement of •OH generation and the increment in stronger acid sites may not follow a linear dependence. These findings support our deduction that the electron polarization induced by fluorine coordination governs the enhancement of Fenton catalysis through the optimization of the iron site configuration.

To clarify this result, we have revised the manuscript description as follows.

Line 321-328, “We observed that the ratio of Lewis acid sites with strong strength greatly increased from 1.9% to 44.9% as the fluorine coordination was integrated. Meanwhile, the ratio of Lewis acid sites with medium strength also increased by more than 2 times, accompanied by the disappearance of weak strength acid sites. According to previous reports (63-65), the type of Lewis acid site typically depends on the electron density of metal sites, which governs the interaction with Lewis bases. The observed increased ratio of stronger acid sites verifies the electron-withdrawing function of fluorine coordination.”

References:

1. Yang, Q.; Xu, W.; Gong, S.; Zheng, G.; Tian, Z.; Wen, Y.; Peng, L.; Zhang, L.; Lu, Z.; Chen, L., Atomically dispersed Lewis acid sites boost 2-electron oxygen reduction activity of carbon-based catalysts. *Nature Communications* **2020**, *11*, (1), 5478.
2. Lawrence, A. S.; Sivakumar, B.; Dhakshinamoorthy, A., Detecting Lewis acid sites in metal-organic frameworks by density functional theory. *Molecular Catalysis* **2022**, *517*, 112042.

3. Brown, I. D.; Skowron, A., Electronegativity and Lewis acid strength. *J. Am. Chem. Soc.* **1990**, *112*, 3401-3403.

Reviewer #4

General comment: *The authors have carefully addressed my early comments and I also shared their response to the comments from other reviewers. Now on my side I have no further comments and the revised work is acceptable in possible publication in Nature Communications.*

Response: Thank you for the thorough review and valuable feedback on our manuscript. We appreciate the time and effort you dedicated to providing constructive comments. Your insights have been immensely helpful in improving the quality and clarity of our work.

REVIEWERS' COMMENTS

Reviewer #3 (Remarks to the Author):

The authors went an extra mile to answer all my additional comments and I support the publication of this revised manuscript.

Response to Referees

Reviewer #3

General comment: *The authors went an extra mile to answer all my additional comments and I support the publication of this revised manuscript.*

Response: Thank you for the thorough review and valuable feedback on our manuscript. We appreciate the time and effort you dedicated to providing constructive comments.